# PaZO: Preconditioned Accelerated Zeroth-Order Optimization for Fine-Tuning LLMs

**Hanzhen Zhao**[1]
hzzhao@pku.edu.cn

**Shihong Ding**[1]
dingshihong@stu.pku.edu.cn

**Cong Fang**[1,2][†]
fangcong@pku.edu.cn

**Zhouchen Lin**[1,2,3][†]
zlin@pku.edu.cn

[1] State Key Lab of General AI, School of Intelligence Science and Technology, Peking University
[2] Institute for Artificial Intelligence, Peking University
[3] Pazhou Laboratory (Huangpu), Guangzhou, Guangdong, China

## Abstract

This paper introduces PaZO, a preconditioned accelerated zeroth-order optimization algorithm for fine-tuning large language models (LLMs). First, we theoretically demonstrate the necessity of preconditioning in zeroth-order optimization, proving that zeroth-order stochastic gradient descent (ZO-SGD) alone fails to achieve the ideal convergence rate. Building on this, we propose a Preconditioned Simultaneous Perturbation Stochastic Approximation (PSPSA) and theoretical version of PaZO, and demonstrate that setting the order of preconditioner as $-1/2$ in PSPSA yields the improved convergence rate for PaZO. Moreover, we design a practical version of PaZO that stabilizes training via diagonal Hessian estimate and moving average technique. Extensive experiments on diverse downstream tasks with models like RoBERTa-large and OPT show PaZO's effectiveness. Compared to other zeroth-order baselines, PaZO achieves better performance across models and tasks. Code is available at Code.

## 1 Introduction

Fine-tuning pre-trained large language models (LLMs) has become one of the dominant methodologies for adapting models to specialized downstream tasks [19] and aligning them with human instructional preferences [42]. However, as models are scaled up [1], the memory overhead extremely increases during fine-tuning, since computing gradients during backpropagation needs to cache model activations and historical gradients (e.g., for Adam-based optimization [28]). Parameter-efficient fine-tuning (PEFT) methods [29, 31, 23] reduce memory overhead by fine-tuning only a small number of extra parameters but still need to cache large quantities of activations. Recently, zeroth-order optimization algorithms (ZO) [37, 59, 58] have enabled the fine-tuning of LLMs with billions of parameters on a single consumer-grade GPU, due to their requirement for only forward passes to estimate gradients, without backpropagation and the storage of activations. Lightweight memory has solidified its role as a critical methodology for fine-tuning tasks in resource-constrained scenarios.

As research on zeroth-order optimization methods for fine-tuning LLMs advances, whether preconditioning zeroth-order algorithms with higher-order information can enhance optimization efficiency has become a pivotal challenge, since adaptive first-order optimizers such as Adam [28] and AdamW

---

[†]Corresponding author.

39th Conference on Neural Information Processing Systems (NeurIPS 2025).

[35], which can be regarded as preconditioned algorithms with $(\mathrm{diag}\{\mathbf{g}\circ\mathbf{g}\})^{-1/2}$ as a preconditioner, show improvement on convergence speed. However, for zeroth-order optimization, one cannot directly estimate the Hessian by first-order information. Direct adaptation of Adam to zeroth-order algorithms (e.g., ZO-Adam [58]) introduces large variances and has a significant impact on the fine-tuning performance [59]. Moreover, Hessian-informed perturbation for estimating zeroth-order information [59, 55] is a significant methodological advancement, but how to incorporate Hessian information into the perturbation process to obtain the best convergence speed and performance remains a significant challenge.

When we delve into and rethink the preconditioned zeroth-order optimization problems, the more pressing challenge lies in whether preconditioned zeroth-order optimization methods can truly achieve a provable convergence rate from a theoretical perspective. This problem may appear counterintuitive, but mature theoretical research [24, 17] on first-order methods has substantiated the following facts: for least squares regression, only SGD can achieve the near-optimal convergence rate $\tilde{\mathcal{O}}(d/T)$ and match the lower bound when ignoring the logarithmic term, which indicates that at least for this problem, preconditioning techniques provide no improvement on convergence, as SGD has already attained the information-theoretic limit of the problem. Therefore, whether this conclusion for zeroth-order optimization remains determines the effect of preconditioning techniques in zeroth-order optimization. Moreover, even if we posit that precondition holds effectiveness for zeroth-order optimization, how to appropriately apply preconditioning techniques emerges as another challenge. Specifically, determining the optimal order of the preconditioner to guarantee the fastest convergence rate becomes a critical consideration. Finally, from the practical perspective, how to estimate Hessian information through zeroth-order perturbation stochastic approximation to integrate abundant information, ensure stability and control memory overhead is also a challenge in practice. Based on the three above, we think that the following three problems demand reasonable resolution in preconditioned zeroth-order optimization for fine-tuning LLMs:

> *A.* Do we truly need preconditions in zeroth-order optimization?
>
> *B.* If the answer to question *A* is "yes", how to achieve the fastest convergence by selecting the optimal order of the preconditioner?
>
> *C.* How to effectively estimate Hessian information through zeroth-order perturbations in practice and improve fine-tuned model performance on downstream tasks?

In this paper, we provide reasonable answers to the three questions above. We propose a preconditioned accelerated zeroth-order optimization algorithm PaZO, with a theoretical guarantee to obtain a faster convergence rate by selecting the optimal order of preconditioner, and better empirical performance on a wide range of downstream tasks for fine-tuning LLMs. Our contributions are:

1. (Answer to Question *A*.) We construct a general Preconditioned Simultaneous Perturbation Stochastic Approximation (PSPSA) and corresponding algorithm PaZO (Theoretical Form 3.2) with any given order of Hessian information $\mathbf{H}^{-\alpha}$. Our theoretical analysis on quadratic functions in Theorem 3.5 demonstrates that only ZO-SGD ($\alpha = 0$) **cannot** achieve the fastest convergence rate. We need preconditions in zeroth-order optimization.

2. (Answer to Question *B*.) We provide the convergence analysis of PaZO for general objective functions. The result in Theorem 3.8 demonstrates that PaZO can achieve the fastest convergence rate if and only if we select $\alpha = 1/2$. In other words, we need to use $\mathbf{H}^{-1/2}$ in PSPSA (or $\mathbf{H}^{-1}$ as the preconditioner) to accelerate zeroth-order optimization.

3. (Answer to Question *C*.) We propose PaZO (Practical Form, Algorithm 1) for fine-tuning LLMs, with unbiased diagonal Hessian estimation incorporating current zeroth-order gradient information and moving average techniques to ensure stability in practice. We conduct extensive experiments across different models (RoBERTa-large, OPT-1.3B), different methods (FT, LoRA, prefix), and different downstream tasks to verify the effect of the PaZO. Results show PaZO achieves better performance across models, tasks and PEFT methods.

**Notations.** Let $\mathcal{O}(\cdot)$ and $\Omega(\cdot)$ denote upper and lower bounds, respectively, with a universal constant, while $\tilde{\mathcal{O}}(\cdot)$ and $\tilde{\Omega}(\cdot)$ ignore polylogarithmic dependencies. For functions $f$ and $g$: $f \lesssim g$ denotes $f = \tilde{\mathcal{O}}(g)$; $f \gtrsim g$ denotes $f = \tilde{\Omega}(g)$; $f \asymp g$ indicates $g \lesssim f \lesssim g$. We use $\lambda_{max}(\cdot)$ and $\lambda_{min}(\cdot)$ to

denote the largest and smallest eigenvalue of a matrix, respectively. Let $\|\boldsymbol{\theta}\|_{\mathbf{A}}$ denote the Mahalanobis (semi) norm where $\mathbf{A}$ is a positive semi-definite matrix as $\|\boldsymbol{\theta}\|_{\mathbf{A}} = \sqrt{\boldsymbol{\theta}^\top \mathbf{A} \boldsymbol{\theta}}$. We use $\boldsymbol{\theta}^*$ to denote the minimizer, i.e. $\boldsymbol{\theta}^* \triangleq \operatorname{argmin}_{\boldsymbol{\theta}} f(\boldsymbol{\theta})$.

## 2 Related Work

**Zeroth-order Optimization**: Zeroth-order optimization, is to estimate the gradient by just forward passes. A substantial body of theoretical research has been devoted to the detailed analysis of convergence rates in zeroth-order optimization in convex settings [3, 16, 26, 39, 44, 46] and non-convex [53]. Representative method SPSA [48] demonstrates strong performance in challenging settings like non-convex multi-agent optimization [21, 50] and black-box adversarial example generation [11, 10, 33]. Notably, MeZO [37] pioneers the adaptation of classical ZO-SGD for LLM fine-tuning, matching conventional performance while drastically cutting memory consumption. Then various following works [58, 59, 12, 49] try to improve zeroth-order optimizers for efficient fine-tuning. However, whether and how precondition works in zeroth-order optimization is still lack of discussion.

**Enhanced Optimizers with Hessian**: Researchers focus on how to incorporate second-order information to provide acceleration for gradient descent during the training. For example, [9, 40] utilized curvature information as the preconditioner; [38] applied diagonal Hessian as the preconditioner; [36] estimated the Hessian information with conjugate gradient. Sophia [32] introduced a lightweight estimate of the diagonal Hessian for pre-training. However, these methods can only be used for first-order methods with a heavy GPU-memory overhead. HiZOO [59] has been proposed as a preconditioned zeroth-order optimizer for fine-tuning LLMs. However, how to effectively leverage preconditioning information in zeroth-order optimization to accelerate convergence remains understudied.

## 3 Theoretical Insights of PaZO

Preconditioned methods in first-order optimization have been generally studied [40, 4, 28, 32]. However, few works discuss the necessity, potential and limitation of preconditioned zeroth-order optimization. In this section, we try to clarify two questions below from the theoretical perspective.

> *A.* Do we truly need preconditions in zeroth-order optimization?
>
> *B.* If the answer to question *A* is "yes", how to achieve the fastest convergence by selecting the optimal order of the preconditioner?

We provide theoretical insights into the two questions *A* and *B*. First, we show the necessity of using preconditions in zero-order optimization, since only ZO-SGD [48] cannot achieve the potential ideal convergence rate $\tilde{\mathcal{O}}(d^2/T)$ for least squares (as stated in Theorem 3.5), while the first-order SGD can match the optimal rate $\tilde{\mathcal{O}}(d/T)$ without preconditions [17]. This difference indicates that preconditions play a key role in ZO, especially. Second, we propose a general Preconditioned Simultaneous Perturbation Stochastic Approximation (PSPSA) using $\mathbf{H}^{-\alpha}$ as preconditioner with any given order $\alpha$ and Hessian $\mathbf{H}$ to extend traditional SPSA [48] for zeroth-order gradient estimate. We provide the convergence analysis of the preconditioned zeroth-order optimization with PSPSA in Theorem 3.8. The results explicitly direct us to choose the optimal $\alpha$ to obtain the fastest rate.

### 3.1 Problem Setup

We consider the standard stochastic unconstrained minimization problem as:

$$\min_{\boldsymbol{\theta} \in \mathbb{R}^d} f(\boldsymbol{\theta}) = \mathbb{E}_{(\mathbf{x},y) \sim \mathcal{D}}[F(\boldsymbol{\theta}; (\mathbf{x}, y))], \tag{1}$$

where the expectation is taken over the data distribution $(\mathbf{x}, y) \sim \mathcal{D}$. Given the Hessian matrix $\mathbf{H}_t$ at the decision point $\boldsymbol{\theta}_t$, we first define the following general Preconditioned Simultaneous Perturbation Stochastic Approximation (PSPSA) as:

**Definition 3.1** (Preconditioned Simultaneous Perturbation Stochastic Approximation (PSPSA)). *Given a model with parameters $\boldsymbol{\theta} \in \mathbb{R}^d$ and the loss function F, PSPSA estimates the zeroth-order*

stochastic gradient $\tilde{\nabla} F(\boldsymbol{\theta}_t)$ at $(\mathbf{x}_t, y_t)$ as

$$\tilde{\nabla} F(\boldsymbol{\theta}_t; (\mathbf{x}_t, y_t)) = \frac{F(\boldsymbol{\theta}_t + \mu \mathbf{H}_t^{-\alpha} \mathbf{u}; (\mathbf{x}_t, y_t)) - F(\boldsymbol{\theta}_t - \mu \mathbf{H}_t^{-\alpha} \mathbf{u}; (\mathbf{x}_t, y_t))}{2\mu} \cdot \mathbf{H}_t^{-\alpha} \mathbf{u}, \quad (2)$$

where $\mathbf{u} \in \mathbb{R}^d$ and $\mathbf{u} \sim \mathcal{N}(0, \mathbf{I}_d)$, $\mu$ is the perturbation scale, $\mathbf{H}_t$ is the Hessian matrix at $\boldsymbol{\theta}_t$, and $\alpha \in \left[-\frac{1}{2}, \frac{1}{2}\right]$ is the precondition order.

With the estimated zeroth-order stochastic gradient generated by PSPSA, the preconditioned zeroth-order optimization algorithm can be stated as follows:

**Definition 3.2** (Preconditioned Accelerated Zeroth-order Optimization, PaZO (Theoretical Form))**.** *PaZO is an optimizer with learning rate $\eta$ that updates parameters as*

$$\boldsymbol{\theta}_{t+1} = \boldsymbol{\theta}_t - \eta \tilde{\nabla} F(\boldsymbol{\theta}_t; (\mathbf{x}_t, y_t)), \quad (3)$$

where $\tilde{\nabla} F(\boldsymbol{\theta}_t; (\mathbf{x}_t, y_t))$ is the PSPSA gradient estimate at $\boldsymbol{\theta}_t$ with $\mathbf{H}_t$.

PSPSA and PaZO can be regarded as the general preconditioned extension of the existing zeroth-order perturbation approximation and algorithms. Intuitively, ignoring the higher-order infinitesimal term of $\mu$, we obtain the expectation of the PSPSA gradient estimate as

$$\mathbb{E}\left[\tilde{\nabla} F(\boldsymbol{\theta}_t; (\mathbf{x}_t, y_t))\right] = \mathbb{E}_{\mathbf{u}}\left[\frac{2\mu \nabla f^\top(\boldsymbol{\theta}_t) \cdot \mathbf{H}_t^{-\alpha} \mathbf{u}}{2\mu} \cdot \mathbf{H}_t^{-\alpha} \mathbf{u}\right] = \mathbf{H}_t^{-2\alpha} \nabla f(\boldsymbol{\theta}_t), \quad (4)$$

which indicates that the PSPSA gradient estimate is equivalent to a $\mathbf{H}_t^{-2\alpha}$ preconditioned gradient. When $\alpha = 0$, PSPSA degenerates to SPSA [48] and PaZO is reduced to ZO-SGD.

We introduce the assumption below to construct the relation between the outer product of the gradient and the Hessian for our analysis.

**Assumption 3.3** (Unbiased Estimate of Hessian)**.** *We assume that the expectation of the outer product of $F(\boldsymbol{\theta}^*, (\mathbf{x}, y))$ is the unbiased estimate of $\mathbf{H}^*$ as:*

$$\mathbb{E}\left[\nabla F(\boldsymbol{\theta}^*; (\mathbf{x}, y)) \nabla^\top F(\boldsymbol{\theta}^*; (\mathbf{x}, y))\right] = \mathbf{H}^*, \quad (5)$$

where $\boldsymbol{\theta}^*$ is a minimizer of the objective $f(\boldsymbol{\theta})$, and $\mathbf{H}^*$ is the Hessian defined at $\boldsymbol{\theta}^*$.

Assumption 3.3 is a common assumption when considering stochastic gradient descent [17, 24, 5, 25], especially for least squares regression [17, 24], whose Hessian is fixed and can be exactly calculated.

## 3.2 Case Study: Least Squares Regression

First, we try to provide an intuitive answer to the question *A*. We consider a representative case of $f$: least squares regression, whose optimization dynamic can be clear and meticulously calculated due to the fixed Hessian as:

$$F(\boldsymbol{\theta}; (\mathbf{x}, y)) = \frac{1}{2C} (y - \langle \boldsymbol{\theta}, \mathbf{x} \rangle)^2. \quad (6)$$

We have access to stochastic gradients zeroth-order obtained by PSPSA with sampling a new example $(\mathbf{x}_t, y_t) \sim \mathcal{D}$. These examples satisfy

$$y = \langle \boldsymbol{\theta}^*, \mathbf{x} \rangle + \epsilon,$$

where $\epsilon$ is a noise on the example pair with $\mathbb{E}[\epsilon] = 0$ and $\mathbb{E}[\epsilon^2] = \sigma^2$, and $\boldsymbol{\theta}^*$ is a minimizer of the objective. Note that the Hessian of the objective $\mathbf{H}^* \stackrel{\text{def}}{=} \nabla^2 f(\boldsymbol{\theta}) = \frac{1}{C}\mathbb{E}[\mathbf{x}\mathbf{x}^\top]$. The following estimate holds

$$\mathbb{E}\left[\nabla F(\boldsymbol{\theta}^*; (\mathbf{x}, y)) \nabla^\top F(\boldsymbol{\theta}^*; (\mathbf{x}, y))\right] = \frac{1}{C^2}\mathbb{E}[\epsilon^2 \mathbf{x}\mathbf{x}^\top] = \frac{\sigma^2}{C}\mathbf{H}^*. \quad (7)$$

By setting $C = \sigma^2$, we exactly obtain the result in Assumption 3.3. The analytical tractability of (6) offers deeper theoretical insights. Specifically, previous studies [17] demonstrate that for first-order algorithms the *optimal* rate achieves $\tilde{\mathcal{O}}(d/T)$ and construct the lower bound, where $d$ is the dimension of problems and $T$ is the iteration steps. Moreover, the studies show that *only SGD* can match the near-optimal rate with only the difference of logarithmic terms. In other words, for least

squares regression and first-order stochastic algorithms, only SGD is enough with any precondition making no effect of acceleration. When turning to zeroth-order optimization, intuitively, we think the *ideal convergence rate* achieves $\tilde{\mathcal{O}}(d^2/T)$ since in zeroth-order optimization we can only access one-dimension information per step. Varieties of theoretical studies of zeroth-order algorithms [2, 41] also show $d$ times slower convergence rate than first-order ones. However, the results stated in Theorem 3.5 indicate that only ZO-SGD is not enough.

**Assumption 3.4** (Fourth Moment Conditions). *Suppose* $\mathbf{B}$ *is a positive semi-definite matrix, and consider data vector* $\mathbf{x}$. *It satisfies* $\mathbb{E}_{\mathbf{x}}\left[\mathbf{x}\mathbf{x}^\top \mathbf{B}\mathbf{x}\mathbf{x}^\top\right] \preceq \mathcal{O}\left(\mathrm{tr}\left(\mathbf{H}^*\mathbf{B}\right)\mathbf{H}^*\right)$.

**Theorem 3.5** (Convergence Rate of PaZO on Least Squares). *Suppose we are given access to the PSPSA, running PaZO for least squares regression* (6) *satisfying Assumption 3.4 with a learning rate* $\eta$ *satisfying* $\frac{1}{\lambda_{min}((\mathbf{H}^*)^{1-2\alpha})T} \lesssim \eta \lesssim \min\left\{\frac{1}{\lambda_{max}((\mathbf{H}^*)^{1-2\alpha})}, \frac{\lambda_{\min}(\mathbf{H}^*)}{\lambda_{\max}(\mathbf{H}^*)\mathrm{tr}((\mathbf{H}^*)^{-2\alpha})\mathrm{tr}((\mathbf{H}^*)^{1-2\alpha})}\right\}$ *for* $2T$ *steps with* $T \gtrsim \frac{\lambda_{\max}(\mathbf{H}^*)\mathrm{tr}((\mathbf{H}^*)^{-2\alpha})\mathrm{tr}((\mathbf{H}^*)^{1-2\alpha})}{\lambda_{min}((\mathbf{H}^*)^{1-2\alpha})\lambda_{\min}(\mathbf{H}^*)}$ *allows PaZO to achieve the following convergence rate:*

$$\mathbb{E}\left[f\left(\frac{1}{T}\sum_{t=T}^{2T-1}\boldsymbol{\theta}_t\right)\right] - f(\boldsymbol{\theta}^*) \leq \frac{\left(1-\eta\lambda_{\min}\left((\mathbf{H}^*)^{1-2\alpha}\right)\right)^T}{\eta T}\|\boldsymbol{\theta}_0 - \boldsymbol{\theta}^*\|^2_{(\mathbf{H}^*)^{2\alpha}} + \frac{D_\alpha}{T}, \tag{8}$$

*where* $D_\alpha = \mathrm{tr}\left((\mathbf{H}^*)^{2\alpha-1}\right) \cdot \mathrm{tr}\left((\mathbf{H}^*)^{1-2\alpha}\right)$ *and* $\alpha$ *is the precondition order defined in PSPSA.*

Theorem 3.5 provides an affirmative answer to question *A*. Since the first term decays exponentially with $T$, the rate depends on the second term $D_\alpha/T$, which is a trade-off between $\mathrm{tr}\left((\mathbf{H}^*)^{2\alpha-1}\right)$ and $\mathrm{tr}\left((\mathbf{H}^*)^{1-2\alpha}\right)$. Through Cauchy-Schwarz inequality, we have $D_\alpha \geq d^2$, where the equality holds if and only if $\alpha = 1/2$. In other words, only ZO-SGD is not enough to match the ideal rate $\tilde{\mathcal{O}}(d^2/T)$. Therefore, Theorem 3.5 demonstrates that different from first-order algorithms, *we need preconditions in zeroth-order optimization.*

Moreover, we consider the convergence analysis with approximate Hessian $\tilde{\mathbf{H}}_t$ in PSPSA. When the gap between $\tilde{\mathbf{H}}_t$ and $\mathbf{H}_t$ can be well controlled, we can also achieve the fastest rate when $\alpha = 1/2$. The detailed assumption and analysis are shown in Appendix B.

### 3.3 General Functions

Second, we propose the theoretical analysis for general smooth functions. Based on the affirmative answer to question *A* provided by Theorem 3.5, we conducted a more in-depth analysis of general functions, thereby establishing a more reasonable solution to question *B*. We may also obtain the results under approximate Hessian. For convenience, we assume its exact.

**Assumption 3.6** (Gradient Uniform Continuity). *For any given sample pair* $(\mathbf{x}, y) \sim \mathcal{D}$, *the stochastic gradient of the objective* $\nabla F(\boldsymbol{\theta}; (\mathbf{x}, y))$ *satisfies uniform continuity.*

**Assumption 3.7** (General Hessian Smooth). *For any given* $\boldsymbol{\theta}_1$, $\boldsymbol{\theta}_2$ *and* $\alpha' \in [-1, 1]$, *the Hessian of the objective* $\mathbf{H}(\boldsymbol{\theta}_1)$ *and* $\mathbf{H}(\boldsymbol{\theta}_2)$ *are invertible and satisfy*

$$\left\|\mathbf{H}^{\alpha'}(\boldsymbol{\theta}_1) - \mathbf{H}^{\alpha'}(\boldsymbol{\theta}_2)\right\| \leq \rho|\alpha'|\|\boldsymbol{\theta}_1 - \boldsymbol{\theta}_2\|^{|\alpha'|}.$$

Assumption 3.7 is the generalization form of Lipschitz continuity of Hessian. When $\alpha = 1$, it reduces to Hessian Lipschitz continuity. We use it to limit the gap between $\mathbf{H}_t^{-2\alpha}$ in PSPSA and $(\mathbf{H}^*)^{-2\alpha}$. When the objective is strongly convex, the Hessian is naturally invertible, while for others we assume its invertible property. We propose the convergence rate of general functions in Theorem 3.8.

**Theorem 3.8** (Convergence Rate of PaZO on General Functions). *Suppose we are given access to the PSPSA, running PaZO for general functions* (1) *satisfying Assumption 3.3, 3.6 and 3.7 with a learning rate* $\eta$ *satisfying* $\frac{1}{\lambda_{min}((\mathbf{H}^*)^{1-2\alpha})T} \lesssim \eta \leq \frac{1}{\lambda_{max}((\mathbf{H}^*)^{1-2\alpha})}$ *for* $2T$ *steps with* $T \gtrsim \frac{\lambda_{max}((\mathbf{H}^*)^{1-2\alpha})}{\lambda_{min}((\mathbf{H}^*)^{1-2\alpha})}$ *where* $\mathbf{H}^*$ *is full-rank and* $\mathbb{E}\|\boldsymbol{\theta}_t - \boldsymbol{\theta}^*\|^p \leq \epsilon_0^p$ *for any* $t \in [T, 2T-1]$ *and* $p \in [0, 3]$ *allows PaZO to achieve the following asymptotic convergence rate:*

$$\mathbb{E}\left[f\left(\frac{1}{T}\sum_{t=T}^{2T-1}\boldsymbol{\theta}_t\right)\right] - f(\boldsymbol{\theta}^*) \lesssim \frac{(1-\eta\lambda_{\min}((\mathbf{H}^*)^{1-2\alpha}))^{2T}}{\eta^2 T^2}\|\boldsymbol{\theta}_0 - \boldsymbol{\theta}^*\|^2_{(\mathbf{H}^*)^{4\alpha-1}}$$

$$+ \frac{\mathrm{tr}\left((\mathbf{H}^*)^{2\alpha-1}\right) \cdot \mathrm{tr}\left((\mathbf{H}^*)^{1-2\alpha}\right)}{T} + \bar{\mathbf{Err}}, \tag{9}$$

**Algorithm 1** PaZO (Practical Form)

---

**Require:** parameters $\Theta = \{\boldsymbol{\theta}_i \in \mathbb{R}^{d_i}\}$, loss $\mathcal{L} : \mathbb{R}^d \to \mathbb{R}$, running steps $T$, perturbation scale $\mu$, learning rate schedule $\eta_t$, smooth scale $\beta_1, \beta_2$, initialized diagonal Hessian $\boldsymbol{\Sigma}_0 = \mathbf{I}$, random seed $s$, a random number generator, Hessian reset frequency $T_0$

  **for** $t = 1, ..., T$ **do**

    **Step 1**: Perturb Parameters through Diagonal Hessian

      Sample batch $\mathcal{B} \subset \mathcal{D}$ and random seed $s$

      $\ell \leftarrow \mathcal{L}(\boldsymbol{\theta}; \mathcal{B})$

      $\boldsymbol{\theta} \leftarrow \text{PerturbParameters}(\boldsymbol{\theta}, \mu, \boldsymbol{\Sigma}_{t-1}^{-1/2}, s)$

      $\ell_+ \leftarrow \mathcal{L}(\boldsymbol{\theta}; \mathcal{B})$

      $\boldsymbol{\theta} \leftarrow \text{PerturbParameters}(\boldsymbol{\theta}, -2\mu, \boldsymbol{\Sigma}_{t-1}^{-1/2}, s)$

      $\ell_- \leftarrow \mathcal{L}(\boldsymbol{\theta}; \mathcal{B})$

      $\boldsymbol{\theta} \leftarrow \text{PerturbParameters}(\boldsymbol{\theta}, \mu, \boldsymbol{\Sigma}_{t-1}^{-1/2}, s)$         {Reset parameters before descent}

    **Step 2**: Estimate Diagonal Hessian

      $\tilde{\mathbf{g}} \leftarrow (\ell_+ - \ell_-) * \boldsymbol{\Sigma}_t^{1/2}\mathbf{u}/2\mu$       {Estimate unbiased zeroth-order gradient}

      $\tilde{\boldsymbol{\Sigma}} = \left((1 - \beta_1)\boldsymbol{\Sigma}_{t-1}^2 + \beta_1 \cdot \text{diag}^2(\tilde{\mathbf{g}} \circ \tilde{\mathbf{g}})\right)^{1/2}$     {Adding information from ZO gradient}

      $\hat{\boldsymbol{\Sigma}}_t = \frac{1}{2\mu^2}(\ell_+ + \ell_- - 2\ell)\left(\tilde{\boldsymbol{\Sigma}}\left(\text{diag}(\mathbf{u} \circ \mathbf{u}) - \mathbf{I}\right)\right)$   {Estimate unbiased diagonal Hessian}

    **Step 3**: Take Moving Average and Reset Diagonal Hessian

      $\boldsymbol{\Sigma}_t \leftarrow (1 - \beta_2)\boldsymbol{\Sigma}_{t-1} + \beta_2|\hat{\boldsymbol{\Sigma}}_t|$       {Take moving average of diagonal Hessian}

    **if** $t \% T_0 = 0$ **then**

      $\boldsymbol{\Sigma}_t \leftarrow \mathbf{I}$              {Frequently reset diagonal Hessian}

    **end if**

    **Step 4**: Update the Parameters

      Reset random number generator with seed $s$          {For sampling $\mathbf{u}_i$}

      $\text{preconditioned\_grad} \leftarrow (\ell_+ - \ell_-) * \boldsymbol{\Sigma}_t^{-1/2}/2\mu$     {Using $\boldsymbol{\Sigma}^{-1}$ as preconditioner}

    **for** $\boldsymbol{\theta}_i \in \Theta$ **do**

      Sample $\mathbf{u}_i \sim \mathcal{N}(0, \mathbf{I}_{d_i})$

      $\boldsymbol{\theta}_i \leftarrow \boldsymbol{\theta}_i - \eta_t * \text{preconditioned\_grad} * \mathbf{u}_i$

    **end for**

  **end for**

---

*where* $\bar{\mathbf{Err}} = \mathcal{O}\left(\eta\rho\epsilon_0^3 + \eta\rho\epsilon_0^{2|\alpha|+1} + \eta^2\epsilon_0 + \eta^2\rho\epsilon_0^{2|\alpha|}\right)$ *represents the higher-order infinitesimal term, and* $\alpha$ *is the precondition order defined in PSPSA.*

We observe that the dominant term of the rate in Theorem 3.8 aligns with the rate in Theorem 3.5, which further demonstrates the generalized validity of our analysis on the role of preconditioning in zeroth-order optimization: for general problems, selecting $\alpha = 0$ alone induces a slower convergence rate than the optimal choice $\alpha = 1/2$. Moreover, the $\bar{\mathbf{Err}}$ is defined as the higher-order infinitesimal term $\mathcal{O}\left(\eta\rho\epsilon_0^3 + \eta\rho\epsilon_0^{2|\alpha|+1} + \eta^2\epsilon_0 + \eta^2\rho\epsilon_0^{2|\alpha|}\right)$ that negligibly impacts the convergence rate of the dominant term. When $T$ grows, we can choose the smaller $\eta$ to obtain the controlled $\bar{\mathbf{Err}}$. Thus, Theorem 3.5 can be regarded as a special case of Theorem 3.8, and we propose the proof in detail in Appendix A. By selecting $\alpha = 1/2$, PaZO achieves the fastest convergence rate $\tilde{\mathcal{O}}\left(d^2/T\right)$ compared with MeZO for general functions, providing a reasonable answer to question $B$.

## 4 Algorithm for Fine-Tuning LLMs in Practice

In this section, we introduce PaZO (Practical Form) in Algorithm 1 for fine-tuning LLMs in practice. We provide an answer to the question below.

> *C*. How to effectively estimate Hessian information through zeroth-order perturbations in practice and improve fine-tuned model performance on downstream tasks?

---

**Algorithm 2** PerturbParameters

---

**Require:** model parameters $\Theta = \{\boldsymbol{\theta}_i \in \mathbb{R}^{d_i}\}$, perturbation scale $\mu$, diagonal Hessian $\boldsymbol{\Sigma}_t^{-1/2}$, random seed $s$, a random number generator

    Reset random number generator with seed $s$                           {For sampling $\mathbf{u}_i$}

    **for** $\boldsymbol{\theta}_i \in \Theta$ **do**

        Sample $\mathbf{u}_i \sim \mathcal{N}(0, \mathbf{I}_{d_i})$

        $\boldsymbol{\theta}_i \leftarrow \boldsymbol{\theta}_i + \mu \boldsymbol{\Sigma}_t^{-1/2} \mathbf{u}_i$                           {Modify parameters in place}

    **end for**

---

Specifically, we apply the theoretically optimal order of preconditioner $\mathbf{H}^{-1/2}$ in the PSPSA process. Then we estimate diagonal Hessian with incorporating the current zeroth-order gradient information and moving average techniques through the same PSPSA process for estimating the preconditioned zeroth-order gradient. Our algorithm can be divided into four steps.

**Step I. Perturb Parameters through Diagonal Hessian.** First, we apply PSPSA to our practical algorithm to obtain the preconditioned zeroth-order gradient. Inspired by our theoretical results, we use $\boldsymbol{\Sigma}^{-1/2}$ as the preconditioner in the PSPSA process, where $\boldsymbol{\Sigma}$ is the estimated diagonal Hessian. Through twice forward passes of PSPSA we obtain

$$\ell_+ = F(\boldsymbol{\theta} + \mu \boldsymbol{\Sigma}^{-1/2}\mathbf{u}; (\mathbf{x}, y)), \quad \ell_- = F(\boldsymbol{\theta} - \mu \boldsymbol{\Sigma}^{-1/2}\mathbf{u}; (\mathbf{x}, y)).$$

Moreover, we run another additional forward pass before adding perturbation to obtain $\ell = F(\boldsymbol{\theta}; (\mathbf{x}, y))$ for estimating $\boldsymbol{\Sigma}$ in the following steps.

**Step II. Estimate Diagonal Hessian.** We try to estimate the diagonal Hessian through $\ell_+, \ell_-$ and $\ell$, with $\mathcal{O}(d)$ memory cost against $\mathcal{O}(d^2)$ for the full Hessian. Specifically, in the theoretical analysis of the Hessian-aware zeroth-order optimization [55], they demonstrate that

$$\mathbb{E}_{\mathbf{u} \sim \mathcal{N}(0, \mathbf{I}_d)} \left[ \frac{1}{2} \mathbf{u}^\top \mathbf{A}^{\frac{1}{2}} \mathbf{H} \mathbf{A}^{\frac{1}{2}} \mathbf{u} \cdot \left( \mathbf{A}^{-\frac{1}{2}} \mathbf{u} \mathbf{u}^\top \mathbf{A}^{-\frac{1}{2}} - \mathbf{A}^{-1} \right) \right] = \mathbf{H}, \tag{10}$$

where $\mathbf{H}$ is the Hessian matrix, and $\mathbf{A}$ is *any* given positive definite matrix. Thus, letting $\boldsymbol{\Sigma}$ be a positive definite diagonal matrix and setting $\mathbf{A} = \boldsymbol{\Sigma}^{-1}$, we obtain the diagonal version of (10) as

$$\mathbb{E} \left[ \frac{1}{2} \underbrace{\mathbf{u}^\top \boldsymbol{\Sigma}^{-\frac{1}{2}} \mathbf{H} \boldsymbol{\Sigma}^{-\frac{1}{2}} \mathbf{u}}_{\mathcal{I}} \cdot \boldsymbol{\Sigma} \left( \operatorname{diag}(\mathbf{u} \circ \mathbf{u}) - \mathbf{I} \right) \right] = \operatorname{diag}(\mathbf{H}). \tag{11}$$

We use $\ell_+, \ell_-$ and $\ell$ to estimate $\mathcal{I}$. Through Talyor expansion, we have

$$\begin{aligned}
\ell_+ &= F(\boldsymbol{\theta}; (\mathbf{x}, y)) + \mu \left\langle \nabla F(\boldsymbol{\theta}; (\mathbf{x}, y)), \boldsymbol{\Sigma}^{-\frac{1}{2}} \mathbf{u} \right\rangle + \frac{\mu^2}{2} \mathcal{I} + \mathcal{O}(\mu^3), \\
\ell_- &= F(\boldsymbol{\theta}; (\mathbf{x}, y)) - \mu \left\langle \nabla F(\boldsymbol{\theta}; (\mathbf{x}, y)), \boldsymbol{\Sigma}^{-\frac{1}{2}} \mathbf{u} \right\rangle + \frac{\mu^2}{2} \mathcal{I} + \mathcal{O}(\mu^3).
\end{aligned} \tag{12}$$

Thus, we can obtain $\mathcal{I}$ by the combination of $\ell_+, \ell_-$ and $\ell$ as

$$\frac{\ell_+ + \ell_- - 2\ell}{\mu^2} = \mathcal{I} + \mathcal{O}(\mu). \tag{13}$$

Moreover, incorporating the current gradient information into the preconditioner is demonstrated to be effective in first-order optimizers [28, 35]. We additionally estimate

$$\tilde{\mathbf{g}} = (\ell_+ - \ell_-) * \frac{\boldsymbol{\Sigma}_t^{1/2} \mathbf{u}}{2\mu} = \mathbf{u}\mathbf{u}^\top \nabla F(\boldsymbol{\theta}; (\mathbf{x}, y)) + \mathcal{O}(\mu)$$

as an unbiased zeroth-order gradient and incorporate $\operatorname{diag}(\tilde{\mathbf{g}} \circ \tilde{\mathbf{g}})$ as a correction item to integrate local first-order estimated information into $\boldsymbol{\Sigma}_t$ through a moving average mechanism as

$$\tilde{\boldsymbol{\Sigma}} = \left( (1 - \beta_1) \boldsymbol{\Sigma}_{t-1}^2 + \beta_1 \cdot \operatorname{diag}^2(\tilde{\mathbf{g}} \circ \tilde{\mathbf{g}}) \right)^{1/2}.$$

Then we use (11) to update the diagonal Hessian as

$$\hat{\boldsymbol{\Sigma}}_t = \frac{1}{2\mu^2} (\ell_+ + \ell_- - 2\ell) \left( \tilde{\boldsymbol{\Sigma}} \left( \operatorname{diag}(\mathbf{u} \circ \mathbf{u}) - \mathbf{I} \right) \right). \tag{14}$$

**Step III. Take Moving Average and Reset Diagonal Hessian.** In practice, we empirically discover the instability of the estimated diagonal Hessian. To solve this problem, we take the moving average of the historical estimate and the current one to maintain the smoothness and stability of $\boldsymbol{\Sigma}_t$ as

$$\boldsymbol{\Sigma}_t = (1 - \beta_2)\boldsymbol{\Sigma}_{t-1} + \beta_2|\hat{\boldsymbol{\Sigma}}_t|, \tag{15}$$

where $|\hat{\boldsymbol{\Sigma}}_t|$ means taking the absolute values of $\hat{\boldsymbol{\Sigma}}_t$ to maintain positive definite. Moreover, when the iteration step exceeds a threshold, excessive accumulated historical information may no longer positively contribute. Therefore, we reset the $\boldsymbol{\Sigma}$ frequently after some steps.

**Step IV. Update the Parameters.** Finally, we layer-wisely compute the preconditioned gradient by PSPSA, where the gradient estimate is equivalent to a $\boldsymbol{\Sigma}^{-1}$ preconditioned zeroth-order gradient.

**Remark 4.1.** *For $\beta_1$, we first clarify that the correction term $\tilde{g} \circ \tilde{g}$ is introduced to mitigate training instability caused by outliers from the stochastic zeroth-order oracle. Specifically, since the preconditioner order is set to $\alpha = 1/2$, excessively small values in the diagonal Hessian estimate can lead to numerical instability (e.g., NaN values) during training. This correction term promotes numerical alignment between gradient magnitudes and adaptive curvature scaling, similar to the mechanism in Adam. However, applying such a correction introduces bias into Eq. (14). By choosing a small $\beta_1$ to constrain this bias, we observe that this nearly negligible term enhances estimation robustness against outliers and ensures training stability. As shown in Appendix C.7, setting $\beta_1$ too large (e.g., $1$ or $10^{-2}$) causes optimization divergence and results in NaNs, as the correction term introduces non-negligible bias. Conversely, when $\beta_1$ is too small (e.g., $0$ or close to $0$), the correction fails to take effect, potentially leading to numerical instability and NaNs. Only within an appropriate range (around $10^{-8}$ to $10^{-10}$) does the algorithm achieve stable and reasonable performance.*

*For $\beta_2$, it is designed to reduce the high variance in Hessian estimates from the zeroth-order oracle. For the Hessian estimate $\boldsymbol{\Sigma}_t$ at step $t$, a small $\beta_2$ ensures that the cumulative variance from the sequence $\{\hat{\boldsymbol{\Sigma}}_k\}_{k=0}^{t-1}$ remains controlled, thereby enabling lower-variance and more stable updates during training. A similar hyperparameter configuration is adopted in other zeroth-order fine-tuning optimizers, such as in [59]. Our experimental results further confirm that, under identical parameter settings, PaZO improves the performance of fine-tuned model across tasks.*

## 5 Experiment

We conduct experiments on both masked LMs (RoBERTa-large, 350M [34]) and large-scale generative LMs (OPT-1.3B [57]) with zero-shot learning, linear probing (LP [22]), in-context learning (ICL [8]), full-parameter tuning and PEFT including LoRA [23] and prefix-tuning [31] (see Appendix C.3 for details). We compare PaZO with other representative zeroth-order optimizers including MeZO and HiZOO (see Appendix C.4 for details). We first show that PaZO achieves significant improvement over zero-shot, ICL, and LP. Compared with first-order optimizers (FT), PaZO drastically reduces the memory cost while maintaining comparable performance. Moreover, PaZO realizes better performance compared with MeZO and HiZOO. Detailed settings are presented in Appendix C.2.

### 5.1 Masked Language Models

We conduct experiments for RoBERTa-large (350M) on sentiment classification, natural language inference, and topic classification tasks. We sample $k$ examples per class for $k = 16$, running zeroth-shot learning, LP, fine-tuning, MeZO and PaZO. We summarize the results in Table 1. First, we show that: (1) PaZO works significantly better than zero-shot and LP; (2) PaZO achieves comparable performance to FT. Moreover, we show the better performance of PaZO compared with MeZO.

**PaZO achieves better performance compared with MeZO.** As shown in Table 1, PaZO achieves improved performance on average across all the datasets, tasks and PEFT (we choose the best results from LoRA and prefix-tuning). For sentiment tasks, the improvement of PaZO is universal, while for NLI and topic tasks the improvement is evident on MNLI and TREC with 9.3% and 5.4%.

### 5.2 Generative Language Models

We extend the experiments to the OPT 1.3B model [57] on classification and multiple-choice tasks on different datasets (see Appendix C.1 for details). We randomly sample 1000, 500, and 1000 examples

Table 1: Experiments on RoBERTa-large (350M parameters, k=16). We use zero-shot learning, linear probing (LP), full-parameter fine-tuning with Adam, MeZO and PaZO on six downstream tasks. We also test PEFT methods including LoRA and prefix tuning with Adam, MeZO and PaZO respectively. All reported numbers are averaged accuracy (standard deviation) across 5 runs.

| Task Type | SST-2 | SST-5 | SNLI | MNLI | RTE | TREC | Average |
|---|---|---|---|---|---|---|---|
| | —— sentiment —— | | —— natural language inference —— | | | — topic — | |
| Zero-shot | 79.0 | 35.5 | 50.2 | 48.8 | 51.4 | 32.0 | 49.5 |
| LP | 76.0 ($\pm$2.8) | 40.3 ($\pm$1.9) | 66.0 ($\pm$2.7) | 56.5 ($\pm$2.5) | 59.4 ($\pm$5.3) | 51.3 ($\pm$5.5) | 58.3 |
| FT | 90.9 ($\pm$1.7) | 44.8 ($\pm$1.6) | 67.5($\pm$2.4) | 58.2 ($\pm$3.1) | 66.4 ($\pm$7.2) | 85.0 ($\pm$2.5) | 68.8 |
| FT (PEFT) | 91.9 ($\pm$1.0) | 43.2 ($\pm$1.1) | 65.5 ($\pm$1.8) | 57.1 ($\pm$1.3) | 65.5 ($\pm$1.9) | 79.8 ($\pm$1.5) | 67.2 |
| MeZO | 90.5 ($\pm$1.2) | 42.3 ($\pm$2.1) | 66.7 ($\pm$3.3) | 51.6 ($\pm$3.0) | **64.0** ($\pm$3.3) | 70.2 ($\pm$1.4) | 64.2 |
| MeZO (PEFT) | 91.3 ($\pm$1.0) | 42.4 ($\pm$2.5) | 62.7 ($\pm$2.8) | 55.6 ($\pm$2.0) | 60.5 ($\pm$3.6) | 73.4 ($\pm$3.6) | 64.3 |
| PaZO | **91.4** ($\pm$0.8) | **44.6** ($\pm$1.7) | **66.7** ($\pm$2.6) | **56.4** ($\pm$2.1) | 63.2 ($\pm$5.2) | 70.8 ($\pm$2.0) | **65.6** |
| PaZO (PEFT) | 91.3 ($\pm$0.3) | 42.9 ($\pm$0.5) | 62.4 ($\pm$1.6) | 55.8 ($\pm$1.7) | 61.5 ($\pm$2.2) | **77.4** ($\pm$3.5) | 65.2 |

Table 2: Performance comparison with MeZO and HiZOO. We fine-tune OPT-1.3B on different downstream datasets and evaluate the performance, applying LoRA and prefix-tuning.

| Task Type | SST-2 | BoolQ | CB | ReCoRD | RTE | WIC | WSC | COPA | MultiRC | Average |
|---|---|---|---|---|---|---|---|---|---|---|
| | | | ———————— classification ———————— | | | | | – multiple choice – | | |
| MeZO | 88.5 | 63.4 | 67.8 | 72.3 | 66.1 | 60.6 | 57.6 | 76.0 | 56.3 | 67.6 |
| MeZO (LoRA) | 88.5 | 63.0 | 60.7 | 70.6 | 59.9 | 58.2 | 54.8 | 77.0 | 58.9 | 65.7 |
| MeZO (prefix) | 91.3 | **64.1** | 67.9 | 71.0 | 62.5 | 54.2 | 51.2 | 75.0 | 57.2 | 66.0 |
| HiZOO | 88.5 | 61.4 | 67.9 | 71.9 | 64.3 | 62.2 | **62.5** | 73.0 | **59.3** | 67.9 |
| HiZOO (LoRA) | 88.5 | 63.1 | 69.6 | **72.5** | 64.6 | 60.6 | 54.8 | 76.0 | 58.9 | 67.6 |
| HiZOO (prefix) | 91.3 | 63.6 | 67.9 | 70.9 | 63.2 | 53.8 | 57.7 | 75.0 | 54.5 | 66.4 |
| PaZO | 89.0 | 63.4 | 69.6 | 72.1 | **66.4** | **63.2** | 61.5 | 75.0 | 57.6 | **68.6** |
| PaZO (LoRA) | 88.5 | 63.4 | **73.2** | 72.1 | 62.8 | 58.2 | 54.8 | **77.0** | 58.9 | 67.7 |
| PaZO (prefix) | **91.3** | 63.4 | 67.9 | 71.0 | 62.3 | 53.8 | 57.7 | 75.0 | 57.2 | 66.6 |

for training, validation, and test sets, respectively, for each dataset. We run MeZO, HiZOO and PaZO for 20K steps, and compare the performance with different zeroth-order optimizers in Table 2.

**PaZO achieves SOTA performance compared with other zeroth-order optimizers.** As shown in Table 2, PaZO achieves SOTA performance compared to other zeroth-order optimizer baselines including MeZO and HiZOO. Specifically, for average performance, PaZO achieves all-round improvement beyond MeZO and HiZOO, no matter the full-parameter version, the LoRA version or the prefix-tuning version. For single-task performance, PaZO and its peft version show advantages in the vast majority of tasks and have little gaps in other tasks.

## 5.3 Memory Usage and Wall-clock Time Analysis

**Memory Usage.** As shown in Figure 1, PaZO has more memory overhead compared to MeZO because of the storage of the diagonal Hessian, and maintains the memory overhead compared to HiZOO. However, PaZO also exhibits extreme saving of memory compared to first-order optimizers, specifically, up to $6\times$ compared to standard FT and $3\times$ compared to FT (prefix-tuning).

**Wall-clock Time.** As shown in Figure 2, PaZO spends $1.5\times$ time per step compared with MeZO, and the same time per step compared with HiZOO, since preconditioned optimizers need an additional forward pass for estimating diagonal Hessian. In Figure 2, Model1 means we use LoRA and Model2 means we use prefix-tuning. Considering the accelerated convergence rate of PaZO with fewer steps to obtain the same loss, PaZO achieves better performance with an acceptable extra time cost.

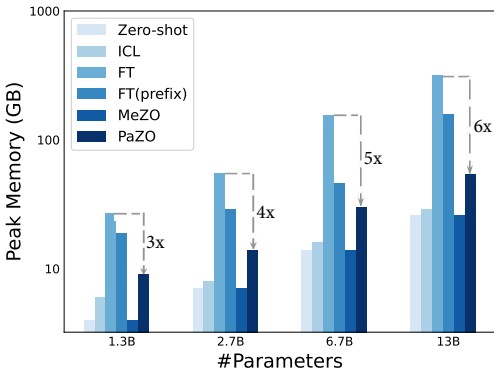

Figure 1: GPU peak memory overhead with different OPT models and tuning methods on MultiRC (400 tokens per example on average). See Appendix C.5 for details.

|            | MeZO    | HiZOO   | PaZO    |
|------------|---------|---------|---------|
| RoBERTa-L  | 0.2091s | 0.3020s | 0.3046s |
| RoBERTa-L1 | 0.1338s | 0.1993s | 0.2013s |
| RoBERTa-L2 | 0.1254s | 0.1869s | 0.1892s |
| OPT-1.3B   | 0.2564s | 0.3812s | 0.3837s |
| OPT-1.3B1  | 0.1664s | 0.2798s | 0.2857s |
| OPT-1.3B2  | 0.1572s | 0.2374s | 0.2419s |

Figure 2: Wallclock time per step among MeZO, HiZOO and PaZO. The increase in wallclock time per step for PaZO compared to MeZO is less than 1.5 times across different model sizes. All results are measured on the same dataset (SST-2) and GPUs (24GB 3090), with each result averaged over 100 steps.

# 6    Conclusion

In this work, we propose PaZO, a preconditioned accelerated zeroth-order optimization method for fine-tuning LLMs. We theoretically analyze the necessity of preconditions in ZO, and demonstrate the optimal order of preconditioners to achieve the fastest convergence rate. We propose the practical form of PaZO and extensive experiments on different models and tasks show the effectiveness.

# 7    Acknowledgements

Z. Lin was supported by National Key R&D Program of China (2022ZD0160300), the NSF China (No. 62276004) and the State Key Laboratory of General Artificial Intelligence. C. Fang was supported by National Key R&D Program of China (2022ZD0160300) and the NSF China (No. 92470117 and No. 62376008).

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

# A  Proof of Theorem 3.5 and Theorem 3.8

We prove Theorem 3.5 and Theorem 3.8 by three steps below. First, we rewrite the update form to obtain the coupled recursive formula of $(\boldsymbol{\theta}_{t_1} - \boldsymbol{\theta}^*)(\boldsymbol{\theta}_{t_2} - \boldsymbol{\theta}^*)^\top$ ignoring higher-order infinitesimal terms. Second, we obtain the estimation of the sum of $(\boldsymbol{\theta}_{t_1} - \boldsymbol{\theta}^*)(\boldsymbol{\theta}_{t_2} - \boldsymbol{\theta}^*)^\top$ with $t_1$ and $t_2$ from $T$ to $2T - 1$. Finally, by Taylor expansion of $f\left(\frac{1}{T}\sum_{t=T}^{2T-1}\boldsymbol{\theta}_t\right)$ on $\boldsymbol{\theta}^*$, we obtain the results in Theorem 3.5 and Theorem 3.8.

Specifically, Theorem 3.5 can be regarded as a special case of Theorem 3.8. Thus we employ a generalized proof framework to establish the proofs of the two Theorems above. The main body of our proof addresses general function (as stated in Theorem 3.8), while the least squares (Theorem 3.5) is distinctly labeled as "**Least Squares**" for clarity.

*Proof.* **Step I.** We first rewrite the update rule from

$$\boldsymbol{\theta}_{t+1} = \boldsymbol{\theta}_t - \eta\tilde{\nabla}F(\boldsymbol{\theta}_t; (\mathbf{x}_t, y_t)) \tag{16}$$

to separate the decay term and higher-order term as below :

$$
\begin{aligned}
\boldsymbol{\theta}_{t+1} - \boldsymbol{\theta}^* &= \boldsymbol{\theta}_t - \boldsymbol{\theta}^* - \eta\left(\tilde{\nabla}F(\boldsymbol{\theta}_t; (\mathbf{x}_t, y_t)) - (\mathbf{H}^*)^{-2\alpha}\nabla f(\boldsymbol{\theta}^*)\right) \\
&= (\mathbf{I} - \eta(\mathbf{H}^*)^{1-2\alpha})(\boldsymbol{\theta}_t - \boldsymbol{\theta}^*) \\
&\quad + \eta\left((\mathbf{H}^*)^{1-2\alpha}(\boldsymbol{\theta}_t - \boldsymbol{\theta}^*) - \mathbb{E}\left[\tilde{\nabla}F(\boldsymbol{\theta}_t; (\mathbf{x}_t, y_t))\right] + (\mathbf{H}^*)^{-2\alpha}\nabla f(\boldsymbol{\theta}^*)\right) \\
&\quad + \eta\left(\mathbb{E}\left[\tilde{\nabla}F(\boldsymbol{\theta}_t; (\mathbf{x}_t, y_t))\right] - \tilde{\nabla}F(\boldsymbol{\theta}_t; (\mathbf{x}_t, y_t))\right).
\end{aligned}
\tag{17}
$$

Denoting $\mathbf{Q}^* = \mathbf{I} - \eta(\mathbf{H}^*)^{1-2\alpha}$, with $\eta \leq \frac{1}{\lambda_{max}((\mathbf{H}^*)^{1-2\alpha})}$ we have $\mathbf{Q}^* \succeq \mathbf{0}$. For any $T \leq t_2 < t_1 \leq 2T$, by recursive formula (17), we have

$$\boldsymbol{\theta}_{t_1} - \boldsymbol{\theta}^* = \underbrace{(\mathbf{Q}^*)^{t_1-t_2}(\boldsymbol{\theta}_{t_2} - \boldsymbol{\theta}^*)}_{\mathcal{A}} + \mathcal{B} + \mathcal{C} \tag{18}$$

where

$$\mathcal{B} = \eta\sum_{j=1}^{t_1-t_2}(\mathbf{Q}^*)^{j-1}\left((\mathbf{H}^*)^{1-2\alpha}(\boldsymbol{\theta}_{t_1-j} - \boldsymbol{\theta}^*) - \mathbb{E}\left[\tilde{\nabla}F(\boldsymbol{\theta}_{t_1-j}; (\mathbf{x}_{t_1-j}, y_{t_1-j}))\right] + (\mathbf{H}^*)^{-2\alpha}\nabla f(\boldsymbol{\theta}^*)\right),$$

and

$$\mathcal{C} = \eta\sum_{j=1}^{t_1-t_2}(\mathbf{Q}^*)^{j-1}\left(\mathbb{E}\left[\tilde{\nabla}F(\boldsymbol{\theta}_{t_1-j}; (\mathbf{x}_{t_1-j}, y_{t_1-j}))\right] - \tilde{\nabla}F(\boldsymbol{\theta}_{t_1-j}; (\mathbf{x}_{t_1-j}, y_{t_1-j}))\right). \tag{19}$$

Then we denote $\mathbf{V}_{t_1,t_2} := (\boldsymbol{\theta}_{t_1} - \boldsymbol{\theta}^*)(\boldsymbol{\theta}_{t_2} - \boldsymbol{\theta}^*)^\top$, by the recursive formula (32) from $\boldsymbol{\theta}_{t_1}$ to $\boldsymbol{\theta}_{t_2}$, we obtain the expectation of $\mathbf{V}_{t_1,t_2}$ as below. When $t_1 > t_2$, we have

$$
\begin{aligned}
\mathbb{E}\left[\mathbf{V}_{t_1,t_2}\right] &= (\mathbf{Q}^*)^{t_1-t_2}\mathbb{E}\left[\mathbf{V}_{t_2,t_2}\right] + \mathcal{O}(\eta\rho\epsilon_0^3 \cdot \mathbf{I}), \\
&= (\mathbf{Q}^*)^{t_1-t_2}\mathbb{E}\left[\mathbf{V}_{t_2,t_2}\right] + \mathbf{Err}
\end{aligned}
\tag{20}
$$

where the second term in the first equality is from $\mathcal{B}(\boldsymbol{\theta}_{t_2} - \boldsymbol{\theta}^*)^\top$. We obtain

$$
\begin{aligned}
\mathbb{E}\left[\mathcal{B}(\boldsymbol{\theta}_{t_2} - \boldsymbol{\theta}^*)^\top\right] &\preceq \mathbb{E}\left\|\mathcal{B}(\boldsymbol{\theta}_{t_2} - \boldsymbol{\theta}^*)^\top\right\| \cdot \mathbf{I} \\
&\overset{(a)}{\preceq} \mathcal{O}\left(\eta\rho\mathbb{E}\left[\sum_{j=1}^{t_1-t_2}\left(\|\boldsymbol{\theta}_{t_1-j} - \boldsymbol{\theta}^*\|^{2|\alpha|} + \|\boldsymbol{\theta}_{t_1-j} - \boldsymbol{\theta}^*\|^2\right) \cdot \|\boldsymbol{\theta}_{t_2} - \boldsymbol{\theta}^*\|\right]\right) \cdot \mathbf{I} \\
&\preceq \mathcal{O}\left(\eta\rho\left(\epsilon_0^{2|\alpha|+1} + \epsilon_0^3\right)\right) \cdot \mathbf{I}.
\end{aligned}
$$

In (a) we apply the Assumption 3.7, $\mathbb{E}\left[\tilde{\nabla}F(\boldsymbol{\theta}_t;(\mathbf{x}_t,y_t))\right] = (\mathbf{H}^*)^{-2\alpha}\nabla f(\boldsymbol{\theta}_t)$ and $\nabla f(\boldsymbol{\theta}^*) = 0$ to $\mathcal{B}$ and obtain that for any $t \in [t_2, t_1 - 1]$ we have

$$\left\|(\mathbf{H}_t)^{-2\alpha}\nabla f(\boldsymbol{\theta}_t) - (\mathbf{H}^*)^{-2\alpha}\nabla f(\boldsymbol{\theta}^*) - (\mathbf{H}^*)^{1-\alpha}(\boldsymbol{\theta}_t - \boldsymbol{\theta}^*)\right\|$$
$$\leq \left\|\left((\mathbf{H}_t)^{-2\alpha} - (\mathbf{H}^*)^{-2\alpha}\right)\nabla f(\boldsymbol{\theta}_t)\right\|$$
$$+ \left\|(\mathbf{H}^*)^{-2\alpha}\left(\nabla f(\boldsymbol{\theta}_t) - \nabla f(\boldsymbol{\theta}^*) - \mathbf{H}^*(\boldsymbol{\theta}_t - \boldsymbol{\theta}^*)\right)\right\| \tag{21}$$
$$\leq \mathcal{O}\left(\rho\|\boldsymbol{\theta}_t - \boldsymbol{\theta}^*\|^{2|\alpha|} + \rho\|\boldsymbol{\theta}_t - \boldsymbol{\theta}^*\|^2\right).$$

Thus, we denote $\mathbf{Err} = \mathcal{O}\left(\eta\rho\left(\epsilon_0^{2|\alpha|+1} + \epsilon_0^3\right)\cdot\mathbf{I}\right)$ to represent the higher-order infinitesimal term. Similarly, when $t_1 < t_2$, we have

$$\mathbb{E}\left[\mathbf{V}_{t_1,t_2}\right] = \mathbb{E}\left[\mathbf{V}_{t_1,t_1}\right]\left((\mathbf{Q}^*)^{t_2-t_1}\right)^\top + \mathbf{Err}. \tag{22}$$

Then we compute the recursive formula when $t_1 = t_2$. Applying $t_2 = t_1 - 1$ to the recursive formula (32) and take the expectation of two sides, we have

$$\mathbb{E}\left[\mathbf{V}_{t_1,t_1}\right] = \mathbf{Q}^*\mathbb{E}\left[\mathbf{V}_{t_1-1,t_1-1}\right](\mathbf{Q}^*)^\top + \eta^2\mathbb{E}[\mathcal{E}\mathcal{E}^\top] + \mathbf{Err}, \tag{23}$$

where $\mathcal{E} = \mathbb{E}\left[\tilde{\nabla}F(\boldsymbol{\theta}_{t_1-1};(\mathbf{x}_{t_1-1},y_{t_1-1}))\right] - \tilde{\nabla}F(\boldsymbol{\theta}_{t_1-1};(\mathbf{x}_{t_1-1},y_{t_1-1}))$. The second term is from $\mathbb{E}\left[\mathcal{C}\mathcal{C}^\top\right]$; the third term $\mathbf{Err}$ is form $\mathbb{E}\left[\mathcal{A}\mathcal{B}^\top + \mathcal{B}\mathcal{A}^\top + \mathcal{B}\mathcal{B}^\top\right]$, which is on the order of $\mathcal{O}(\eta\rho\|\boldsymbol{\theta} - \boldsymbol{\theta}^*\|^3\cdot\mathbf{I})$; and $\mathbb{E}\left[\mathcal{A}\mathcal{C}^\top + \mathcal{B}\mathcal{C}^\top + \mathcal{C}\mathcal{A}^\top + \mathcal{C}\mathcal{B}^\top\right] = 0$. We calculate the second term as

$$\mathcal{E}\mathcal{E}^\top = \mathcal{E}\mathcal{E}^\top - \mathcal{E}^*\mathcal{E}^{*\top} + \mathcal{E}^*\mathcal{E}^{*\top}, \tag{24}$$

where $\mathcal{E}^* = \mathbb{E}\left[\tilde{\nabla}F(\boldsymbol{\theta}^*;(\mathbf{x}_{t_1-1},y_{t_1-1}))\right] - \tilde{\nabla}F(\boldsymbol{\theta}^*;(\mathbf{x}_{t_1-1},y_{t_1-1}))$. Then we obtain that $\mathbb{E}\left[\mathcal{E}\mathcal{E}^\top - \mathcal{E}^*\mathcal{E}^{*\top}\right]$ is on the order of $\mathcal{O}(\epsilon_0)$ due to the gradient uniform continuity in Assumption 3.6. For simplicity, we denote $\tilde{\nabla}F(\boldsymbol{\theta}_t;(\mathbf{x}_{t_1-1},y_{t_1-1})) = \tilde{\nabla}F_t$ and $\tilde{\nabla}F(\boldsymbol{\theta}^*;(\mathbf{x}_{t_1-1},y_{t_1-1})) = \tilde{\nabla}F^*$

$$\mathbb{E}\left[\mathcal{E}\mathcal{E}^\top - \mathcal{E}^*\mathcal{E}^{*\top}\right] = \mathbb{E}\left[\tilde{\nabla}F_t\tilde{\nabla}^\top F_t - \tilde{\nabla}F^*\tilde{\nabla}^\top F^*\right] - \mathbb{E}\left[\tilde{\nabla}F_t\right]\mathbb{E}\left[\tilde{\nabla}^\top F_t\right], \tag{25}$$

For the first term we have

$$\mathbb{E}\left[\tilde{\nabla}F_t\tilde{\nabla}^\top F_t - \tilde{\nabla}F^*\tilde{\nabla}^\top F^*\right]$$
$$= \mathbb{E}\left[(\mathbf{H}_t)^{-\alpha}\mathbf{u}\mathbf{u}^\top(\mathbf{H}_t)^{-\alpha}\nabla F_t\nabla^\top F_t(\mathbf{H}_t)^{-\alpha}\mathbf{u}\mathbf{u}^\top(\mathbf{H}_t)^{-\alpha}\right]$$
$$- \mathbb{E}\left[(\mathbf{H}^*)^{-\alpha}\mathbf{u}\mathbf{u}^\top(\mathbf{H}^*)^{-\alpha}\nabla F^*\nabla^\top F^*(\mathbf{H}^*)^{-\alpha}\mathbf{u}\mathbf{u}^\top(\mathbf{H}^*)^{-\alpha}\right]$$
$$= \mathbb{E}\left[(\mathbf{H}_t)^{-\alpha}\mathbf{u}\mathbf{u}^\top(\mathbf{H}_t)^{-\alpha}\nabla F_t\underbrace{\left(\nabla^\top F_t(\mathbf{H}_t)^{-\alpha}\mathbf{u}\mathbf{u}^\top(\mathbf{H}_t)^{-\alpha} - \nabla^\top F^*(\mathbf{H}^*)^{-\alpha}\mathbf{u}\mathbf{u}^\top(\mathbf{H}^*)^{-\alpha}\right)}_{\zeta_1}\right]$$
$$- \mathbb{E}\left[\underbrace{\left((\mathbf{H}_t)^{-\alpha}\mathbf{u}\mathbf{u}^\top(\mathbf{H}_t)^{-\alpha}\nabla F_t - (\mathbf{H}^*)^{-\alpha}\mathbf{u}\mathbf{u}^\top(\mathbf{H}^*)^{-\alpha}\nabla F^*\right)}_{\zeta_2}\nabla^\top F^*(\mathbf{H}^*)^{-\alpha}\mathbf{u}\mathbf{u}^\top(\mathbf{H}^*)^{-\alpha}\right]$$

Due to Assumption 3.6, we have

$$\mathbb{E}\|\zeta_1\| \leq \mathbb{E}\left\|\nabla^\top F_t\left((\mathbf{H}_t)^{-\alpha}\mathbf{u}\mathbf{u}^\top(\mathbf{H}_t)^{-\alpha} - (\mathbf{H}^*)^{-\alpha}\mathbf{u}\mathbf{u}^\top(\mathbf{H}^*)^{-\alpha}\right)\right\|$$
$$+ \mathbb{E}\left\|\left(\nabla^\top F_t - \nabla^\top F^*\right)(\mathbf{H}^*)^{-\alpha}\mathbf{u}\mathbf{u}^\top(\mathbf{H}^*)^{-\alpha}\right\|$$
$$\leq \mathbb{E}\left\|\left(\nabla^\top F_t - \nabla^\top F^*\right)\left((\mathbf{H}_t)^{-\alpha}\mathbf{u}\mathbf{u}^\top(\mathbf{H}_t)^{-\alpha} - (\mathbf{H}^*)^{-\alpha}\mathbf{u}\mathbf{u}^\top(\mathbf{H}^*)^{-\alpha}\right)\right\|$$
$$+ \mathbb{E}\left\|\nabla^\top F^*\left((\mathbf{H}_t)^{-\alpha}\mathbf{u}\mathbf{u}^\top(\mathbf{H}_t)^{-\alpha} - (\mathbf{H}^*)^{-\alpha}\mathbf{u}\mathbf{u}^\top(\mathbf{H}^*)^{-\alpha}\right)\right\| \tag{26}$$
$$+ \mathbb{E}\left\|\left(\nabla^\top F_t - \nabla^\top F^*\right)(\mathbf{H}^*)^{-\alpha}\mathbf{u}\mathbf{u}^\top(\mathbf{H}^*)^{-\alpha}\right\|$$
$$\leq \mathcal{O}\left(\mathbb{E}\left[\rho\|\nabla F^*\|\|\boldsymbol{\theta}_t - \boldsymbol{\theta}^*\|^{2|\alpha|}\right] + \mathbb{E}\left[\rho\|\boldsymbol{\theta}_t - \boldsymbol{\theta}^*\|^{1+2|\alpha|}\right] + \mathbb{E}\|\boldsymbol{\theta}_t - \boldsymbol{\theta}^*\|\right)$$
$$\leq \mathcal{O}\left(\rho\epsilon_0^{2|\alpha|} + \epsilon_0\right).$$

Similarly we have

$$\mathbb{E}\|\zeta_2\| \leq \mathcal{O}\left(\rho\epsilon_0^{2|\alpha|} + \epsilon_0\right). \tag{27}$$

Thus $\mathbb{E}\left[\tilde{\nabla}F_t\tilde{\nabla}^\top F_t - \tilde{\nabla}F^*\tilde{\nabla}^\top F^*\right] = \mathcal{O}\left(\left(\rho\epsilon_0^{2|\alpha|} + \epsilon_0\right) \cdot \mathbf{I}\right)$. For the term $\mathbb{E}\left[\tilde{\nabla}F_t\right]\mathbb{E}\left[\tilde{\nabla}^\top F_t\right]$ we have

$$\begin{aligned}
\mathbb{E}\left[\tilde{\nabla}F_t\right]\mathbb{E}\left[\tilde{\nabla}^\top F_t\right] &= (\mathbf{H}_t)^{-2\alpha}\nabla f(\boldsymbol{\theta}_t)\nabla^\top f(\boldsymbol{\theta}_t)(\mathbf{H}_t)^{-2\alpha} \\
&\preceq \mathcal{O}\left(\|\nabla f(\boldsymbol{\theta}_t) - \nabla f(\boldsymbol{\theta}^*)\|^2 \cdot \mathbf{I}\right) \\
&\preceq \mathcal{O}(\|\boldsymbol{\theta}_t - \boldsymbol{\theta}^*\|^2 \cdot \mathbf{I}) \\
&\preceq \mathcal{O}(\epsilon_0 \cdot \mathbf{I}).
\end{aligned} \tag{28}$$

Thus we have $\mathbb{E}\left[\mathcal{E}\mathcal{E}^\top - \mathcal{E}^*\mathcal{E}^{*\top}\right] = \mathcal{O}\left(\left(\rho\epsilon_0^{2|\alpha|} + \epsilon_0\right) \cdot \mathbf{I}\right)$. Then we obtain

$$\begin{aligned}
\mathbb{E}\left[\mathcal{E}\mathcal{E}^\top\right] &= \mathbb{E}\left[\mathcal{E}^*\mathcal{E}^{*\top}\right] + \mathcal{O}\left(\left(\rho\epsilon_0^{2|\alpha|} + \epsilon_0\right) \cdot \mathbf{I}\right) \\
&= \mathbb{E}\left[(\mathbf{H}^*)^{-\alpha}\mathbf{u}\mathbf{u}^\top(\mathbf{H}^*)^{-\alpha}\nabla F^*\nabla^\top F^*(\mathbf{H}^*)^{-\alpha}\mathbf{u}\mathbf{u}^\top(\mathbf{H}^*)^{-\alpha}\right] + \mathcal{O}\left(\left(\rho\epsilon_0^{2|\alpha|} + \epsilon_0\right) \cdot \mathbf{I}\right) \\
&= \mathbb{E}_{\mathbf{u}}\left[(\mathbf{H}^*)^{-\alpha}\mathbf{u}\mathbf{u}^\top(\mathbf{H}^*)^{-\alpha}\mathbf{H}^*(\mathbf{H}^*)^{-\alpha}\mathbf{u}\mathbf{u}^\top(\mathbf{H}^*)^{-\alpha}\right] + \mathcal{O}\left(\left(\rho\epsilon_0^{2|\alpha|} + \epsilon_0\right) \cdot \mathbf{I}\right),
\end{aligned}$$

where in the second equality we use $\mathbb{E}\left[\tilde{\nabla}F(\boldsymbol{\theta}^*; (\mathbf{x}_{t_1-1}, y_{t_1-1}))\right] = 0$ and $\tilde{\nabla}F(\boldsymbol{\theta}^*; (\mathbf{x}_{t_1-1}, y_{t_1-1})) = (\mathbf{H}^*)^{-\alpha}\mathbf{u}\mathbf{u}^\top(\mathbf{H}^*)^{-\alpha}\nabla F(\boldsymbol{\theta}^*; (\mathbf{x}_{t_1-1}, y_{t_1-1}))$ when ignoring the higher-order infinitesimal term of $\mu$; in the third equality we use Assumption 3.3. Denoting $\mathbf{M}^* = \mathbb{E}_{\mathbf{u}}\left[(\mathbf{H}^*)^{-\alpha}\mathbf{u}\mathbf{u}^\top(\mathbf{H}^*)^{-\alpha}\mathbf{H}^*(\mathbf{H}^*)^{-\alpha}\mathbf{u}\mathbf{u}^\top(\mathbf{H}^*)^{-\alpha}\right]$, we have

$$\begin{aligned}
\mathbb{E}\left[\mathbf{V}_{t_1,t_1}\right] &= \mathbf{Q}^*\mathbb{E}\left[\mathbf{V}_{t_1-1,t_1-1}\right]\left(\mathbf{Q}^*\right)^\top + \eta^2\mathbf{M}^* + \mathbf{Err} + \mathcal{O}\left(\eta^2\left(\rho\epsilon_0^{2|\alpha|} + \epsilon_0\right) \cdot \mathbf{I}\right) \\
&= \mathbf{Q}^*\mathbb{E}\left[\mathbf{V}_{t_1-1,t_1-1}\right]\left(\mathbf{Q}^*\right)^\top + \eta^2\mathbf{M}^* + \tilde{\mathbf{Err}}.
\end{aligned} \tag{29}$$

In summary, we obtain the recursive formula of $\mathbb{E}\left[\mathbf{V}_{t_1,t_2}\right]$ as

$$\mathbb{E}\left[\mathbf{V}_{t_1,t_2}\right] = \begin{cases} (\mathbf{Q}^*)^{t_1-t_2}\mathbb{E}\left[\mathbf{V}_{t_2,t_2}\right] + \tilde{\mathbf{Err}} & \text{if } t_1 > t_2, \\ \mathbf{Q}^*\mathbb{E}\left[\mathbf{V}_{t_1-1,t_1-1}\right]\left(\mathbf{Q}^*\right)^\top + \eta^2\mathbf{M}^* + \tilde{\mathbf{Err}} & \text{if } t_1 = t_2, \\ \mathbb{E}\left[\mathbf{V}_{t_1,t_1}\right]\left((\mathbf{Q}^*)^{t_2-t_1}\right)^\top + \tilde{\mathbf{Err}} & \text{if } t_1 < t_2, \end{cases} \tag{30}$$

where $\tilde{\mathbf{Err}} = \mathcal{O}\left(\left(\eta\rho\epsilon_0^3 + \eta\rho\epsilon_0^{2|\alpha|+1} + \eta^2\epsilon_0 + \eta^2\rho\epsilon_0^{2|\alpha|}\right) \cdot \mathbf{I}\right)$.

**Least Squares.** For least squares regression (6) with $C = \sigma^2$, we notice that the Hessian matrix is fixed as $\mathbf{H}^* = \frac{1}{\sigma^2}\mathbb{E}[\mathbf{x}\mathbf{x}^\top]$ and the gradient can be written as

$$\begin{aligned}
\nabla F(\boldsymbol{\theta}_t, (\mathbf{x}_t, y_t)) &= -\frac{1}{\sigma^2}\left(y_t - \langle\boldsymbol{\theta}_t, \mathbf{x}_t\rangle\right)\mathbf{x}_t = \frac{1}{\sigma^2}(\langle\boldsymbol{\theta}_t - \boldsymbol{\theta}^*, \mathbf{x}_t\rangle + \epsilon)\mathbf{x}_t \\
&= \frac{\mathbf{x}_t\mathbf{x}_t^\top(\boldsymbol{\theta}_t - \boldsymbol{\theta}^*)}{\sigma^2} + \frac{\epsilon\mathbf{x}_t}{\sigma^2}.
\end{aligned} \tag{31}$$

Thus we have $\mathbb{E}[\tilde{\nabla}F(\boldsymbol{\theta}_t, (\mathbf{x}_t, y_t))] = (\mathbf{H}^*)^{-2\alpha}\mathbf{H}^*(\boldsymbol{\theta}_t - \boldsymbol{\theta}^*)$. Thus the second term in the second equality in (17) is 0. The recursive formula of $\boldsymbol{\theta}_{t_1}$ and $\boldsymbol{\theta}_{t_2}$ can be exactly obtained as

$$\begin{aligned}
\boldsymbol{\theta}_{t_1} - \boldsymbol{\theta}^* &= \underbrace{(\mathbf{Q}^*)^{t_1-t_2}(\boldsymbol{\theta}_{t_2} - \boldsymbol{\theta}^*)}_{\mathcal{A}} \\
&+ \underbrace{\eta\sum_{j=1}^{t_1-t_2}(\mathbf{Q}^*)^{j-1}\left(\mathbb{E}\left[\tilde{\nabla}F(\boldsymbol{\theta}_{t_1-j}; (\mathbf{x}_{t_1-j}, y_{t_1-j}))\right] - \tilde{\nabla}F(\boldsymbol{\theta}_{t_1-j}; (\mathbf{x}_{t_1-j}, y_{t_1-j}))\right)}_{\mathcal{C}},
\end{aligned}$$

for any $T \leq t_1 < t_2 \leq 2T$. Then we similarly obtain the expectation of $\mathbf{V}_{t_1,t_2}$ when $t_1 > t_2$ as

$$\mathbb{E}\left[\mathbf{V}_{t_1,t_2}\right] = (\mathbf{Q}^*)^{t_1-t_2}\mathbb{E}\left[\mathbf{V}_{t_2,t_2}\right], \tag{32}$$

due to $\mathbb{E}[\mathcal{C}(\boldsymbol{\theta}_{t_2} - \boldsymbol{\theta}^*)^\top] = 0$ without **Err**. When $t_1 = t_2$, we obtain

$$\mathbb{E}\left[\mathbf{V}_{t_1,t_1}\right] = \mathbf{Q}^*\mathbb{E}\left[\mathbf{V}_{t_1-1,t_1-1}\right](\mathbf{Q}^*)^\top + \eta^2\mathbb{E}\left[\mathcal{E}\mathcal{E}^\top\right], \tag{33}$$

where $\mathcal{E} = \mathbb{E}\left[\tilde{\nabla}F(\boldsymbol{\theta}_{t_1-j}; (\mathbf{x}_{t_1-j}, y_{t_1-j}))\right] - \tilde{\nabla}F(\boldsymbol{\theta}_{t_1-j}; (\mathbf{x}_{t_1-j}, y_{t_1-j}))$. For quadratic functions, we have $\mathbf{H_t} = \mathbf{H}^*$. Thus we directly obtain

$$\mathbb{E}\left[\mathrm{tr}\left(\mathbf{H}^*\mathcal{E}\mathcal{E}^\top\right)\right] \lesssim \mathrm{tr}^2\left((\mathbf{H}^*)^{1-2\alpha}\right)\mathrm{tr}\left(\mathbf{H}^*\mathbb{E}\left[\mathbf{V}_{t_1-1,t_1-1}\right]\right) + \mathrm{tr}\left(\mathbf{H}^*\mathbf{M}^*\right), \tag{34}$$

where the last inequality is derived from the assumption that $\mathbb{E}_{\mathbf{x}_t}\left[\mathbf{x}_t\mathbf{x}_t^\top\mathbf{B}\mathbf{x}_t\mathbf{x}_t^\top\right] \preceq \mathcal{O}\left(\mathrm{tr}\left(\mathbf{H}^*\mathbf{B}\right)\mathbf{H}^*\right)$ when $\mathbf{B}$ and $\mathbf{H}^*$ share the same orthonormal basis for least squares regression. Thus we obtain the exact recursive formula of $\mathbb{E}[\mathbf{V}_{t_1,t_2}]$ for least squares regression as

$$\mathbb{E}\left[\mathbf{V}_{t_1,t_2}\right] \preceq \begin{cases} (\mathbf{Q}^*)^{t_1-t_2}\mathbb{E}\left[\mathbf{V}_{t_2,t_2}\right] & \text{if } t_1 > t_2, \\ \mathbf{Q}^*\mathbb{E}\left[\mathbf{V}_{t_1-1,t_1-1}\right](\mathbf{Q}^*)^\top + \eta^2\phi(\mathbf{V}_{t_1-1,t_1-1}) & \text{if } t_1 = t_2, \\ \mathbb{E}\left[\mathbf{V}_{t_1,t_1}\right]\left((\mathbf{Q}^*)^{t_2-t_1}\right)^\top & \text{if } t_1 < t_2, \end{cases} \tag{35}$$

where $\phi(\mathbf{V}_{t_1-1,t_1-1}) := \mathcal{O}\left(\|\boldsymbol{\theta}_{t_1-1} - \boldsymbol{\theta}^*\|^2\right)(\mathbf{H}^*)^{2\alpha} + \mathbf{M}^*$.

**Step II.** In this step, we obtain the estimate of the sum of $\mathbb{E}\left[\mathbf{V}_{t_1,t_2}\right]$ for $t_1$ and $t_2$ from $T$ to $2T-1$. First, by the recursive formula (30), we have

$$\mathbb{E}\left[\mathbf{V}_{t_1,t_2}\right] = (\mathbf{Q}^*)^{t_1-T}\mathbb{E}\left[\mathbf{V}_{T,T}\right]\left((\mathbf{Q}^*)^{t_2-T}\right)^\top + \eta^2\underbrace{\sum_{t=T}^{\min\{t_1,t_2\}-1}(\mathbf{Q}^*)^{t_1-t-1}\mathbf{M}^*\left((\mathbf{Q}^*)^{t_2-t-1}\right)^\top}_{\mathcal{I}_{t_1,t_2}} + \tilde{\mathbf{Err}}$$

In this step, we try to estimate $\sum_{t_1,t_2=T}^{2T-1}\mathcal{I}_{t_1,t_2}$. Specifically, for any $t \in [T, 2T-1]$, we denote $\mathcal{I}_{t_1,t_2}(t) = \eta^2(\mathbf{Q}^*)^{t_1-t-1}\mathbf{M}^*\left((\mathbf{Q}^*)^{t_2-t-1}\right)^\top$. Thus we have

$$\sum_{t_1,t_2=t+1}^{2T-1}\mathcal{I}_{t_1,t_2}(t) = \eta^2\sum_{t_1=t+1}^{2T-1}(\mathbf{Q}^*)^{t_1-t-1}\mathbf{M}^*\left(\left(\mathbf{I} - (\mathbf{Q}^*)^{2T-t-1}\right)\left(\mathbf{I} - \mathbf{Q}^*\right)^{-1}\right)^\top$$

$$= \eta^2\left(\left(\mathbf{I} - (\mathbf{Q}^*)^{2T-t-1}\right)\left(\mathbf{I} - \mathbf{Q}^*\right)^{-1}\right)\mathbf{M}^*\left(\left(\mathbf{I} - (\mathbf{Q}^*)^{2T-t-1}\right)\left(\mathbf{I} - \mathbf{Q}^*\right)^{-1}\right)^\top,$$

where we first calculate the sum of $t_2$ from $t+1$ to $2T-1$ given $t_1$; then compute the sum of $t_1$ from $t+1$ to $2T-1$. Both use the matrix-formed summation formula for geometric series. We obtain that

$$\sum_{t=T}^{2T-1}\sum_{t_1,t_2=t+1}^{2T-1}\mathcal{I}_{t_1,t_2}(t) = T\eta^2(\mathbf{I} - \mathbf{Q}^*)^{-1}\mathbf{M}^*\left((\mathbf{I} - \mathbf{Q}^*)^{-1}\right)^\top$$

$$- \eta^2\sum_{t=T}^{2T-1}(\mathbf{Q}^*)^{2T-t-1}(\mathbf{I} - \mathbf{Q}^*)^{-1}\mathbf{M}^*\left((\mathbf{I} - \mathbf{Q}^*)^{-1}\right)^\top$$

$$- \eta^2\sum_{t=T}^{2T-1}(\mathbf{I} - \mathbf{Q}^*)^{-1}\mathbf{M}^*\left((\mathbf{I} - \mathbf{Q}^*)^{-1}\right)^\top\left((\mathbf{Q}^*)^{2T-t-1}\right)^\top$$

$$+ \eta^2\sum_{t=T}^{2T-1}(\mathbf{Q}^*)^{2T-t-1}(\mathbf{I} - \mathbf{Q}^*)^{-1}\mathbf{M}^*\left((\mathbf{I} - \mathbf{Q}^*)^{-1}\right)^\top\left((\mathbf{Q}^*)^{2T-t-1}\right)^\top,$$

where $\sum_{t_1,t_2=T}^{2T-1} \mathcal{I}_{t_1,t_2} = \sum_{t=T}^{2T-1} \sum_{t_1,t_2=t+1}^{2T-1} \mathcal{I}_{t_1,t_2}(t)$. Then, applying Lemma D.3 with $\mathbf{M} = \mathbf{I} - \mathbf{Q}^*$ and $\bar{\mathbf{M}} = \mathbf{M}^*$ to (36), we obtain

$$
\begin{aligned}
\sum_{t=T}^{2T-1} \sum_{t_1,t_2=t+1}^{2T-1} \mathcal{I}_{t_1,t_2}(t) = {} & T(\mathbf{H}^*)^{-(1-2\alpha)}\mathbf{M}^* \left((\mathbf{H}^*)^{-(1-2\alpha)}\right)^\top \\
& - \left(\mathbf{I} - (\mathbf{Q}^*)^T\right)\left(\mathbf{I} - \mathbf{Q}^*\right)^{-1}(\mathbf{H}^*)^{-(1-2\alpha)}\mathbf{M}^* \left((\mathbf{H}^*)^{-(1-2\alpha)}\right)^\top \\
& - (\mathbf{H}^*)^{-(1-2\alpha)}\mathbf{M}^* \left((\mathbf{H}^*)^{-(1-2\alpha)}\right)^\top \left(\left(\mathbf{I} - \mathbf{Q}^*\right)^{-1}\right)^\top \left(\mathbf{I} - (\mathbf{Q}^*)^T\right)^\top \\
& + \sum_{t=T}^{2T-1} (\mathbf{Q}^*)^{2T-t-1}(\mathbf{H}^*)^{-(1-2\alpha)}\mathbf{M}^* \left((\mathbf{H}^*)^{-(1-2\alpha)}\right)^\top \left((\mathbf{Q}^*)^{2T-t-1}\right)^\top.
\end{aligned}
$$

We notice that with $\eta \gtrsim \frac{1}{\lambda_{min}((\mathbf{H}^*)^{1-2\alpha}T)}$, $\mathbf{Q}^* \preceq \left(1 - \eta\lambda_{min}\left((\mathbf{H}^*)^{1-2\alpha}\right)\right)\mathbf{I} \preceq \mathbf{I}$. Thus we have

$$
\begin{aligned}
\sum_{t=T}^{2T-1} \sum_{t_1,t_2=t+1}^{2T-1} \mathcal{I}_{t_1,t_2}(t) \preceq {} & T(\mathbf{H}^*)^{-(1-2\alpha)}\mathbf{M}^* \left((\mathbf{H}^*)^{-(1-2\alpha)}\right)^\top \\
& + \sum_{t=T}^{2T-1} (\mathbf{Q}^*)^{2T-t-1}(\mathbf{H}^*)^{-(1-2\alpha)}\mathbf{M}^* \left((\mathbf{H}^*)^{-(1-2\alpha)}\right)^\top \left((\mathbf{Q}^*)^{2T-t-1}\right)^\top \\
\preceq {} & \left(T + \frac{1 - \left(1 - \eta\lambda_{min}\left((\mathbf{H}^*)^{1-2\alpha}\right)\right)^{2T}}{1 - \left(1 - \eta\lambda_{min}\left((\mathbf{H}^*)^{1-2\alpha}\right)\right)^2}\right)(\mathbf{H}^*)^{-(1-2\alpha)}\mathbf{M}^* \left((\mathbf{H}^*)^{-(1-2\alpha)}\right)^\top \\
\preceq {} & \left(T + \frac{1}{\eta\lambda_{min}\left((\mathbf{H}^*)^{1-2\alpha}\right)}\right)(\mathbf{H}^*)^{-(1-2\alpha)}\mathbf{M}^* \left((\mathbf{H}^*)^{-(1-2\alpha)}\right)^\top
\end{aligned}
$$

The last equality is due to $\eta\lambda_{min}\left((\mathbf{H}^*)^{1-2\alpha}\right) \leq \eta\lambda_{max}\left((\mathbf{H}^*)^{1-2\alpha}\right) \leq 1$.

**Step III.** In this step, we finish the convergence analysis of Theorem 3.8. We first utilize the Taylor expansion of $f\left(\frac{1}{T}\sum_{t=T}^{2T-1}\boldsymbol{\theta}_t\right)$ at $\boldsymbol{\theta}^*$ as below:

$$
f\left(\frac{1}{T}\sum_{t=T}^{2T-1}\boldsymbol{\theta}_t\right) \leq f(\boldsymbol{\theta}^*) + \frac{1}{2}\left(\frac{1}{T}\sum_{t=T}^{2T-1}(\boldsymbol{\theta}_t - \boldsymbol{\theta}^*)\right)^\top \mathbf{H}^* \left(\frac{1}{T}\sum_{t=T}^{2T-1}(\boldsymbol{\theta}_t - \boldsymbol{\theta}^*)\right), \tag{36}
$$

since $\nabla f(\boldsymbol{\theta}^*) = 0$. Then we take the expectation of both sides of (36) and obtain

$$
\begin{aligned}
\mathbb{E}\left[f\left(\frac{1}{T}\sum_{t=T}^{2T-1}\boldsymbol{\theta}_t\right)\right] - f(\boldsymbol{\theta}^*) \overset{(a)}{\leq} {} & \frac{1}{2}\mathbb{E}\left[\mathrm{tr}\left(\mathbf{H}^*\left(\frac{1}{T}\sum_{t=T}^{2T-1}(\boldsymbol{\theta}_t - \boldsymbol{\theta}^*)\right)\left(\frac{1}{T}\sum_{t=T}^{2T-1}(\boldsymbol{\theta}_t - \boldsymbol{\theta}^*)\right)^\top\right)\right] \\
= {} & \frac{1}{2T^2}\mathrm{tr}\left(\mathbf{H}^*\mathbb{E}\left[\sum_{t=T}^{2T-1}\sum_{t_1,t_2=t+1}^{2T-1}\mathbf{V}_{t_1,t_2}\right]\right) \\
\overset{(b)}{=} {} & \frac{1}{2T^2}\mathrm{tr}\left(\mathbf{H}^*\mathbb{E}\left[\sum_{t=T}^{2T-1}\sum_{t_1,t_2=t+1}^{2T-1}\mathcal{I}_{t_1,t_2}(t)\right]\right) \\
& + \frac{1}{2\eta^2 T^2}\mathrm{tr}\left((\mathbf{H}^*)^{4\alpha-1}\mathbb{E}\left[\mathbf{V}_{T,T}\right]\right) + \bar{\mathbf{Err}} \\
\overset{(c)}{\lesssim} {} & \frac{(1 - \eta\lambda_{\min}((\mathbf{H}^*)^{1-2\alpha}))^{2T}}{\eta^2 T^2}\mathrm{tr}\left((\mathbf{H}^*)^{4\alpha-1}\mathbb{E}\left[\mathbf{V}_{0,0}\right]\right) \\
& + \left(\frac{1}{2T} + \frac{1}{2T^2\eta\lambda_{min}((\mathbf{H}^*)^{1-2\alpha})}\right) \cdot D_\alpha + \frac{1}{\eta T^2}D_\alpha + \bar{\mathbf{Err}},
\end{aligned}
$$

where $D_\alpha = \mathrm{tr}\left(\mathbf{H}^*(\mathbf{H}^*)^{-(1-2\alpha)}\mathbf{M}^*\left((\mathbf{H}^*)^{-(1-2\alpha)}\right)^\top\right)$ and $\bar{\mathbf{Err}} = \mathcal{O}\left(\eta\rho\epsilon_0^3 + \eta\rho\epsilon_0^{2|\alpha|+1} + \eta^2\epsilon_0 + \eta^2\rho\epsilon_0^{2|\alpha|}\right)$. To obtain the inequality (a), we use $\mathbf{a}^\top\mathbf{H}\mathbf{a} = \mathrm{tr}\left(\mathbf{H}\mathbf{a}\mathbf{a}^\top\right)$ for any vector $\mathbf{a}$ and

matrix $\mathbf{H}$. (b) is derived from combining (36) with the recursive expression of $\mathbb{E}\left[\mathbf{V}_{T,T}\right]$ in (30) when given $T$. By integrating (36) with the recursive computation procedure for $\mathrm{tr}\left((\mathbf{H}^*)^{4\alpha-1}\mathbb{E}\left[\mathbf{V}_{T,T}\right]\right)$, we have the inequality (c).

Next, we compute the trace expression $\mathrm{tr}\left(\mathbf{H}^*(\mathbf{H}^*)^{-(1-2\alpha)}\mathbf{M}^*\left((\mathbf{H}^*)^{-(1-2\alpha)}\right)^\top\right)$ through the following derivation:

$$\mathrm{tr}\left(\mathbf{H}^*(\mathbf{H}^*)^{-(1-2\alpha)}\mathbf{M}^*\left((\mathbf{H}^*)^{-(1-2\alpha)}\right)^\top\right) = \mathrm{tr}\left((\mathbf{H}^*)^{4\alpha-1}\cdot\mathbf{M}^*\right), \tag{37}$$

where $\mathbf{M}^*$ satisfies:

$$\begin{aligned}
\mathbf{M}^* &= \mathbb{E}_{\mathbf{u}\sim\mathcal{N}(0,\mathbf{I}_d)}\left[(\mathbf{H}^*)^{-\alpha}\mathbf{u}\mathbf{u}^\top(\mathbf{H}^*)^{-\alpha}\mathbf{H}^*(\mathbf{H}^*)^{-\alpha}\mathbf{u}\mathbf{u}^\top(\mathbf{H}^*)^{-\alpha}\right] \\
&\overset{(d)}{\preceq} \mathcal{O}\left((\mathbf{H}^*)^{-2\alpha}\mathrm{tr}\left((\mathbf{H}^*)^{1-2\alpha}\right)\right).
\end{aligned} \tag{38}$$

Inequality (d) is derived from the fact that $\mathbb{E}[\mathbf{A}\mathbf{u}\mathbf{u}^\top\mathbf{B}\mathbf{u}\mathbf{u}^\top\mathbf{A}^\top]\preceq\mathcal{O}\left(\mathbf{A}\mathbf{A}^\top\mathrm{tr}(\mathbf{B})\right)$ when $\mathbf{u}\sim\mathcal{N}(0,\mathbf{I}_d)$ and $\mathbf{A},\mathbf{B}\in\mathbb{R}^{d\times d}$ share the same orthonormal basis. Thus, combing (37) and (38), we obtain

$$\mathrm{tr}\left(\mathbf{H}^*(\mathbf{H}^*)^{-(1-2\alpha)}\mathbf{M}^*\left((\mathbf{H}^*)^{-(1-2\alpha)}\right)^\top\right)\preceq\mathrm{tr}\left((\mathbf{H}^*)^{2\alpha-1}\right)\cdot\mathrm{tr}\left((\mathbf{H}^*)^{1-2\alpha}\right). \tag{39}$$

In the end, we have

$$\begin{aligned}
\mathbb{E}\left[f\left(\frac{1}{T}\sum_{t=T}^{2T-1}\boldsymbol{\theta}_t\right)\right] - f(\boldsymbol{\theta}^*) &\lesssim \frac{(1-\eta\lambda_{\min}((\mathbf{H}^*)^{1-2\alpha}))^{2T}}{\eta^2 T^2} + \frac{1}{\eta T^2}\mathrm{tr}\left((\mathbf{H}^*)^{1-2\alpha}\right) \\
&\quad + \frac{\mathrm{tr}\left((\mathbf{H}^*)^{2\alpha-1}\right)\cdot\mathrm{tr}\left((\mathbf{H}^*)^{1-2\alpha}\right)}{T} + \overline{\mathbf{Err}}.
\end{aligned} \tag{40}$$

We complete the proof of Theorem 3.8.

**Least Squares.** For least squares regression, we obtain

$$\begin{aligned}
\mathbb{E}\left[f\left(\frac{1}{T}\sum_{t=T}^{2T-1}\boldsymbol{\theta}_t\right)\right] - f(\boldsymbol{\theta}^*) &\overset{(a)}{=} \frac{1}{2}\mathbb{E}\left[\mathrm{tr}\left(\mathbf{H}^*\left(\frac{1}{T}\sum_{t=T}^{2T-1}(\boldsymbol{\theta}_t-\boldsymbol{\theta}^*)\right)\left(\frac{1}{T}\sum_{t=T}^{2T-1}(\boldsymbol{\theta}_t-\boldsymbol{\theta}^*)\right)^\top\right)\right] \\
&\overset{(b)}{\leq} \frac{1}{\eta T^2}\sum_{t=T}^{2T-1}\mathrm{tr}\left((\mathbf{H}^*)^{2\alpha}\mathbb{E}\left[\mathbf{V}_{t,t}\right]\right) \\
&\overset{(c)}{\leq} \frac{\left(1-\eta\lambda_{\min}\left((\mathbf{H}^*)^{1-2\alpha}\right)\right)^T}{\eta T}\mathrm{tr}\left((\mathbf{H}^*)^{2\alpha}\mathbb{E}\left[\mathbf{V}_{0,0}\right]\right) + \frac{D_\alpha}{T}, \quad (41)
\end{aligned}$$

where $D_\alpha = \mathrm{tr}\left(\mathbf{H}^*(\mathbf{H}^*)^{-(1-2\alpha)}\mathbf{M}^*\left((\mathbf{H}^*)^{-(1-2\alpha)}\right)^\top\right)$. By $\mathbf{a}^\top\mathbf{H}\mathbf{a} = \mathrm{tr}\left(\mathbf{H}\mathbf{a}\mathbf{a}^\top\right)$ for any vector $\mathbf{a}$ and matrix $\mathbf{H}$, we have inequality (a). (b) follows from the recursive expression of $\mathbb{E}\left[\mathbf{V}_{t_1,t_2}\right]$ in (35) and (c) is obtained from the estimation

$$\begin{aligned}
\mathrm{tr}\left((\mathbf{H}^*)^{2\alpha}\mathbb{E}\left[\mathbf{V}_{t,t}\right]\right) &\leq \underbrace{\mathrm{tr}\left((\mathbf{H}^*)^{2\alpha}(\mathbf{Q}^*)^t\mathbb{E}\left[\mathbf{V}_{0,0}\right]\left((\mathbf{Q}^*)^t\right)^\top\right)}_{\mathcal{I}} \\
&\quad + \eta^2\sum_{t'=0}^{t-1}\mathcal{O}\left(\mathbb{E}\left[\|\boldsymbol{\theta}_{t'}-\boldsymbol{\theta}^*\|^2\right]\right)\mathrm{tr}\left((\mathbf{H}^*)^{2\alpha}(\mathbf{Q}^*)^{t-1-t'}(\mathbf{H}^*)^{2\alpha}\left((\mathbf{Q}^*)^{t-1-t'}\right)^\top\right) \\
&\quad + \underbrace{\eta^2\sum_{t'=0}^{t-1}\mathrm{tr}\left((\mathbf{H}^*)^{2\alpha}(\mathbf{Q}^*)^{t-1-t'}\mathbf{M}^*\left((\mathbf{Q}^*)^{t-1-t'}\right)^\top\right)}_{\mathcal{II}} \\
&\overset{(d)}{\leq} \left(1-\eta\lambda_{\min}\left((\mathbf{H}^*)^{1-2\alpha}\right)\right)^T\mathrm{tr}\left((\mathbf{H}^*)^{2\alpha}\mathbb{E}\left[\mathbf{V}_{0,0}\right]\right) + \eta D_\alpha, \tag{42}
\end{aligned}$$

for any $t \in [T : 2T - 1]$, where (d) is derived from combining the following recursion

$$\mathbb{E}\left[\|\boldsymbol{\theta}_t - \boldsymbol{\theta}^*\|^2\right] = \text{tr}\left(\mathbb{E}\left[\mathbf{V}_{t,t}\right]\right) \leq \text{tr}\left(\mathbf{Q}^*\mathbb{E}\left[\mathbf{V}_{t-1,t-1}\right](\mathbf{Q}^*)^\top\right)$$
$$+ \eta^2\left[\text{tr}\left((\mathbf{H}^*)^{-2\alpha}\right)\text{tr}\left((\mathbf{H}^*)^{1-2\alpha}\right)\text{tr}\left(\mathbf{H}^*\mathbb{E}\left[\mathbf{V}_{t-1,t-1}\right]\right) + \text{tr}\left(\mathbf{M}^*\right)\right]$$
$$\overset{(e)}{\leq} \left(1 - \eta\lambda_{\min}\left((\mathbf{H}^*)^{1-2\alpha}\right)\right)\text{tr}\left(\mathbb{E}\left[\mathbf{V}_{t-1,t-1}\right]\right) + \eta^2\text{tr}\left(\mathbf{M}^*\right),$$
$$(43)$$

with explicit computational procedures applied to parameters $\mathcal{I}$ and $\mathcal{II}$, where (e) is achieved through the setting of step size that $\eta \leq \frac{\lambda_{\min}(\mathbf{H}^*)}{\lambda_{\max}(\mathbf{H}^*)\text{tr}((\mathbf{H}^*)^{-2\alpha})\text{tr}((\mathbf{H}^*)^{1-2\alpha})}$. According to (41), we complete the proof of Theorem 3.5. $\qquad\square$

## B  Extensive Analysis under Approximate Hessian

In this section, we further consider the PSPSA using approximate Hessian $\tilde{\mathbf{H}}_t$ to replace $\mathbf{H}_t$. Previous work[†] shows that without exact calculation, approximate Hessian can be obtained through zeroth-order oracles with a controlled gap between $\tilde{\mathbf{H}}_t$ and $\mathbf{H}_t$. Specifically, for least squares regression, due to $\mathbf{H}_t = \mathbf{H}^*$, we formally propose the Assumption B.1 below to characterize the approximate error of Hessian.

**Assumption B.1.** *Given $\alpha > 0$, the Hessian estimation matrix $\tilde{\mathbf{H}}_t$ satisfies*

$$\left\|\tilde{\mathbf{H}}_t^{2\alpha} - (\mathbf{H}^*)^{2\alpha}\right\| \leq \alpha\epsilon^{2\alpha}. \qquad (44)$$

With Assumption B.1, we obtain the convergence rate of PaZO with approximate Hessian for least squares regression in Theorem B.2. The complexity of estimating Hessian can be lower bounded by the rate in Theorem B.2 since we only need to estimate the Hessian one time for least squares regression due to $\mathbf{H}_t = \mathbf{H}^*$.

**Theorem B.2.** *Suppose $\alpha \in [0, 1/2]$ and the Hessian approximation error $\epsilon$ defined in Assumption B.1 satisfies $\epsilon \leq \mathcal{O}\left((\kappa(\mathbf{H}^*))^{-1/(2\alpha)}\right)\lambda_{\min}(\mathbf{H}^*)$ where $\kappa(\mathbf{H}^*) = \lambda_{\max}(\mathbf{H}^*)/\lambda_{\min}(\mathbf{H}^*)$. Consider running PaZO with approximate Hessian $\tilde{\mathbf{H}}_t$ satisfying Assumption B.1 for the least squares regression problem (6) under Assumption 3.4, with a learning rate $\eta$ satisfying $\eta = \tilde{\mathcal{O}}\left((\lambda_{\min}(\mathbf{H}^*))^{2\alpha-1}T^{-1}\right)$ for $T$ iterations. Then PaZO achieves the following convergence rate:*

$$\mathbb{E}\left[\|\boldsymbol{\theta}_T - \boldsymbol{\theta}^*\|_{\mathbf{H}^*}^2\right] \lesssim \left(1 - \eta(\lambda_{\min}(\mathbf{H}^*))^{1-2\alpha}\right)^T \|\boldsymbol{\theta}_0 - \boldsymbol{\theta}^*\|_{\mathbf{H}^*}^2 + \frac{3d^2\sigma^2(\kappa(\mathbf{H}^*))^{2-4\alpha}}{T}, \quad (45)$$

*where $\alpha$ is the precondition order defined in PSPSA.*

In Theorem B.2, the first term decays exponentially as $T$, and the dominant term of the rate is $3d^2\sigma^2(\kappa(\mathbf{H}^*))^{2-4\alpha}/T$. Since $\kappa$ is defined as the condition number of a given positive definite matrix, we notice that $(\kappa(\mathbf{H}^*))^{2-4\alpha} \geq 1$ and the equality holds if and only if $\alpha = 1/2$. This result amazingly aligns with the results in Theorem 3.5, which demonstrates that the optimal selection of $\alpha$ in PSPSA is $1/2$. Without an effect preconditioner, ZO-SGD only achieves $\tilde{\mathcal{O}}(d^2\sigma^2(\kappa(\mathbf{H}^*))^2/T)$, not matching the ideal rate $\tilde{\mathcal{O}}(d^2\sigma^2/T)$. We provide the proof of Theorem B.2 as follows.

*Proof.* According to the update rule

$$\boldsymbol{\theta}_{t+1} = \boldsymbol{\theta}_t - \eta\tilde{\nabla}F(\boldsymbol{\theta}_t; (\mathbf{x}_t, y_t)), \qquad (46)$$

---

[†] Qian Yu et al. "Stochastic Zeroth-Order Optimization under Strongly Convexity and Lipschitz Hessian: Minimax Sample Complexity". In: arXiv preprint arXiv:2406.19617 (2024).

we have

$$\mathbb{E}\left[\|\boldsymbol{\theta}_{t+1} - \boldsymbol{\theta}^*\|_{\mathbf{H}^*}^2\right] \overset{(a)}{\leq} \mathbb{E}\left[\|\boldsymbol{\theta}_t - \boldsymbol{\theta}^*\|_{\mathbf{H}^*}^2\right] - 2\eta\mathbb{E}\left[\left\langle\boldsymbol{\theta}_t - \boldsymbol{\theta}^*, \tilde{\mathbf{H}}_t^{-2\alpha}\mathbf{H}^*(\boldsymbol{\theta}_t - \boldsymbol{\theta}^*)\right\rangle_{\mathbf{H}^*}\right]$$

$$+ \eta^2\mathrm{tr}^2\left(\tilde{\mathbf{H}}_t^{-2\alpha}\mathbf{H}^*\right)\mathbb{E}\left[\|\boldsymbol{\theta}_t - \boldsymbol{\theta}^*\|_{\mathbf{H}^*}^2\right] + \eta^2\sigma^2\mathrm{tr}^2(\tilde{\mathbf{H}}_t^{-2\alpha}\mathbf{H}^*)$$

$$\overset{(b)}{\leq} \left(1 - 2\eta\left(\lambda_{\min}(\mathbf{H}^*)\right)^{1-2\alpha}\right)\mathbb{E}\left[\|\boldsymbol{\theta}_t - \boldsymbol{\theta}^*\|_{\mathbf{H}^*}^2\right]$$

$$+ 2\eta\frac{\lambda_{\max}(\mathbf{H}^*)\epsilon^{2\alpha}}{\left(\lambda_{\min}^{2\alpha}(\mathbf{H}^*) - \epsilon^{2\alpha}\right)\lambda_{\min}^{2\alpha}(\mathbf{H}^*)}\mathbb{E}\left[\|\boldsymbol{\theta}_t - \boldsymbol{\theta}^*\|_{\mathbf{H}^*}^2\right]$$

$$+ \eta^2\frac{\mathrm{tr}^2(\mathbf{H}^*)}{\left(\lambda_{\min}^{2\alpha}(\mathbf{H}^*) - \epsilon^{2\alpha}\right)^2}\mathbb{E}\left[\|\boldsymbol{\theta}_t - \boldsymbol{\theta}^*\|_{\mathbf{H}^*}^2\right]$$

$$+ 2\eta^2\sigma^2\left[\mathrm{tr}^2\left((\mathbf{H}^*)^{1-2\alpha}\right) + \left(\frac{\mathrm{tr}(\mathbf{H}^*)\epsilon^{2\alpha}}{\left(\lambda_{\min}^{2\alpha}(\mathbf{H}^*) - \epsilon^{2\alpha}\right)\lambda_{\min}^{2\alpha}(\mathbf{H}^*)}\right)^2\right]$$

$$\overset{(c)}{\leq} \left(1 - \eta\left(\lambda_{\min}(\mathbf{H}^*)\right)^{1-2\alpha}\right)\mathbb{E}\left[\|\boldsymbol{\theta}_t - \boldsymbol{\theta}^*\|_{\mathbf{H}^*}^2\right]$$

$$+ 3\eta^2\sigma^2\mathrm{tr}^2\left((\mathbf{H}^*)^{1-2\alpha}\right),$$

where (a) is derived from Assumption 3.4, (b) follows the fact $\lambda_{\min}(\tilde{\mathbf{H}}_t^{-2\alpha}) \leq \left(\lambda_{\min}^{2\alpha}(\mathbf{H}^*) - \epsilon^{2\alpha}\right)^{-1}$ and $\left\|\tilde{\mathbf{H}}_t^{-2\alpha} - (\mathbf{H}^*)^{-2\alpha}\right\| \leq \epsilon^{2\alpha}/\left[\left(\lambda_{\min}^{2\alpha}(\mathbf{H}^*) - \epsilon^{2\alpha}\right)\lambda_{\min}^{2\alpha}(\mathbf{H}^*)\right]$, and (c) is obtained from the setting of $\eta$ and Assumption B.1. According to the recursive expression of $\mathbb{E}\left[\|\boldsymbol{\theta}_t - \boldsymbol{\theta}^*\|_{\mathbf{H}^*}^2\right]$, we obtain

$$\mathbb{E}\left[\|\boldsymbol{\theta}_T - \boldsymbol{\theta}^*\|_{\mathbf{H}^*}^2\right] \leq \left(1 - \eta\left(\lambda_{\min}(\mathbf{H}^*)\right)^{1-2\alpha}\right)^T\|\boldsymbol{\theta}_0 - \boldsymbol{\theta}^*\|_{\mathbf{H}^*}^2$$

$$+ \frac{3\eta\sigma^2}{\lambda_{\min}^{1-2\alpha}(\mathbf{H}^*)}\mathrm{tr}^2\left((\mathbf{H}^*)^{1-2\alpha}\right). \tag{47}$$

By applying the chosen value of $\eta$ to (47), we complete the proof. $\qquad\square$

## C  Experiment Setup

### C.1  Dataset

For RoBERTa-large, we consider classification datasets: SST-2 [47], SST-5 [47], TREC [51], MNLI [54], SNLI [7], and RTE [20, 14, 18, 6]. We follow [37] to limit the test set with 1,000 examples for fast iteration. For training and validation, we set $k = 16$, which means that we have 16 examples per class for both training and validation.

For OPT-1.3B, we consider the SuperGLUE dataset collection [52], including: BoolQ [13], CB [15], COPA [45], MultiRC [27], ReCoRD [56], RTE [20, 14, 18, 6], WiC [43], and WSC [30]. We also consider SST-2 [47] and report the results on the above 9 dataset with randomly sampling 1,000 examples for training, 500 examples for validation, and 1,000 examples for testing.

### C.2  Hyperparameters

We use the hyperparameters in Table 3 for experiments on RoBERTa-large. Previous work [37] shows that the choice of $\epsilon$ seems to not significantly impact the performance, and using a larger batch size consistently yielded faster optimization. We use the hyperparameters in Table 4 for zeroth-order methods on OPT-1.3B. We use linear learning scheduling for first-order fine-tuning methods with backpropagation, and constant learning rate for all zeroth-order methods.

For RoBERTa-large experiments, we evaluate the model on validation sets every $1/10$ of total training steps and save the best validation checkpoint. All FT experiments use $1K$ steps and zeroth-order methods use $100K$ steps. For OPT-1.3B experiments, we evaluate the model on validation sets every $1/5$ of the total training steps and save the best validation checkpoint. All zeroth-order methods in experiments use $20K$ steps.

**Algorithm 3** MeZO

**Require:** parameters $\Theta = \{\boldsymbol{\theta}_i \in \mathbb{R}^{d_i}\}$, loss $\mathcal{L} : \mathbb{R}^d \to \mathbb{R}$, running steps $T$, perturbation scale $\mu$, learning rate schedule $\eta_t$, random seed $s$, a random number generator
  **for** $t = 1, ..., T$ **do**
    **Step 1**: Perturb Parameters through Diagonal Hessian
      Sample batch $\mathcal{B} \subset \mathcal{D}$ and random seed $s$
      $\boldsymbol{\theta} \leftarrow \text{PerturbParameters}(\boldsymbol{\theta}, \mu, \mathbf{I}, s)$
      $\ell_+ \leftarrow \mathcal{L}(\boldsymbol{\theta}; \mathcal{B})$
      $\boldsymbol{\theta} \leftarrow \text{PerturbParameters}(\boldsymbol{\theta}, -2\mu, \mathbf{I}, s)$
      $\ell_- \leftarrow \mathcal{L}(\boldsymbol{\theta}; \mathcal{B})$
      $\boldsymbol{\theta} \leftarrow \text{PerturbParameters}(\boldsymbol{\theta}, \mu, \mathbf{I}, s)$
    **Step 2**: Update the Parameters
      Reset random number generator with seed $s$
      projected_grad $\leftarrow (\ell_+ - \ell_-)/2\mu$
    **for** $\boldsymbol{\theta}_i \in \Theta$ **do**
      Sample $\mathbf{u}_i \sim \mathcal{N}(0, \mathbf{I}_{d_i})$
      $\boldsymbol{\theta}_i \leftarrow \boldsymbol{\theta}_i - \eta_t * \text{projected\_grad} * \mathbf{u}_i$
    **end for**
  **end for**

## C.3 Parameter-efficient Fine-tuning

Storing and fine-tuning a large language model for each downstream task can be quite costly. Parameter-efficient fine-tuning (PEFT) techniques help mitigate this issue: instead of fine-tuning all model parameters, PEFT only modifies a small percentage of additional parameters (usually less than 1%) and often achieves comparable or better performance [23, 31]. The zeroth-order optimizer is compatible with PEFT methods because it can operate on any subset of the model parameters. We conduct experiments with the following two common PEFT methods: LoRA [23] and prefix-tuning [31].

**LoRA [23]** enhances a linear layer during fine-tuning by adding a tunable low-rank delta. Initially, the linear layer is defined as $\mathbf{Wx} + \mathbf{b}$ during pre-training, where $\mathbf{W} \in \mathbb{R}^{m \times n}$. During fine-tuning, LoRA introduces two smaller matrices $\mathbf{A} \in \mathbb{R}^{m \times r}$ and $\mathbf{B} \in \mathbb{R}^{r \times n}$ such that $r \ll \min\{m, n\}$. Consequently, the modified linear layer becomes

$$\left(\mathbf{W} + \frac{\alpha}{r}\mathbf{AB}\right)\mathbf{x} + \mathbf{b}, \tag{48}$$

where $\alpha$ and $r$ are hyperparameters. $\mathbf{A}$ and $\mathbf{B}$ are trained on the downstream tasks while $\mathbf{W}$ is frozen at its pre-trained value. $r$ is empirically small and we choose $r = 8$ and $\alpha = 16$ in our experiments.

**Prefix-tuning [31]** is a technique where a prefix of $m$ tunable representations is added at each layer, while the remaining parts of the model are frozen. These added representations function as new keys and values, serving as additional context during the attention operation. The initialization of these tunable representations involves randomly sampling tokens from the vocabulary and passing them through the LLMs to obtain their keys and values at various attention layers. In our experiments, setting $m = 5$ proved sufficient to achieve good performance on most tasks.

## C.4 Zeroth-order Optimziers

Zeroth-order optimization for fine-tuning LLMs has become a matter of concern recently, showing great potential for reducing the memory overhead during fine-tuning tasks. We introduce two representative zeroth-order optimizers: MeZO [37] and HiZOO [59], and explain that they are both special case of the PSPSA we propose with a specific choice of $\alpha$.

**MeZO [37]** is stated in Algorithm 3, with Simultaneous Perturbation Stochastic Approximation or SPSA [48] to estimate the zeroth-order stochastic gradient with two forward passes. When $\mu \to 0$, it can be regarded to use an 1-rank stochastic gradient for the update. From the perspective of PSPSA, MeZO can be regarded to set $\alpha = 0$ in PSPSA, as we state in Algorithm 3 with $\mathbf{I}$ as a "preconditioner".

Table 3: The hyperparameter grids used for RoBERTa-large experiments. MeZO and PaZO uses a constant learning rate schedule. All MeZO and PaZO experiments use 100K steps.

| Experiment | Hyperparameters | Values |
|---|---|---|
| MeZO | Batch size | 64 |
| | Learning rate | $\{1e-7, 1e-6, 1e-5\}$ |
| | $\mu$ | $1e-3$ |
| | Weight Decay | 0 |
| MeZO (prefix) | Batch size | 64 |
| | Learning rate | $\{1e-2, 5e-3, 1e-3\}$ |
| | $\mu$ | $1e-1$ |
| | Weight Decay | 0 |
| | # prefix tokens | 5 |
| MeZO (LoRA) | Batch size | 64 |
| | Learning rate | $\{1e-5, 5e-5, 1e-4\}$ |
| | $\mu$ | $1e-3$ |
| | Weight Decay | 0.1 |
| | $(r, \alpha)$ | $(8, 16)$ |
| PaZO | Batch size | 64 |
| | Learning rate | $\{1e-7, 1e-6, 1e-5\}$ |
| | $\mu$ | $1e-3$ |
| | Weight Decay | 0 |
| PaZO (prefix) | Batch size | 64 |
| | Learning rate | $\{1e-2, 5e-3, 1e-3\}$ |
| | $\mu$ | $1e-1$ |
| | Weight Decay | 0 |
| | # prefix tokens | 5 |
| PaZO (LoRA) | Batch size | 64 |
| | Learning rate | $\{1e-5, 5e-5, 1e-4\}$ |
| | $\mu$ | $1e-3$ |
| | Weight Decay | 0.1 |
| | $(r, \alpha)$ | $(8, 16)$ |
| FT | Batch size | $\{2, 4, 8\}$ |
| | Learning rate | $\{1e-5, 3e-5, 5e-5\}$ |
| | Weight Decay | 0 |
| FT (prefix) | Batch size | $\{8, 16, 32\}$ |
| | Learning rate | $\{1e-2, 3e-2, 5e-2\}$ |
| | Weight Decay | 0 |
| | # prefix tokens | 5 |
| FT (LoRA) | Batch size | $\{4, 8, 16\}$ |
| | Learning rate | $\{1e-4, 3e-4, 5e-4\}$ |
| | $(r, \alpha)$ | $(8, 16)$ |

**HiZOO [59]** is stated in Algorithm 4, with preconditioned SPSA with $\mathbf{H}^{-1/2}$ as the preconditioner in the perturbation, and $\mathbf{H}$ as the preconditioner in the estimated stochastic gradient. In other words, HiZOO can be regarded as setting $\alpha = 1/2$ in PSPSA. In our theoretical analysis, the optimal selection of $\alpha$ is $1/2$, however, we empirically show the best performance of PaZO compared with MeZO and HiZOO with the same hyperparameter setting through our experiments.

## C.5 Details about Memory Usage

We show the detailed peak memory overhead results in Table 5. We set the per-device batch size to 1 to obtain the minimum peak memory overhead of the corresponding models and methods, We also do not turn on any advanced memory-saving options, e.g., gradient checkpointing. We directly use Nvidia's *nvidia-smi* command to monitor the GPU peak memory overhead.

Table 4: The hyperparameter grids used for OPT-1.3B experiments. All weight decay is set to 0. PaZO uses 20K steps and constant learning rates.

| Experiment | Hyperparameters | Values |
|---|---|---|
| MeZO | Batch size | 16 |
| | Learning rate | $\{1e-6, 5e-7, 1e-7\}$ |
| | $\mu$ | $1e-3$ |
| MeZO (prefix) | Batch size | 16 |
| | Learning rate | $\{5e-2, 1e-2, 5e-3\}$ |
| | $\mu$ | $1e-1$ |
| | # prefix tokens | 5 |
| MeZO (LoRA) | Batch size | 16 |
| | Learning rate | $\{1e-4, 5e-5, 1e-5\}$ |
| | $\mu$ | $1e-2$ |
| | $(r, \alpha)$ | $(8, 16)$ |
| HiZOO | Batch size | 16 |
| | Learning rate | $\{1e-6, 5e-7, 1e-7\}$ |
| | $\mu$ | $1e-3$ |
| HiZOO (prefix) | Batch size | 16 |
| | Learning rate | $\{5e-2, 1e-2, 5e-3\}$ |
| | $\mu$ | $1e-1$ |
| | # prefix tokens | 5 |
| HiZOO (LoRA) | Batch size | 16 |
| | Learning rate | $\{1e-4, 5e-5, 1e-5\}$ |
| | $\mu$ | $1e-2$ |
| | $(r, \alpha)$ | $(8, 16)$ |
| PaZO | Batch size | 16 |
| | Learning rate | $\{1e-6, 5e-7, 1e-7\}$ |
| | $\mu$ | $1e-3$ |
| PaZO (prefix) | Batch size | 16 |
| | Learning rate | $\{5e-2, 1e-2, 5e-3\}$ |
| | $\mu$ | $1e-1$ |
| | # prefix tokens | 5 |
| PaZO (LoRA) | Batch size | 16 |
| | Learning rate | $\{1e-4, 5e-5, 1e-5\}$ |
| | $\mu$ | $1e-2$ |
| | $(r, \alpha)$ | $(8, 16)$ |

Table 5: Peak memory on the MultiRC (average tokens=400) dataset.

| Method | zero-shot/MeZO | PaZO | ICL | FT | FT (prefix) |
|---|---|---|---|---|---|
| 1.3B | 1xA6000 (4GB) | 1xA6000 (9GB) | 1xA6000 (6GB) | 1xA6000 (27GB) | 1xA6000 (19GB) |
| 2.7B | 1xA6000 (7GB) | 1xA6000 (14GB) | 1xA6000 (8GB) | 2xA6000 (55GB) | 1xA6000 (29GB) |
| 6.7B | 1xA6000 (14GB) | 1xA6000 (30GB) | 1xA6000 (16GB) | 4xA6000 (156GB) | 1xA6000 (46GB) |
| 13B | 1xA6000 (26GB) | 2xA6000 (54GB) | 1xA6000 (29GB) | 8xA6000 (316GB) | 4xA6000 (158GB) |

## C.6 Wall-clock Time

We report the steps and wall-clock time required to reach 60% accuracy on a representative task RTE in Table 6. These results support our claim that PaZO achieves better convergence speed than existing zeroth-order methods by leveraging an ideal choice of the preconditioner order. Both the required number of steps and the total training time to reach the target accuracy are smaller for PaZO, validating our theoretical insights. Moreover, although the per-step cost of PaZO is slightly higher than MeZO—as we transparently report in Figure 2—this is more than offset by its improved convergence rate. In particular, PaZO reduces the total wall-clock time by approximately 10% compared to MeZO, demonstrating that it is efficient in practical settings.

**Algorithm 4** HiZOO

---

**Require:** parameters $\Theta = \{\boldsymbol{\theta}_i \in \mathbb{R}^{d_i}\}$, loss $\mathcal{L} : \mathbb{R}^d \to \mathbb{R}$, step budget $T$, perturbation scale $\mu$, learning rate schedule $\eta_t$, smooth scale $\beta_t$, diagonal Hessian $\boldsymbol{\Sigma}_0$

1: **for** $t = 1, ..., T$ **do**
2:     Sample batch $\mathcal{B} \subset \mathcal{D}$ and random seed $s$
3:     $\ell \leftarrow \mathcal{L}(\boldsymbol{\theta}; \mathcal{B})$
4:     $\boldsymbol{\theta} \leftarrow \text{PerturbParameters}(\boldsymbol{\theta}, \mu, \boldsymbol{\Sigma}_{t-1}^{1/2}, s)$
5:     $\ell_+ \leftarrow \mathcal{L}(\boldsymbol{\theta}; \mathcal{B})$
6:     $\boldsymbol{\theta} \leftarrow \text{PerturbParameters}(\boldsymbol{\theta}, -2\mu, \boldsymbol{\Sigma}_{t-1}^{1/2}, s)$
7:     $\ell_- \leftarrow \mathcal{L}(\boldsymbol{\theta}; \mathcal{B})$
8:     $\boldsymbol{\theta} \leftarrow \text{PerturbParameters}(\boldsymbol{\theta}, \mu, \boldsymbol{\Sigma}_{t-1}^{1/2}, s)$
9:     $\boldsymbol{\Sigma}'_t = \frac{1}{2\mu^2}(\ell_+ + \ell_- - 2\ell)(\boldsymbol{\Sigma}_{t-1}^{-1/2} \mathbf{u}\mathbf{u}^\top \boldsymbol{\Sigma}_{t-1}^{-1/2})$
10:    $\boldsymbol{\Sigma}_t^{-1} = (1 - \alpha_t)\boldsymbol{\Sigma}_{t-1}^{-1} + \beta_t \left|\text{diag}(\boldsymbol{\Sigma}'_t)\right|$
11:    projected\_grad $\leftarrow (\ell_+ - \ell_-) * \boldsymbol{\Sigma}_t^{1/2}/2\mu$
12:    Reset random number generator with seed $s$
13:    **for** $\boldsymbol{\theta}_i \in \Theta$ **do**
14:       Sample $u_i \sim \mathcal{N}(0, \mathbf{I}_{d_i})$
15:       $\boldsymbol{\theta}_i \leftarrow \boldsymbol{\theta}_i - \eta_t * \text{projected\_grad} * \mathbf{u}_i$
16:    **end for**
17: **end for**

---

Table 6: Steps and wall-clock time required to reach 60% accuracy for OPT-1.3B on RTE.

|  | MeZO | HiZOO | PaZO |
|---|---|---|---|
| Steps | 15000 | 10000 | 9000 |
| Wall-clock Time (s) | 3848 | 3785 | 3453 |

## C.7 Ablation Experiments

We conduct experiments to research the influence of $\beta_1$ and $\beta_2$ in the practical version of PaZO in Algorithm 1. Specifically, we use PaZO to fine-tune OPT-1.3B model on SST2. We fix $\beta_1 = 1\text{e}{-}8$ and change $\beta_2$ from 0 to $1\text{e}{-}10$ first. Then we fix $\beta_2 = 1\text{e}{-}8$ and change $\beta_1$ from 0 to $1\text{e}{-}10$. We report the results in Table 7.

The results show that PaZO is sensitive to the smooth hyperparameters $\beta_1$ and $\beta_2$. The excessive choice of $\beta_1$ will seriously affect the convergence, due to the large variance of $\tilde{\mathbf{g}} \circ \tilde{\mathbf{g}}$, while too small choice of $\beta_1$ also affects the performance since it takes little information of $\tilde{\mathbf{g}}$. The choice of $\beta_2$ is relatively lenient, but still needs to be on the same order of the learning rate $\eta$. The best choice of $\beta_2$ may vary across different dataset. In our experiment, we uniformly set $\beta_1$ and $\beta_2$ as $1\text{e}{-}8$ for fair comparison.

We also conduct an ablation study on the effect of the reset period $T_0$ in Table 8. In this experiment, we fine-tune the OPT-1.3B model and evaluate its performance on the SST2 dataset, while varying the value $T_0 \in \{64, 128, 256, 512\}$ and $\infty$ (without resetting). We aim to examine how the frequency of resetting the moving average of the preconditioning matrix influences training stability and final accuracy. The results show that as the number of iterations increases, the error in the Hessian estimation obtained from the zeroth-order oracle tends to accumulate over steps. This accumulated error may grow with the number of iterations. Therefore, when no resetting mechanism is used (as in the case of $\infty$ in the table above), the performance drop when overly old historical accumulated error may mislead the current update direction, ultimately harming final performance. On the other hand, considering that $\boldsymbol{\Sigma}$ is initialized as the identity matrix $\mathbf{I}$, there exists a gap between $\mathbf{I}$ and the Hessian $\mathbf{H}^*$. When the resetting frequency is too small (e.g., 128), the information accumulated in $\boldsymbol{\Sigma}_t$ is insufficient to effectively bridge this gap. The inaccurate estimation similarly leads to worse final model performance. The hyperparameter $T_0$ plays the role of a trade-off between error accumulation and Hessian estimation accuracy. Our experimental results also indicate that the model achieves the best performance when $T_0$ falls within a certain range.

Table 7: Influence of $\beta_1$ and $\beta_2$ in Algorithm 1 for OPT-1.3B on SST2.

| $\beta_1$ $(\beta_2)$ | 1 | 1e−2 | 1e−4 | 1e−6 | 1e−8 | 1e−10 |
|---|---|---|---|---|---|---|
| fixed $\beta_1 = 1e{-}8$ | NaN | NaN | NaN | 88.9 ($\pm$0.3) | 89.0 ($\pm$0.2) | 89.0 ($\pm$0.2) |
| fixed $\beta_2 = 1e{-}8$ | NaN | NaN | NaN | NaN | 89.0 ($\pm$0.1) | 88.9 ($\pm$0.2) |

Table 8: Influence of $T_0$ in Algorithm 1 for OPT-1.3B on SST2.

| $T_0$ | 64 | 128 | 256 | 512 | $\infty$ |
|---|---|---|---|---|---|
| | 88.5 | 88.8 | 89.0 | 88.3 | 87.7 |

## D  Auxiliary Lemmas

**Lemma D.1.** *For a matrix $\mathbf{A} \in \mathbb{R}^{m\times m}$ and vectors $\boldsymbol{u} \in \mathbb{R}^m, \boldsymbol{v} \in \mathbb{R}^m$, if $\mathbf{A} + \boldsymbol{u}\boldsymbol{v}^\top$ is invertible, we have*

$$
\begin{aligned}
\left(\mathbf{A} + \boldsymbol{u}\boldsymbol{v}^\top\right)^{-1} =& \mathbf{A}^\dagger - (\boldsymbol{v}_2^\top \boldsymbol{v}_2)^{-1}\boldsymbol{v}_2\boldsymbol{v}_1^\top \mathbf{A}^\dagger - (\boldsymbol{u}_2^\top \boldsymbol{u}_2)^{-1}\mathbf{A}^\dagger \boldsymbol{u}_1\boldsymbol{u}_2^\top \\
&+ (\boldsymbol{v}_2^\top \boldsymbol{v}_2)^{-1}(\boldsymbol{u}_2^\top \boldsymbol{u}_2)^{-1}\left(1 + \boldsymbol{v}_1^\top \mathbf{A}^\dagger \boldsymbol{u}_1\right)\boldsymbol{v}_2\boldsymbol{u}_2^\top.
\end{aligned} \tag{49}
$$

*where $\boldsymbol{u} = \boldsymbol{u}_1 + \boldsymbol{u}_2$ with $\boldsymbol{u}_1 \in \mathrm{col}(\mathbf{A})$, $\boldsymbol{u}_2 \perp \mathrm{col}(\mathbf{A})$ and $\boldsymbol{v} = \boldsymbol{v}_1 + \boldsymbol{v}_2$ with $\boldsymbol{v}_1 \in \mathrm{col}(\mathbf{A}^\top)$, $\boldsymbol{v}_2 \perp \mathrm{col}(\mathbf{A}^\top)$. In addition, we can obtain that $(\mathbf{A} + \lambda\boldsymbol{u}\boldsymbol{v}^\top)^{-1}\boldsymbol{u} = \lambda^{-1}(\boldsymbol{v}_2^\top \boldsymbol{v}_2)^{-1}\boldsymbol{v}_2$ for any $\lambda > 0$.*

**Lemma D.2.** *We assume matrix $\mathbf{M} = \hat{\mathbf{M}} \otimes \mathbf{I}_d - \gamma(\boldsymbol{c}_1\boldsymbol{c}_2^\top) \otimes \breve{\mathbf{M}}$ where $\hat{\mathbf{M}} \in \mathbb{R}^{m\times m}$, $\boldsymbol{c}_1 \in \mathbb{R}^m$, $\boldsymbol{c}_2 \in \mathbb{R}^m$ and $\breve{\mathbf{M}} \in \mathbb{R}^{d\times d}$ is symmetric, and matrix $\bar{\mathbf{M}} \in \mathbb{R}^{d\times d}$ is positive semi-definite. Given positive semi-definite matrix $\mathbf{B}$, we suppose that the max singular value of $\mathbf{M}$ is strictly smaller than 1, and matrices $\mathbf{B}$ and $\breve{\mathbf{M}}$ share a common set of orthonormal eigenvectors. Specifically, their spectral decompositions can be expressed as:*

$$
\mathbf{B} = \mathbf{P}\widetilde{\boldsymbol{\Lambda}}\mathbf{P}^{-1}, \quad \breve{\mathbf{M}} = \mathbf{P}\boldsymbol{\Lambda}\mathbf{P}^{-1},
$$

*where $\mathbf{P} \in \mathbb{R}^{d\times d}$ is an orthogonal matrix, and $\widetilde{\boldsymbol{\Lambda}}$ and $\boldsymbol{\Lambda}$ are real diagonal matrices. Then we obtain*

$$
\mathrm{tr}\left((\mathbf{I}_m \otimes \mathbf{B})\,\mathbf{M}^t\left((\boldsymbol{d}\boldsymbol{d}^\top) \otimes \bar{\mathbf{M}}\right)(\mathbf{M}^\top)^t\right) \leq \bar{C}_{\mathbf{M}}\|\mathbf{B}\|_2\|\boldsymbol{d}\|_2^2(1 - \gamma\mu_{\mathbf{M}})^{2t}\mathrm{tr}(\bar{\mathbf{M}}), \tag{50}
$$

*where $\bar{C}_{\mathbf{M}}$ and $\mu_{\mathbf{M}} > 0$ are two positive constants depend on $\mathbf{M}$.*

*Proof.* We prove estimation Eq. (50) at first. There exists an orthonormal matrix $\mathbf{P} \in \mathbb{R}^{d\times d}$ such that $\breve{\mathbf{M}} = \mathbf{P}\boldsymbol{\Lambda}\mathbf{P}^{-1}$ where $\boldsymbol{\Lambda} = \mathrm{diag}\{\lambda_1, \cdots, \lambda_d\}$. Therefore, we have that

$$
\mathbf{M} = (\mathbf{I}_m \otimes \mathbf{P})\,\mathbf{Q}^\top \mathrm{diag}\left\{\hat{\mathbf{M}} - \gamma\lambda_1\boldsymbol{c}_1\boldsymbol{c}_2^\top, \cdots, \hat{\mathbf{M}} - \gamma\lambda_d\boldsymbol{c}_1\boldsymbol{c}_2^\top\right\}\mathbf{Q}\left(\mathbf{I}_m \otimes \mathbf{P}^{-1}\right), \tag{51}
$$

where $\mathbf{Q} \in \mathbb{R}^{md\times md}$ is an orthogonal matrix. For simplicity, we denote $\hat{\mathbf{D}} := \mathrm{diag}\{\hat{\mathbf{M}} - \gamma\lambda_1\boldsymbol{c}_1\boldsymbol{c}_2^\top, \cdots, \hat{\mathbf{M}} - \gamma\lambda_d\boldsymbol{c}_1\boldsymbol{c}_2^\top\}$ and $\hat{\mathbf{D}}_i = \hat{\mathbf{M}} - \gamma\lambda_i\boldsymbol{c}_1\boldsymbol{c}_2^\top$. Therefore, we can obtain

$$
\begin{aligned}
\mathbf{M}^t\left((\boldsymbol{d}\boldsymbol{d}^\top) \otimes \bar{\mathbf{M}}\right)(\mathbf{M}^\top)^t \overset{(a)}{=}& (\mathbf{I}_m \otimes \mathbf{P})\,\mathbf{Q}^\top \hat{\mathbf{D}}^t\left(\mathbf{P}^{-1}\bar{\mathbf{M}}\mathbf{P}^{-\top} \otimes (\boldsymbol{d}\boldsymbol{d}^\top)\right)(\hat{\mathbf{D}}^\top)^t\mathbf{Q}\left(\mathbf{I}_m \otimes \mathbf{P}^\top\right) \\
=& (\mathbf{I}_m \otimes \mathbf{P})\,\mathbf{Q}^\top \mathbf{A}\mathbf{Q}\left(\mathbf{I}_m \otimes \mathbf{P}^\top\right),
\end{aligned} \tag{52}
$$

where

$$
\mathbf{A} = \begin{bmatrix} \mathbf{A}_{11} & \cdots & \mathbf{A}_{1d} \\ \vdots & \ddots & \vdots \\ \mathbf{A}_{d1} & \cdots & \mathbf{A}_{dd} \end{bmatrix},
$$

with $\mathbf{A}_{ij} \in \mathbb{R}^{m\times m}$ satisfies $\mathbf{A}_{ij} = (\mathbf{P}^{-1}\bar{\mathbf{M}}\mathbf{P}^{-\top})_{ij}\hat{\mathbf{D}}_i(\boldsymbol{d}\boldsymbol{d}^\top)\hat{\mathbf{D}}_j^\top$ for any $i, j \in [1:d]$, (a) is derived from the fact that

$$
\left(\mathbf{I}_m \otimes \mathbf{P}^{-1}\right)\left((\boldsymbol{d}\boldsymbol{d}^\top) \otimes \bar{\mathbf{M}}\right)\left(\mathbf{I}_m \otimes \mathbf{P}^{-\top}\right) = (\boldsymbol{d}\boldsymbol{d}^\top) \otimes \mathbf{P}^{-1}\bar{\mathbf{M}}\mathbf{P}^{-\top},
$$

and

$$
\mathbf{Q}\left((\boldsymbol{d}\boldsymbol{d}^\top) \otimes \mathbf{P}^{-1}\bar{\mathbf{M}}\mathbf{P}^{-\top}\right)\mathbf{Q}^\top = \mathbf{P}^{-1}\bar{\mathbf{M}}\mathbf{P}^{-\top} \otimes (\boldsymbol{d}\boldsymbol{d}^\top).
$$

According to the property of $\mathbf{Q}$, we have

$$\text{tr}\left((\mathbf{I}_m \otimes \mathbf{B})(\mathbf{I}_m \otimes \mathbf{P})\mathbf{Q}^\top \mathbf{A}\mathbf{Q}(\mathbf{I}_m \otimes \mathbf{P}^\top)\right) = \text{tr}\left(\mathbf{B}\mathbf{P}\hat{\mathbf{A}}\mathbf{P}^\top\right), \tag{53}$$

where $\hat{\mathbf{A}} \in \mathbb{R}^{d\times d}$ satisfies $\hat{\mathbf{A}}_{ij} = (\mathbf{P}^{-1}\bar{\mathbf{M}}\mathbf{P}^{-\top})_{ij}\langle\hat{\mathbf{D}}_i^t\boldsymbol{d}, \hat{\mathbf{D}}_j^t\boldsymbol{d}\rangle$ for any $i, j \in [1:d]$. Since $\mathbf{P}^\top\mathbf{B}\mathbf{P} = \widetilde{\boldsymbol{\Lambda}}$, we derive that

$$\text{tr}\left(\mathbf{B}\mathbf{P}\hat{\mathbf{A}}\mathbf{P}^\top\right) \leq \|\mathbf{P}^\top\mathbf{B}\mathbf{P}\|_2\text{tr}(\hat{\mathbf{A}}) \overset{(b)}{\leq} C_{\mathbf{M}}\|\mathbf{P}\|_2^4\|\mathbf{P}^{-1}\|_2^4\|\boldsymbol{d}\|_2^2\|\mathbf{B}\|_2(1-\gamma\mu_{\mathbf{M}})^{2t}\text{tr}(\bar{\mathbf{M}}) \tag{54}$$

for any positive semi-definite matrix $\mathbf{B} \in \mathbb{R}^{d\times d}$, where (b) follows from the fact that

$$\hat{\mathbf{D}}_i^t\boldsymbol{d} = \left(\mathbf{I}_m \otimes \mathbf{P}^{-1}\right)\mathbf{Q}\mathbf{M}^t\left(\mathbf{I}_m \otimes \mathbf{P}\right)\mathbf{Q}^\top e_i \otimes \boldsymbol{d}, \tag{55}$$

where $e_i \in \mathbb{R}^d$ denotes a vector whose element at the $i$-th position is equal to 1, while the elements in all remaining positions are equal to 0, and the assumption that the max singular value of $\mathbf{M}$ is strictly smaller than 1. $\qquad\square$

**Lemma D.3.** *We assume matrix* $\mathbf{M} = \hat{\mathbf{M}} \otimes \mathbf{I}_d + (\boldsymbol{c}_1\boldsymbol{c}_2^\top) \otimes \check{\mathbf{M}}$ *where* $\hat{\mathbf{M}} \in \mathbb{R}^{m\times m}$, $\boldsymbol{c}_1 \in \mathbb{R}^m$, $\boldsymbol{c}_2 \in \mathbb{R}^m$ *and* $\check{\mathbf{M}} \in \mathbb{R}^{d\times d}$, *and matrix* $\bar{\mathbf{M}} \in \mathbb{R}^{d\times d}$ *is symmetric. If* $\check{\mathbf{M}}$ *is also symmetric, and both* $\mathbf{M}$ *and* $\check{\mathbf{M}}$ *are invertible, we have*

$$\mathbf{M}^{-1}\left((\boldsymbol{c}_1\boldsymbol{c}_1^\top) \otimes \bar{\mathbf{M}}\right)\mathbf{M}^{-\top} = \|\boldsymbol{c}_{22}\|_2^{-4}\left(\boldsymbol{c}_{22}\boldsymbol{c}_{22}^\top\right) \otimes \left(\check{\mathbf{M}}^{-1}\bar{\mathbf{M}}\check{\mathbf{M}}^{-\top}\right), \tag{56}$$

*where* $\boldsymbol{c}_2 = \boldsymbol{c}_{21} + \boldsymbol{c}_{22}$, $\boldsymbol{c}_{21} \in \text{col}(\hat{\mathbf{M}}^\top)$ *and* $\boldsymbol{c}_{22} \perp \text{col}(\hat{\mathbf{M}}^\top)$.

*Proof.* Similarly, there exists an invertible matrix $\mathbf{P} \in \mathbb{R}^{d\times d}$ such that $\check{\mathbf{M}} = \mathbf{P}\boldsymbol{\Lambda}\mathbf{P}^{-1}$ where $\boldsymbol{\Lambda} = \text{diag}\{\lambda_1, \cdots, \lambda_d\}$. Therefore, we have that

$$\mathbf{M} = (\mathbf{I}_m \otimes \mathbf{P})\mathbf{Q}^\top \text{diag}\left\{\hat{\mathbf{M}} + \lambda_1\boldsymbol{c}_1\boldsymbol{c}_2^\top, \cdots, \hat{\mathbf{M}} + \lambda_d\boldsymbol{c}_1\boldsymbol{c}_2^\top\right\}\mathbf{Q}\left(\mathbf{I}_m \otimes \mathbf{P}^{-1}\right), \tag{57}$$

where $\mathbf{Q} \in \mathbb{R}^{md\times md}$ is an orthogonal matrix. For simplicity, we denote $\hat{\mathbf{D}} := \text{diag}\{\hat{\mathbf{M}} + \lambda_1\boldsymbol{c}_1\boldsymbol{c}_2^\top, \cdots, \hat{\mathbf{M}} + \lambda_d\boldsymbol{c}_1\boldsymbol{c}_2^\top\}$. Furthermore, we can obtain that

$$\begin{aligned}
&\mathbf{M}^{-1}\left((\boldsymbol{c}_1\boldsymbol{c}_1^\top) \otimes \bar{\mathbf{M}}\right)\mathbf{M}^{-\top}\\
&= (\mathbf{I}_m \otimes \mathbf{P})\mathbf{Q}^\top\hat{\mathbf{D}}^{-1}\mathbf{Q}\left((\boldsymbol{c}_1\boldsymbol{c}_1^\top) \otimes (\mathbf{P}^{-1}\bar{\mathbf{M}}\mathbf{P}^{-\top})\right)\mathbf{Q}^\top\hat{\mathbf{D}}^{-\top}\left(\mathbf{I}_m \otimes \mathbf{P}^\top\right).
\end{aligned} \tag{58}$$

Since $\mathbf{Q}$ is, in fact, a coordinate transformation matrix, we have

$$\mathbf{Q}\left((\boldsymbol{c}_1\boldsymbol{c}_1^\top) \otimes (\mathbf{P}^{-1}\bar{\mathbf{M}}\mathbf{P}^{-\top})\right)\mathbf{Q}^\top = (\mathbf{P}^{-1}\bar{\mathbf{M}}\mathbf{P}^{-\top}) \otimes (\boldsymbol{c}_1\boldsymbol{c}_1^\top).$$

Therefore, we can derive that

$$\hat{\mathbf{D}}^{-1}\left((\mathbf{P}^{-1}\bar{\mathbf{M}}\mathbf{P}^{-\top}) \otimes (\boldsymbol{c}_1\boldsymbol{c}_1^\top)\right)\hat{\mathbf{D}}^{-\top} = \begin{bmatrix} \mathbf{A}_{11} & \cdots & \mathbf{A}_{1d} \\ \vdots & \ddots & \vdots \\ \mathbf{A}_{d1} & \cdots & \mathbf{A}_{dd} \end{bmatrix}, \tag{59}$$

by using Lemma D.1 where $\mathbf{A}_{ij} \in \mathbb{R}^{m\times m}$ and

$$\begin{aligned}
\mathbf{A}_{ij} &= \left\{\mathbf{P}^{-1}\bar{\mathbf{M}}\mathbf{P}^{-\top}\right\}_{ij}\left(\hat{\mathbf{M}} + \lambda_i\boldsymbol{c}_1\boldsymbol{c}_2^\top\right)^{-1}\left(\boldsymbol{c}_1\boldsymbol{c}_1^\top\right)\left(\hat{\mathbf{M}} + \lambda_j\boldsymbol{c}_1\boldsymbol{c}_2^\top\right)^{-\top}\\
&= \|\boldsymbol{c}_{22}\|_2^{-4}\lambda_i^{-1}\lambda_j^{-1}\left\{\mathbf{P}^{-1}\bar{\mathbf{M}}\mathbf{P}^{-\top}\right\}_{ij}\boldsymbol{c}_{22}\boldsymbol{c}_{22}^\top,
\end{aligned} \tag{60}$$

According to the property of $\mathbf{Q}$, we obtain

$$\mathbf{Q}^\top\hat{\mathbf{D}}^{-1}\left(\left(\mathbf{P}^{-1}\bar{\mathbf{M}}\mathbf{P}^{-\top}\right) \otimes \left(\boldsymbol{c}_1\boldsymbol{c}_1^\top\right)\right)\hat{\mathbf{D}}^{-\top}\mathbf{Q} = \|\boldsymbol{c}_{22}\|_2^{-4}\left(\boldsymbol{c}_{22}\boldsymbol{c}_{22}^\top\right) \otimes \left(\boldsymbol{\Lambda}^{-1}\mathbf{P}^{-1}\bar{\mathbf{M}}\mathbf{P}^{-\top}\boldsymbol{\Lambda}^{-1}\right). \tag{61}$$

Combining Eq. (58) and Eq. (61), we complete the proof. $\qquad\square$

