# OpenReview forum: "PaZO: Preconditioned Accelerated Zeroth-Order Optimization for Fine-Tuning LLMs"
_NeurIPS.cc/2025/Conference — NeurIPS 2025 poster_

### Official Review · Reviewer_HFxg · 2025-06-10

**Clarity:** 3
**Significance:** 2
**Originality:** 2
**Rating:** 4
**Confidence:** 2

**Summary:**

This paper investigates the role of preconditioning in zeroth-order optimization for memory-efficient fine-tuning of large language models and introduces PaZO. The authors define a general PSPSA estimator that perturbs parameters with preconditioner $H^{-\alpha}$ and prove that only choosing $\alpha = 1/2$ yields the optimal convergence rate. The authors then proposes a practical algorithm named PaZO that utilizes diagonal Hessian estimator via three forward-pass perturbations. Empirical evaluation across multiple downstream tasks and models demostrate the effectiveness of the proposed algorithm.

**Questions:**

1. While estimating the diagonal Hessian matrix, 1) why do we need to incorporate $diag( \tilde{g} \circ  \tilde{g})$ as a correction item to integrate local first-order estimated information into $\Sigma_{t-1}$ through moving average mechanism? 2) why $\tilde{\Sigma}$ is computed through $((1-\beta_1) \Sigma_{t-1}^2 + \beta_1 \cdot diag( \tilde{g} \circ  \tilde{g}))^{1/2}$, not $(1-\beta_1) \Sigma_{t-1} + \beta_1 diag( \tilde{g} \circ  \tilde{g})$. 3) In theory, such correction is not applied in estimating the term $\mathcal{I}$ while computing $\Sigma_t$, which makes Equation. (14) no longer an unbiased estimation to the diagonal Hessian. 4) In practice, the authors conducted an ablation experiment on studying the influence of parameter choice of $\beta_1$, why too small choice of $\beta_1$ (including 0) result  in NaN. From my understanding, setting $\beta_1$ to be zero will make PaZO similar as that in HiZOO. In this ablation study, I can't figure out the real effect of incorporating the estimated gradient as a correction.

2. Apart from the correction term, another difference between PaZO and HiZOO is the periodical reset mechanism for diagonal Hessian estimation. However, there is no ablation study design for this mechanism. Why this this step is necessary?

**Ethical Concerns:**

["NO or VERY MINOR ethics concerns only"]

**Final Justification:**

I have no additional questions and thank you for the detailed response, which has resolved my previous concerns.

**Limitations:**

yes.

**Quality:**

2

**Strengths And Weaknesses:**

Strengths:
1. The authors give the answer to an important question regarding the necessity of preconditioning in ZO and identify the optimal order in preconditioner.

2. The three-point perturbation scheme combined with gradient-based correction and periodic reset is a novel, lightweight way to incorporate second-order information without prohibitive memory.

3. Extensive empirical evaluation on both masked (RoBERTa-large) and generative (OPT-1.3B) models across a diverse suite of NLP tasks, demonstrating robustness of PaZO.

Weakness:
1. There are several formulation mistakes in estimating the diagonal Hessian. The RHS of Equation (11) should be $diag(H)$. The LHS of Equation (13) should be $\ell_{+} + \ell_{-} - 2\ell$.

2. Some hyperparameter choices (e.g., reset period $T_0$ schedule) are introduced without intuitive justification.

---

> ### Author Rebuttal · Authors · 2025-07-31
>
> # Response to Reviewer HFxg
>
> We sincerely thank the reviewer HFxg for their thorough insightful feedback and constructive suggestions. We deeply appreciate the effort you dedicated to assessing our work.  We address the main concerns raised by the reviewer HFxg point by point.
>
> ## 1. Correct some typos (regarding Weakness 1 and Question 1)
> We thank the reviewer for pointing out the typos. We have carefully re-checked all symbols, equations, and tables throughout the paper. Below, we list all the corrections we identified and will incorporate them in the camera-ready version:
> | Location | Original Text                        | Corrected Text                                                              |
> | -------- | ------------------------------------ | ---------------------------------------------------------------------------------- |
> | Equation (11) | $\mathbb{E}\left[\underbrace{\frac{1}{2}u^\top \mathbf{\Sigma}^{-\frac{1}{2}} \mathbf{H} \mathbf{\Sigma}^{-\frac{1}{2}} u}_{\mathcal{I}} \cdot  \mathbf{\Sigma}\left(diag(uu^\top)  - \mathbf{I}\right)\right] = \mathbf{H}$ | $\mathbb{E}\left[\underbrace{\frac{1}{2}u^\top \mathbf{\Sigma}^{-\frac{1}{2}} \mathbf{H} \mathbf{\Sigma}^{-\frac{1}{2}} u}_{\mathcal{I}} \cdot  \mathbf{\Sigma}\left(diag(uu^\top)  - \mathbf{I}\right)\right] = diag(\mathbf{H})$ |
> | Equation (13) | $\frac{\ell_+ +\ell_- -\ell}{\mu^2} = \mathcal{I} + \mathcal{O}(\mu)$ | $\frac{\ell_+ +\ell_- -2\ell}{\mu^2} = \mathcal{I} + \mathcal{O}(\mu)$ |
> | "Adding information from ZO gardient" in Algorithm 1 and below line 223 | $\mathbf{\tilde\Sigma} =\left( (1-\beta_1)\mathbf{\Sigma}_{t-1}^2 + \beta_1 \cdot diag(\tilde{g} \circ \tilde{g}) \right)^{1/2}$ | $\mathbf{\tilde\Sigma} =\left( (1-\beta_1)\mathbf{\Sigma}_{t-1}^2 + \beta_1 \cdot diag^2(\tilde{g} \circ \tilde{g}) \right)^{1/2}$ |
> | the first row, second column in Table 6 | 0 | 1|
>
>
> We acknowledge that some of these typos may have caused confusion and may be directly responsible for the series of concerns you raised in **Question 1**. We sincerely apologize for this and now proceed to clarify your questions in detail. Regrettably, the rebuttal process does not allow us to submit a revised manuscript. Nevertheless, we will make sure to fix all the aforementioned issues and include a detailed discussion in response to your concerns in the final camera-ready version.
>
> ## 2. The necessity of the correction term in estimating the diagonal Hessian (regarding Question 1, (1))
>
> In summary, this correction term can reduce the instability of training caused by the outliers from zeroth-order oracle with randomness. Specifically, due to the order of preconditioner is $-1/2$ ($\alpha=1/2$), too small value in the diagonal estimated Hessian may cause numerical instability (NaN) in training process. The correction term encourages numerical alignment between gradient magnitude and adaptive curvature scaling, similar in spirit to adaptive methods like Adam [1] and Adafactor [2]. To illustrate the issue more intuitively, we offer some insights through a non-strict analysis below.
>
>
> In practice, we first obtain $\ell_+$, $\ell_-$ and $\ell$ through three forward pass. We obtain three numerical finite values and then $\mathcal{I}$, which, although dependent on $u$, are treated as a fixed scalar in our experiments. Thus if we do not have the correction term, we have $$\mathbb{E}\left[\mathbf{\Sigma}\left(diag(uu^\top)  - \mathbf{I}\right)\right] \approx 0,$$
>
> which means that in practice, we may obtain some extremely small values close to zero, causing the numerical outliers (NaN) considering the order $-1/2$. When adding the correction term, similarly ignoring the scaler, we have $$\mathbb{E}\left[\mathbf{\Sigma}\left(\mathbf{I}+diag(uu^\top)\right)\left(diag(uu^\top)  - \mathbf{I}\right)\right] \approx 2\mathbf{\Sigma},$$
>
> which means that $\Sigma_{t}$ is numerical remains numerically consistent with $\Sigma_{t-1}$. Thus we obtain a more stable update of $\Sigma$, which helps ensure the stability of training in our experiments. Moreover, we control the scale of correction term via $\beta_1$, which means that we can achieve numerical stability with only a small-scale correction term.
>
>
> ## 3. Why we compute $\mathbf{\tilde\Sigma} =\left( (1-\beta_1)\mathbf{\Sigma}_{t-1}^2 + \beta_1 \cdot diag^2(\tilde{g} \circ \tilde{g}) \right)^{1/2}$ (regarding Question 1, (2) and (3))
>
> We sincerely apologize for the typo that we forgot the order of $diag$. As you mentioned, it is common to use linear-weighted average form $\mathbf{\tilde\Sigma} = (1-\beta_1)\mathbf{\Sigma}_{t-1} + \beta_1 \cdot diag(\tilde{g} \circ \tilde{g}) $. However, in zeroth-order optimization, we observe that root-sum-square average $\sqrt{A^2+B^2}$ performs better compared with $A+B$. First, $\sqrt{A^2+B^2}$ is bounded by $A+B$ as $\frac{A+B}{\sqrt{2}} \leq \sqrt{A^2+B^2} \leq A+B$, which means that the two are close in order of magnitude. Second,  the square-root form acts as a soft maximum. When $A$ is close to $B$, $\sqrt{A^2+B^2}$ are within the same order of magnitude, differing only by a constant factor. When $A \gg B$ or $B \gg A$, $\sqrt{A^2 + B^2}$ behaves like a softmax operator, approximating the larger value.
>
>
>
>  As you mentioned, applying such a correction causes bias in Equation (14). However, in practice, by the softmax operator $\sqrt{A^2+B^2}$ and the choice of $\beta_1$, we find that the deviation from unbiasedness is negligible empirically. On the contrary, such an almost negligible term enhances the robustness of the estimation against outliers, which is critical for the stability of zeroth-order optimization.
> We believe that for theory-inspired algorithms, making empirically motivated adjustments when applying them to practical tasks is both reasonable and necessary, so we named the algorithm "practical form" in Algorithm 1.
>
> ## 4. The choice of $\beta_1$ in the ablation study (regarding Question 1, (4))
>
> At first we clarify that even though setting $\beta_1 =0$, PaZO is **not similar** to HiZOO due to the different order of estimated diagonal Hessian (different setting of $\alpha$). One of the most important contributions of this work is to demonstrate that the ideal choice of $\alpha$ is $-1/2$ instead of $0$ for MeZO and $1/2$ for HiZOO, no matter what $\beta_1$ we select. After correcting the typo in Table 6, we state the results of ablation study as below for OPT-1.3B on SST2.
>
> | $\beta_1$ | 1 | 1e-2 |1e-4 |1e-6 | 1e-8 |1e-10 | 0 |
> | -- | -- | -- | -- | -- | -- | -- | -- |
> | setting $\beta_2$ = 1e-8 | NaN | NaN | NaN| NaN | 89.0 | 88.9 | NaN |
>
> The results show that when $\beta_1$ is set too large (e.g., 1, 1e-2), the optimization diverges and results in NaNs, because the proportion of correction term is too large to introduce non-negligible bias. When $\beta_1$ is too small (e.g., 0 or approaching 0), the correction term fails to take effect, which may lead to numerical instability during training and result in NaNs. Only in a proper range (around $1e\text{-}8$ to $1e\text{-}10$) does the algorithm yield reasonable performance. This behavior confirms our discussion earlier about the necessity of correction term, as we state in "2. The necessity of correction term". The use of a carefully chosen $\beta_1$ serves as a regularization mechanism, stabilizing the training process.
>
> ## 5. Ablation study about $T_0$ (regarding Weakness 2 and Question 2)
>
> We briefly explain why we reset the $\Sigma$ frquently from line 228 to line 230 in the main body of our paper.  This idea has been considered in periodic error feedback resetting used in optimization with communication compression [4].
>
> As a supplement, we an ablation study on the effect of the reset period
> $T_0$.
> In this experiment, we fine-tune the OPT-1.3B model and evaluate its performance on the SST2 dataset, while varying the value $T_0 \in\{64,128,256, 512 \} $ and $\infty$ (without resetting). The results are summarized in the table below:
>
> | $T_0$| 64| 128 |256 | 512 | $\infty$ |
> | -- | -- |--| -- | -- | -- |
> |SST2  |88.5|88.8 |89.0 | 88.3| 87.7|
>
>
> The reason behind this design and observation is as follows. As the number of iterations increases, the error in the Hessian estimation obtained from the zeroth-order oracle tends to accumulate over steps. This accumulated error may grow with the number of iterations. Therefore, when no resetting mechanism is used (as in the case of $\infty$ in the table above), the performance drops. On the other hand,  considering that $\Sigma$ is initialized as the identity matrix $I$, there exists a gap between $I$ and the Hessian $H^*$. When the resetting frequency is too small (e.g., 128), the information accumulated in $\Sigma_t$ is insufficient to effectively bridge this gap. The hyperparameter $T_0$ plays the role of a trade-off between error accumulation and Hessian estimation accuracy. Our experimental results also indicate that the model achieves the best performance when $T_0$ falls within a certain range. We will discuss more about the ablation study on $T_0$ in the camera-ready version.
>
> We sincerely thank the reviewer HFxg for the constructive and insightful feedback. We greatly appreciate the opportunity to improve our work through this review process, and we would be truly grateful if the reviewer HFxg could consider a positive adjustment to the scores.
>
>
> ## Reference
> [1] Kingma D P, Ba J. Adam: A method for stochastic optimization[J]. arXiv preprint arXiv:1412.6980, 2014.
>
> [2] Shazeer N, Stern M. Adafactor: Adaptive learning rates with sublinear memory cost[C]//International Conference on Machine Learning. PMLR, 2018: 4596-4604.
>
> [3] Chen X, Liu S, Xu K, et al. Zo-adamm: Zeroth-order adaptive momentum method for black-box optimization[J]. Advances in neural information processing systems, 2019, 32.
>
> [4] Xie X, Lin Z, Toh K C, et al. Loco: Low-bit communication adaptor for large-scale model training[J]. IEEE Transactions on Pattern Analysis and Machine Intelligence, 2025.

---

> ### Comment · Reviewer_HFxg · 2025-08-03
>
> Thank you for the detailed response, which has resolved my previous concerns.
>
> Just two more minor points in your response:
>
> 1. Regarding the definition of  $\mathcal{I}$, I think it should be $\mathbb{E}\left[ \frac{1}{2} \underbrace{u^\top \mathbf{\Sigma}^{-\frac{1}{2}} \mathbf{H} \mathbf{\Sigma}^{-\frac{1}{2}} u}_{\mathcal{I}} \cdot  \mathbf{\Sigma}\left(diag(uu^\top)  - \mathbf{I}\right)\right] = diag(\mathbf{H})$?
>
>
> 2. the statement "... the ideal choice of $\alpha$ is $-1/2$ instead of $0$ for MeZO and $1/2$ for HiZOO" requires clarification. In fact, the choice of $\alpha$ both in your algorithm and in HiZOO are $1/2$. The confusion likely arises from the way the moving average of the diagonal Hessian (inverse) is updated. In HiZOO, it is written as $\Sigma_t^{-1} = \cdots$, which is moving average of diagonal Hessian *inverse*, whereas in your algorithm it is expressed as $\Sigma_t = \cdots$, which is moving average of diagonal Hessian.
>
> For the second point, I think multiple positions in the paper need to be fixed.

---

> > ### Author Response · Authors · 2025-08-04
> >
> > We sincerely thank the reviewer for carefully reading our previous response and pointing out some minors. We are glad to hear that our clarifications have addressed your concerns. We hope that the following additional clarifications can help address the two minor points you raised.
> >
> > First we directly show the choice of $\alpha$ in the table below.
> > |  | MeZO | HiZOO | PaZO |
> > | -- | -- | -- | -- |
> > | $\alpha$ | 0  | -1/2 | 1/2 |
> >
> > In fact, **we have repeatedly emphasized in the main paper that the choice of $\alpha$ is one of our key contributions. Specifically, we adopt the theoretically ideal $\alpha = 1/2$ to achieve faster convergence, as discussed in several parts of the main body of paper.** For example, in line 197 in the main text, we mentioned that "for general problems, selecting $\alpha=0$ alone induces a slower convergence rate than the optimal choice $\alpha=1/2$". In line 201, we mentioned that "By selecting $\alpha=1/2$, PaZO achieves the fastest convergence rate $\tilde{\mathcal{O}}\left(d^2/T\right)$ compared with MeZO and HiZOO for general functions, providing a reasonable answer to question \textit{B}".  In line 259, we compared the choice of $\alpha$ as "PaZO achieves SOTA performance compared to other zeroth-order optimizer baselines including MeZO ($\alpha=0$) and HiZOO ($\alpha=-1/2$)". We sincerely apologize for the typos in our first response. However, **the choice of $\alpha$ in HiZOO is $-1/2$ but not $1/2$ as you mentioned in minor 2.** You can see HiZOO algorithm in page 24 in the supplement, where HiZOO chooses $\Sigma^{1/2}$ as the preconditioner, which means that $\alpha=-1/2$ in HiZOO.
> >
> > Then considering the moving average, HiZOO takes the moving average on the inverse of estimated Hessian. This idea is motivated by practical considerations, but it lacks theoretical support. In particular, for matrices, $(A + B)^{-1} \neq A^{-1} + B^{-1}$ generally holds. Therefore, instead of accumulating inverse estimates, we adopt a more principled and reasonable approach. We maintain a moving average of the estimated diagonal Hessian matrix, and only compute the inverse when needed. Since the estimates are diagonal, this inversion step incurs negligible computational overhead.
> >
> >
> >
> > Overall, we believe that the theoretical motivation behind our method has been clearly articulated in the main paper, particularly through Theorems 3.5 and 3.8 and their proofs, which explicitly demonstrate how the choice of $\alpha$ influences the optimization dynamics and why the theoretically ideal value is $\alpha = 1/2$. We are also very pleased that the  clarification provided in our first response has resolved your concerns about the practical form of our method and hyperparameter settings. Although there were indeed a few typos (and we sincerely thank you for pointing them out including the fact that in minor 1, $1/2$ should not be included in $\mathcal{I}$), they do not affect the overall understanding of our work and contributions. We will correct all such typos in the camera-ready version to ensure clarity of presentation. We hope our discussions have adequately addressed your concerns. We would greatly appreciate your consideration of an increasing score.

---

> ### Comment · Reviewer_HFxg · 2025-08-04
>
> Thank you for your further clarification. The main point is that $\Sigma_t$ in PaZO serves as an estimate of the diagonal Hessian, whereas $\Sigma_t$ in HiZOO corresponds to an estimate of the inverse diagonal Hessian (as described in the begining of section 3.2 of HiZOO paper and the pseudocode of HiZOO). Given the definition of the preconditioner $H^{-2\alpha}$, where $H$ denotes the Hessian matrix, I still believe it is not appropriate to describe the choice of $\alpha$ in HiZOO as $-1/2$.
>
> I have also checked the code implementation of HiZOO, which is a little bit different from the pseudocode description:
>
>
>     def efficient_Hessian_perturb_parameters(self, model: nn.Module, random_seed, Hessian_matrix=None, scaling_factor=1):
>         torch.manual_seed(random_seed)
>         for name, param in self.named_parameters_to_optim:
>             z = torch.normal(mean=0, std=1, size=param.data.size(), device=param.data.device, dtype=param.data.dtype)
>             param.data = param.data + scaling_factor / torch.sqrt(Hessian_matrix[name]) * z * self.args.zo_eps
>         return model
>
>
>     def zo_Hessian_step_update(self, model, inputs, zo_learning_rate, Hessian_smooth):
>
>         self.named_parameters_to_optim = []
>         for name, param in model.named_parameters():
>             if param.requires_grad:
>                 self.named_parameters_to_optim.append((name, param))
>
>         random_seed = np.random.randint(1000000000)
>         with torch.no_grad():
>             loss_original = self.zo_forward(model, inputs)
>
>             # first function evaluation
>             model = self.efficient_Hessian_perturb_parameters(model, random_seed, self.Hessian_matrix, scaling_factor=1)
>             loss1 = self.zo_forward(model, inputs)
>
>             # second function evaluation
>             model = self.efficient_Hessian_perturb_parameters(model, random_seed, self.Hessian_matrix, scaling_factor=-2)
>             loss2 = self.zo_forward(model, inputs)
>
>             model = self.efficient_Hessian_perturb_parameters(model, random_seed, self.Hessian_matrix, scaling_factor=1)
>
>             torch.manual_seed(random_seed)
>             for name, param in self.named_parameters_to_optim:
>                 z = torch.normal(mean=0, std=1, size=param.data.size(), device=param.data.device, dtype=param.data.dtype)
>
>                 Hessian_temp = self.Hessian_matrix[name] * z * z
>                 Hessian_estimator = (torch.abs(loss1+loss2-2 * loss_original)* Hessian_temp * Hessian_smooth /(2 * self.args.zo_eps*self.args.zo_eps))
>
>                 self.Hessian_matrix[name] = ((1-Hessian_smooth) * self.Hessian_matrix[name] +  Hessian_estimator)
>
>                 grad = (loss1-loss2)/(2 * self.args.zo_eps) * z / torch.sqrt(self.Hessian_matrix[name])
>                 param.data = param.data - zo_learning_rate * (grad + self.args.weight_decay * param.data)
>
>             loss_out = self.zo_forward(model, inputs)
>         return loss_out
> `
>
> The implementation of HiZOO is with the similar way of PaZO with the choice of $\alpha=1/2$. Specifically, they compute the moving average of diagonal Hessian instead diagonal Hessian inverse, and the grad is computed through: $(l_{+} - l_{-}) / (2\mu) * \Sigma_t^{-1/2}$. The only difference lies in:
>
> `
> Hessian_estimator = (torch.abs(loss1+loss2-2 * loss_original)* Hessian_temp * Hessian_smooth /(2 * self.args.zo_eps*self.args.zo_eps))
> `
>
> In which they didn't minus the identity matrix $I$.

---

> > ### Author Response · Authors · 2025-08-05
> >
> > We sincerely appreciate your insightful comments and thoughtful suggestions.  Your comments are well-grounded and we will clarify our discussion accordingly in the final version.  Thank you once again for your constructive feedback and for taking the time to engage so carefully.

---

### Official Review · Reviewer_DhFc · 2025-07-01

**Clarity:** 2
**Significance:** 2
**Originality:** 2
**Rating:** 4
**Confidence:** 2

**Summary:**

The paper proposes a preconditioned accelerated zeroth-order method, PaZO, for fine-tuning LLMs. Theoretically, the paper justifies the use of preconditioners for ZO methods by showing the convergence rate gap between with and without Hessian preconditioning. Motivated by the theory, the paper proposes a practical realization of PaZO using diagonal Hessian estimate and moving average. Numerical experiments on masked LLMs and generative LLMs demonstrate the memory and timestep efficiency of PaZO compared to first-order methods and ZO methods without preconditioning acceleration.

**Questions:**

1. In Theorem 3.5, can you provide more explanation on the relation between $D_\alpha$ and $d^2$? In particular, how does the lower bound scale with $\alpha$? While the paper claims that $\alpha=0$ leads to strict inequality and is thus suboptimal, it would be better to have more explicit characterizations.

2. In Theorem 3.8, how large is $\epsilon_0$? It seems to be treated as small so that $Err$ represents high-order errors, but how is it guaranteed?

3. In Appendix C.6, Table 6, why are $\beta_1$ and $\beta_2$ so small (around $10^{-8}$)? This would be a particular issue given that many LLMs use low precision (e.g., bfloat16) during training to save memory, and $10^{-8}$ seems to be too small. This makes the results less convincing.

4. How much more memory overhead does PaZO have compared to MeZO? Figure 1 uses a log scale, which makes such a comparison less transparent.

5. Is it possible to plot the validation loss vs steps? It would better illustrate the efficiency of PaZO compared to other methods.

6. What are the standard deviations in Table 6? From the numbers, PaZO seems to have marginal improvement over HiZOO, especially in the LoRA and prefix settings.

**Ethical Concerns:**

["NO or VERY MINOR ethics concerns only"]

**Final Justification:**

The authors' detailed responses address my questions.

**Limitations:**

1. Sensitivity to $\beta_1$ and $\beta_2$ should be mentioned in the main text, as they are important hyperparameters and are abnormally small.

**Paper Formatting Concerns:**

I do not notice any major formatting issues.

**Quality:**

2

**Strengths And Weaknesses:**

Strength:

1. Reducing the memory usage for LLM fine-tuning is an important topic and has significance in practice.

2. The paper is properly framed under questions A, B and C. It provides their method along with the motivating theory.

Weaknesses:

While the paper is well structured, its content leaves many puzzles lacking explanations. See Questions.

---

> ### Author Rebuttal · Authors · 2025-07-31
>
> # Response to Reviewer DhFc
>
> We sincerely thank the reviewer DhFc for the thoughtful and constructive feedback. We appreciate the time and effort spent in evaluating our work. We provide detailed responses to each of the raised concerns in the following part, and we will incorporate the corresponding revisions and discussions into the camera-ready version.
>
> ## 1. The explanation on the relation between $D_\alpha$ and $d^2$ (regarding Question 1)
> Thanks for your question on our clarification. In Theorem 3.5, the key parameter is $\alpha$. We define $\beta=2\alpha-1$. Let $\lambda\_1 \geq \lambda\_2 \geq \cdots \geq \lambda\_d > 0$ denote the eigenvalues of $\mathbf{H}\^*$. Consequently, $D\_\alpha$ can be written as:
> $$
> D_{\alpha}= \left(\sum_{i=1}^d\lambda_i^{\beta}\right)\cdot\left(\sum_{i=1}^d\lambda_i^{-\beta}\right).
> $$
>
> **Derivation of the Lower Bound.**  Applying the Cauchy-Schwarz inequality yields $D_{\alpha}\geq d^2$. Equality holds if and only if $\beta=0$ (which corresponds to $\alpha=1/2$), at which point $D_{\alpha}$ attains its theoretical minimum value of $d^2$.
>
> **Strict Inequality for $\alpha\neq 1/2$.** When $\alpha\neq 1/2$ (i.e.,$\beta\neq 0$), the equality condition for the Cauchy-Schwarz inequality – namely, that all eigenvalues $\lambda_i$ must be equal – is generally not satisfied (unless the model is isotropic). Consequently, $D_{\alpha}>d^2$ strictly. Moreover, the magnitude of $D_{\alpha}$ relative to $d^2$ increases under two conditions: (1) as $|\beta|$ increases (i.e., as $\alpha$ deviates further from 1/2), and (2) as the dispersion of the Hessian eigenvalues $\{\lambda_i\}$ increases.
>
> ## 2. The scale of $\epsilon_0$ and the guarantee (regarding Question 2)
> We thank the reviewer for highlighting this crucial assumption. The quantity $\epsilon_0$, introduced in the proof of Theorem 3.8, serves to bound the expected distance between $\mathbf{\theta}_t$ and $\mathbf{\theta}^*$.
>
> **Definition and Guarantee of** \$\epsilon\_0\$
> The magnitude of \$\epsilon\_0\$ reflects the degree to which the parameters \$\mathbf{\theta}\_t\$ (for \$t \geq T\$) approach the optimal solution $\mathbf{\theta}^*$ after \$T\$ iterations. Specifically, we assume that at iteration step \$t\$, the expectation satisfies:
>
> $$
> \mathbb{E}\left[\|\mathbf{\theta}_t - \mathbf{\theta}^*\|^p\right] \leq \epsilon_0^p.
> $$
>
> This inequality defines a neighborhood of radius \$\epsilon\_0\$ around \$\boldsymbol{\theta}^\*\$ within which the algorithm iterates. The achievability of such a neighborhood size \$\epsilon\_0\$ is guaranteed by the algorithm's prior convergence properties.
>
> **Prior Convergence Properties and the Scale of $\epsilon_0$.** Established analyses of Stochastic Gradient Descent (SGD) for non-convex optimization problems (e.g., [1-3]) demonstrate that, under appropriate conditions, the iterates $\mathbf{\theta}_t$ converge globally—either with high probability or in expectation—to a neighborhood containing the global optimum $\mathbf{\theta}^*$.
>
> $\epsilon_0$ can typically be bounded by a polynomial function of the step size $\eta$. By selecting a sufficiently large total iteration number $T$, we can ensure that $\epsilon_0$ becomes sufficiently small. This, in turn, guarantees that the higher-order term $\bar{\textbf{Err}}$
> in Equation (9) is asymptotically negligible relative to the dominant error term.
>
> ## 3. The scale of \$\beta\_1\$ and \$\beta\_2\$ in practice and the relation to low precision (regarding Question 3 and Limitations 1)
>
> Thank you for pointing out the lack of clarification about the choice of \$\beta\_1\$ and \$\beta\_2\$. We explain the necessity of small choices of \$\beta\_1\$ and \$\beta\_2\$ in practice. For \$\beta\_1\$, we first want to clarify that the correction term \$\tilde g \circ \tilde g\$ is designed to reduce the instability of training caused by the outliers from zeroth-order oracle with randomness. Specifically, due to the order of preconditioner is \$-1/2\$ (\$\alpha=1/2\$), too small value in the diagonal estimated Hessian may cause numerical instability (NaN) in training process. The correction term encourages numerical alignment between gradient magnitude and adaptive curvature scaling like Adam. However, applying such a correction causes bias in Equation (14). By the choice of small \$\beta\_1\$ to control the bias, we find that such an almost negligible term enhances the robustness of the estimation against outliers and ensure the stability of training. As the results in ablation study, when \$\beta\_1\$ is set too large (e.g., 1, 1e-2), the optimization diverges and results in NaNs, because the proportion of correction term is too large to introduce non-negligible bias. When \$\beta\_1\$ is too small (e.g., 0 or approaching 0), the correction term fails to take effect, which may lead to numerical instability during training and result in NaNs. Only in a proper range (around \$1e\text{-}8\$ to \$1e\text{-}10\$) does the algorithm yield reasonable performance.
>
> For \$\beta\_2\$, it is originated from the large variance of estimated Hessian by zeroth-order oracle. We propose a non-strict analysis of variance to clarify it further as below.
> From an intuitive (though informal) perspective, consider the update rule:
> \$\Sigma\_t = (1 - \beta\_2)\Sigma\_{t-1} + \beta\_2 |\widehat{\Sigma}\_t|\$,
> where \$\widehat{\Sigma}\_t\$ denotes the current estimate of the (diagonal) Hessian, and the initialization \$\Sigma\_0 = I\$. Since the iterative update forms the moving average, \$\beta\_2\$ effectively controls the sensitivity to new information and hence the variance of the sequence \${\Sigma\_t}\$. Specifically, due to the identity initialization, the influence of \$\Sigma\_0 = I\$ remains dominant unless \$\beta\_2\$ is set large. In particular, when \$\beta\_2\$ is small, the update rule behaves approximately as:
>
> $$
> \Sigma_t \approx (1 - \beta_2)^t I + \beta_2 \sum_{k=0}^{t-1} (1 - \beta_2)^k |\widehat{\Sigma}_{t-k}|,
> $$
>
> which can be interpreted as a weighted combination of the initial value \$I\$ and a series of estimated observations.
> We consider the variance of \$\Sigma\_t\$ above by the analysis on the second term (though not a completely strict analysis, as some correlations are ignored, we hope to provide some intuition here). The expectation of each \$\widehat{\Sigma}\_{t-k}\$ can be directly obtained as:
>
> $$
> \mathbb{E}\left[\underbrace{\frac{1}{2}u^\top \widehat\Sigma^{-\frac{1}{2}} \mathbf{H} \widehat\Sigma^{-\frac{1}{2}} u}_{\mathcal{I}} \cdot \widehat\Sigma\left(\text{diag}(uu^\top) - I\right)\right] = \text{diag}(\mathbf{H}).
> $$
>
> Thus, we can obtain the variance of each coordinate \$i\$ of \$\widehat \Sigma\$ as:
>
> $$
> \text{Var}[\hat h_{ii}] \approx \widehat\Sigma_{ii}^2 \mathbb{E}\left[\mathcal{I}^2 (u_i^2 -1)^2 \right] - H_{ii}^2 = \frac{\widehat\Sigma_{ii}^2}{4} \sum_{j,k,s,t} A_{jk} A_{st} \mathbb{E}[u_ju_ku_su_t(u_i^2-1)^2] - H_{ii}^2,
> $$
>
>
> where \$\hat h = \mathcal{I}{\widehat\Sigma}\left(\text{diag}(uu^\top) - I\right)\$, \$A = \widehat\Sigma^{-1/2}H\widehat\Sigma^{-1/2}\$.
> Noticing that when \$u \sim \mathcal{N}(0,1)\$, the coupling term \$\mathbb{E}\[u\_ju\_ku\_su\_t(u\_i^2-1)^2]\$ is not 0 with special choice of \$j,k,s,t\$. For example, only considering the terms where \$j=s\$ and \$k=t\$, and denoting for any \$k \leq t\$, \$\max\_i (\widehat\Sigma\_k)\_{ii} \leq B\$, we have:
>
> $$
> \text{Var}[\widehat\Sigma_k] \approx \mathcal{O}(d^2B^2).
> $$
>
> **It means that some values in estimated Hessian is numerically too large to introduce the risk of obtaining outliers in training.**
> To solve this, considering:
>
> $$
> \beta_2 \sum_{k=0}^{t-1} (1 - \beta_2)^k = 1 - (1 - \beta_2)^t,
> $$
>
> a small \$\beta\_2\$ yields the controlled variance of the second term to ensure a lower-variance, more stable update in the training process. Meanwhile, small \$\beta\_2\$ still ensures that the Hessian estimates are valid from a global perspective, because when \$t \to \infty\$, the first term \$(1 - \beta\_2)^t \to 0\$ and the second term \$\beta\_2 \sum\_{k=0}^{t-1} (1 - \beta\_2)^k \to 1\$.
> Similar hyper-parameter setting is used in another zeroth-order optimizers for fine-tuning like \[4]. Our experimental results also demonstrate that under the same parameter configurations, PaZO improves the performance of fine-tuned model across tasks. We will mention the sensitivity and choice of \$\beta\_1\$ and \$\beta\_2\$ in the main text of camera-ready version.
>
> ---
>
> Moreover, your concerns about how small hyperparameters are combined with low precision training like bfloat16 are extremely important. The experiment setting in zeroth-order optimization is a little different while low-precision formats like bfloat16 are widely adopted in first-order optimization to reduce memory usage. **In both the pioneering work of applying zeroth-order optimization to large language model finetuning \[5], as well as subsequent follow-up studies such as \[4], experiments are by default conducted in fp32 without bfloat16 to ensure numerical stability and reliable convergence.** We follow the default experimental setting for fair comparison. Specifically, zeroth-order methods already lead to a significant reduction in memory usage due to its bp free—often far more than what bfloat16 achieves in standard first-order methods. In a sense, zeroth-order optimization can be viewed as an aggressive strategy to trade off between precision and memory efficiency at the algorithmic level. When applying bfloat16 in the context of zeroth-order optimization, the choice of hyperparameters may need to be adjusted accordingly to prevent loss of numerical precision. We believe this is an interesting direction for future research, but it is out of the scope of this paper.
>
>
>
>
>
>
>
> Due to space limitations, I will immediately post the responses to the remaining questions starting from the next stage. Thank you for your insightful comments again.

---

> > ### Author Response · Authors · 2025-08-01
> >
> > Thank you for your patience in reading our response. We discuss some questions you mentioned as supplements below.
> >
> > ## 4. Memory overhead of PaZO and MeZO (regarding Question 4)
> >
> > We thank the reviewer for pointing out the lack of clarity in Figure 1 due to the use of a logarithmic scale. We just followed previous works [4,5] to design this figure. To further improve the presentation, we will revise Figure 1 in the camera-ready version to include the raw values. To clarify, PaZO incurs approximately 2 times more memory overhead compared to MeZO, and maintains the memory overhead of HiZOO, primarily due to the need to store the estimate of the diagonal Hessian as below.
> >
> > | Peak Memory (GB) |  1.3B | 2.7B | 6.7B | 13B |
> > | -- | -- | -- | -- | -- |
> > |MeZO |4 |7 |14 | 26|
> > |PaZO |9 |14 |30 |54 |
> >
> > However, we would like to emphasize that PaZO still requires significantly less memory than first-order optimizers, such as Adam, since zeroth-order methods do not involve backpropagation. This is reflected in the multiplicative ratios explicitly annotated in Figure 1. As the model size increases, the memory efficiency of PaZO becomes increasingly prominent compared with FT. We believe PaZO represents a theoretically ideal algorithm within the zeroth-order paradigm, while still preserving a substantial memory advantage over first-order methods — a trade-off we consider both meaningful and impactful for practical deployment. Moreover, if with the continued advancement of AI infrastructure, sharding techniques can be effectively applied to zeroth-order optimization, then the additional memory overhead caused by storing the estimated Hessian could be significantly mitigated through engineering techniques. Meanwhile, the contribution of our algorithm in terms of preconditioning order would still remain effective.
> >
> >
> >
> >
> > ## 5. The validation loss v.s. steps (regarding Question 5)
> >
> > We greatly appreciate your insightful suggestion. Unfortunately, due to the rebuttal policy, we are not allowed to submit any PDF files or external links. As a result, we are unable to include the validation loss vs. steps plot at this stage. However, we would like to offer two points of discussion that may help address this concern. First, all of our experiments are conducted in fine-tuning tasks. This means that, rather than focusing solely on minimizing the final validation loss value, our primary objective is to improve the performance of the fine-tuned model on downstream tasks—as shown in Tables 1 and 2. For fine-tuning, a lower loss does not necessarily imply better task performance. What we seek is a learning algorithm that effectively enhances model utility. Second, we can briefly describe an example of the validation loss v.s. steps: considering fine-tuning OPT-1.3B on RTE, the steps by which the validation loss achieves 0.62 are roughly 8000, 6000 and 5000 of MeZO, HiZOO and PaZO, respectively. These illustrate the efficiency of PaZO compared to others to some degree. We will be sure to include such curves and visualizations in the camera-ready version to better illustrate the efficiency of PaZO.

---

> > ### Comment · Reviewer_DhFc · 2025-08-02
> >
> > I thank the authors for their detailed rebuttal, and it pretty well addresses my concerns 1-2 and 4-6. However, I still have concerns about 3, i.e., the tiny choice of $\beta_1$, $\beta_2$ and the related numerical instability. Conceptually, I understand the necessity of choosing properly small $\beta_1$ and $\beta_2$. But the values are just too small, and it's hard to believe that they are required for numerical stability given that the initial estimate $I$ is quite large and the moving average is taken with absolute value of $\Sigma_t$ (Step 3 in Algorithm 1).
> >
> > In particular, NaN happens with $(\beta_1,\beta_2)=(10^{-6},10^{-8})$, while the total number of optimization steps is 20k (line 256). Even if we set all new terms to be all zero, the initial identity matrix should still give at least $(1-10^{-6})^{20000}\approx 0.98$ to the averaged diagonal Hessian, which I don't think will results in severe numerical issues. Therefore, the result looks strange to me. I wish the authors can explain more on my specific questions in the following:
> >
> > 3.1: When does the instability happen? Does it happen at a very early stage or very late? And what is the magnitude of estimated Hessian at time it starts to collapse?
> >
> > 3.2: How does $\mu$ affect the result? It is in the denominator so could affect the estimate values.
> >
> > 3.3: Can the numerical instability be solved by simply replacing the square root inverse by numerically stable methods? Like solving the linear equation $\Sigma^{1/2}x=u$ to get $\Sigma^{-1/2}u$ by gradient methods. This could potentially prevent the use of too small $\beta$'s.
> >
> > I would like to raise my score to accept if the authors could resolve my questions. Thanks.

---

> > > ### Author Response · Authors · 2025-08-04
> > >
> > > We sincerely thank the reviewer for carefully reading our previous response and for raising additional insightful questions, particularly regarding the magnitude of the $\beta$ parameters. Below, we address each of the newly raised concerns in detail.
> > >
> > > ## 1. Details of instability (regarding Question 3.1)
> > >
> > > The instability happens in the early stage in our observation. Specifically, for example, considering the hyperparameters setting $(\beta_1, \beta_2)= (10^{-6},10^{-8})$ , after 35 iterations from beginning, the instability happens due to the numerical explosion of estimated Hessian after updating by $\beta_1$. The F-norm of the diagonal estmated Hessian rapidly changes from $10^2$ level to inf at time it starts to collapse. As you pointed out, the theoretical intuition suggests that as long as $\beta \in (0, 1)$, the proposed method should work properly. However, in practice, due to the involvement of random Gaussian vectors in the estimation process, and more importantly, the fact that most second-order information is computed based on squared Gaussian terms, even a single coordinate with an unusually large magnitude can result in extremely large outliers. Without a proper scaling, such an outlier can cause the estimated Hessian diagonal to explode, ultimately leading to NaN in parameter updates and destabilizing subsequent training. We acknowledge that this instability is a practical issue rather than a theoretical flaw. To mitigate it, we carefully designed a proper choice of $\beta$  that effectively suppress such rare but harmful events. While this solution may appear strange from an intuitively theoretical perspective, it is essential for maintaining training stability in real tasks.
> > >
> > > ## 2. How $\mu$ affects the results (regarding Question 3.2)
> > >
> > > Thank you for pointing out this important issue. The choice of $\mu$ is indeed critical. For zeroth-order oracles, the gap between the estimated quantity and the true value is known to be on the order of $\mathcal{O}(\mu)$. As $\mu \to 0$, the estimator theoretically converges to the true value. However, in practical implementations, a large $\mu$ leads to less accurate estimates, while a small $\mu$ increases the risk of numerical issues because $\mu$ often appears in the denominator. We conduct an ablation study on $\mu$ on fine-tuning OPT-1.3B on SST2 as below.
> > >
> > > | $\mu$ | 1e-1 | 1e-2 |1e-3 |1e-4 |
> > > | -- | -- | -- | -- | -- |
> > > | Acc | 54.0|66.9 | 89.0| 86.4|
> > >
> > > The results show that too large a choice of $\mu$ leads to less accuracy. Moreover, when setting too small $\mu$ like 1e-8, NaN happens in our experiments. Therefore, we believe that $\mu$ should be chosen within a proper range to ensure a good balance between estimation accuracy and training stability. We report the choice of $\mu$ in Table 3 and Table 4 in Appendix C. We will add the ablation study on $\mu$ in the camera-ready version.
> > >
> > >
> > > ## 3. About other numerical stable methods (regarding Question 3.3)
> > >
> > > We appreciate the reviewer’s insightful suggestion. In principle, it is possible to obtain $\Sigma^{-1/2} u$ by solving the linear equation $\Sigma^{1/2} x = u$ using gradient descent. However, in our setting, this strategy poses a critical limitation that solving $\Sigma^{1/2} x = u$ requires another gradient descent iterations for the subproblem, resulting in unacceptable time cost in practice. Specifically, when fine-tuning OPT-1.3B on the SST2 dataset, each iteration used in our proposed method takes approximately **0.3837 seconds**, whereas the iteration with solving the linear subproblem to obtain $\Sigma^{-1/2} u$ requires **5.35 seconds**, due to the need for sufficiently accurate gradient descent steps to ensure a reliable solution to the subproblem. This represents nearly a **14× increase in time cost**, which we consider unacceptable in real tasks. Although solving the subproblem can indeed improve numerical stability to some extent (e.g., for $(\beta_1, \beta_2)= (10^{-6},10^{-8})$), the trade-off in computational efficiency makes it impractical for real tasks. Moreover, because $\Sigma$ is positive and diagonally defined, we can directly obtain the solution of such a linear equation, which is aligned with our practical form algorithm. This means that it may be difficult to achieve essential improvement by solving such a sub-problem, and a small $\beta$ is still necessary. Nonetheless, we sincerely appreciate your insightful suggestion.
> > >
> > >
> > > In summary, we would like to once again thank you for your thoughtful reading of our response and for raising constructive questions. We sincerely hope that our detailed replies have effectively addressed your concerns. We would appreciate it if you could consider raising your score.

---

> > > > ### Comment · Reviewer_DhFc · 2025-08-04
> > > >
> > > > I thank the authors for their response which addresses my concerns in 3.2 and 3.3. I raised my score.
> > > >
> > > > Additional comment on 3.1: previously I thought the numerical instability comes from inversion of too small $\Sigma$, but here it seems to be due to too large $\Sigma$. It is strange to me that it causes NaN rather than zero update, because $\Sigma$ in perturb parameters and preconditioning are in the form of inversion, which should give 0 but not NaN. I think this issue is worth more detailed investigation.

---

> > > > > ### Author Response · Authors · 2025-08-05
> > > > >
> > > > > We sincerely thank you for your constructive suggestions and your support of our work. Your additional comment is reasonable. In practice, as the Hessian update is computed before the inversion step, excessively large values can compromise the effectiveness of subsequent computations and lead to instability. We agree that it is worth further investigation. Once again, we deeply appreciate your insightful feedback and support.

---

> ### Author Response · Authors · 2025-08-01
>
> ## 6. The standard deviations in Table 6 and improvement with LoRA and prefix (regarding Question 6)
>
> We thank the reviewer for this important question. In Table 6, our primary objective was to demonstrate the stable operating range for the hyperparameters $\beta_1$ and $\beta_2$, rather than to emphasize performance differences. Specifically, we observed that when
> $\beta_1$ and $\beta_2$ fall outside the proper range, the training often becomes unstable (e.g., resulting in NaNs). Therefore, we reported the mean performance within the stable region to guide practical usage. In fact, the standard deviations in these experiments are very small (below 0.2 in accuracy), and they do not affect the conclusions regarding proper $\beta_1$ and $\beta_2$ settings. For completeness, we provide the results including standard deviations below (here we correct the typo mentioned in 3).
>
> | $\beta_1(\beta_2)$ | 1 | 1e-2 |1e-4 |1e-6 | 1e-8 |1e-10 |
> | -- | -- | -- | -- | -- | -- | -- |
> | setting $\beta_1$ = 1e-8 | NaN | NaN | NaN| 88.9 ($\pm 0.3$) | 89.0 ($\pm 0.2 $) | 88.9 ($\pm 0.2$) |
> | setting $\beta_2$ = 1e-8 | NaN | NaN | NaN| NaN | 89.0 ($\pm 0.1$) | 88.9 ($\pm 0.2$) |
>
> This result clearly demonstrates that the algorithm only runs robustly when the hyperparameters are within a suitable range. When the parameters are set too large, the algorithm tends to become unstable and may even diverge.
>
> Moreover, we would like to clarify that our primary goal is to design a zeroth-order optimizer with theoretically ideal convergence in zeroth-order optimization and practical memory efficiency compared with first-order optimization. As you mentioned, just for more difficult settings like PEFT (LoRA, prefix tuning), such improvements seem to be minor. However, these PEFT setups naturally constrain the model capacity, making it more difficult for any optimizer to yield large performance gains. Despite this, PaZO also achieves consistent improvements over HiZOO across types of models, dataset and tuning methods. Moreover, for RoBERTa-large(350M), the improvement of PaZO with PEFT is not marginal.  Nonetheless, PaZO maintains stable and robust performance across various settings, and we believe this consistency, combined with its theoretical and practical advantages, makes it a compelling choice.
>
>
> In summary, we truly appreciate the reviewer’s insightful questions and suggestions. In the camera-ready version, we will incorporate the clarifications above, including the discussion on theoretical analysis, hyperparameter choices and robustness (regarding Table 6), the explanation of memory overhead (regarding Figure 1), and figures regarding validation loss curves against steps which we cannot show here due to the rebuttal policy. We will also include the standard deviations. Your valuable comments have greatly helped us improve the clarity and completeness of our work. Hope our responses and discussions can address your concerns. We would be sincerely grateful if the reviewer DhFc would consider increasing the score in light of the above clarifications and revisions.
>
> ## Reference
>
> [1]  HaoChen, J. Z., Wei, C., Lee, J., & Ma, T. (2021). Shape matters: Understanding the implicit bias of the noise covariance. In Conference on learning theory.
>
> [2]  Li, Y., Ma, T., & Zhang, H. (2018). Algorithmic regularization in over-parameterized matrix sensing and neural networks with quadratic activations. In Conference on learning theory.
>
> [3]  Xiong, N., Ding, L., & Du, S. S. (2023). How over-parameterization slows down gradient descent in matrix sensing: The curses of symmetry and initialization. In The twelfth international conference on learning representations.
>
> [4]  Yanjun Zhao et al. Second-order fine-tuning without pain for llms: A hessian informed zeroth-order optimizer. In: arXiv preprint arXiv:2402.15173 (2024).
>
> [5] Sadhika Malladi et al.Fine-tuning language models with just forward passes.In:Advances in Neural Information Processing Systems 36(2023),pp.53038–53075.

---

### Official Review · Reviewer_bWDk · 2025-07-03

**Clarity:** 3
**Significance:** 3
**Originality:** 3
**Rating:** 4
**Confidence:** 3

**Summary:**

This paper introduces PaZO, a preconditioned zeroth-order optimization algorithm designed for the memory-efficient fine-tuning of LLMs. The authors provide a solid theoretical foundation, demonstrating the necessity of preconditioning in ZO optimization to achieve faster convergence. They derive the optimal preconditioner and develop a practical algorithm that estimates the diagonal Hessian using only forward passes. The experiments show that PaZO outperforms existing ZO methods like MeZO and HiZOO and achieves performance comparable to first-order fine-tuning while using significantly less memory.

**Questions:**

To strengthen the claims about "convergence speed", could you provide practical "time-to-accuracy" results? For example, on a representative task like SST-2, how much wall-clock time does it take for PaZO, MeZO, and FT to reach 95% of their respective final accuracies?

**Ethical Concerns:**

["NO or VERY MINOR ethics concerns only"]

**Limitations:**

I did not see potential negative societal impact.

**Paper Formatting Concerns:**

No formatting concerns.

**Quality:**

3

**Strengths And Weaknesses:**

Pros:

- The practical PaZO algorithm is an application of the theoretical insights. The design choices, such as estimating the diagonal Hessian to maintain O(d) memory and using moving averages and resets for stability, are well-justified and make the algorithm practical for fine-tuning modern LLMs.

- The core theoretical strength is not just that the authors prove preconditioning is necessary, but that they derive the optimal form of the preconditioner from a formal convergence analysis. The technical argument in Theorem 3.5 is particularly elegant.

- The paper's main strength lies in its rigorous theoretical analysis. It methodically answers three key questions about preconditioning in ZO optimization. The convergence analysis in Theorems 3.5 and 3.8 provides a clear, formal justification for why preconditioning is necessary and demonstrates that using the inverse square root of the Hessian (H^{-1/2}) as a preconditioner is optimal for achieving the fastest convergence rate. This is a significant theoretical contribution.

Cons:

- The authors are transparent about PaZO's per-step time, noting it is about 1.5x slower than MeZO due to an additional forward pass (Figure 2). They argue this is offset by a better convergence rate. However, the analysis would be much stronger with a direct comparison of the total time (or number of steps) required to reach a target accuracy. A plot of accuracy vs. wall-clock time for PaZO against the baselines would more directly substantiate the claim of overall faster training.

- While the theoretical framework is novel, some components of the practical algorithm in Section 4 are based on existing techniques. For example, estimating the Hessian diagonal via forward passes and using moving averages to stabilize training are known methods. The main novelty is the specific formulation justified by the theory for alpha = 1/2, which is a strong contribution, but the practical implementation itself is a clever combination of established components rather than a completely new mechanism.

---

> ### Author Rebuttal · Authors · 2025-07-31
>
> # Response to Reviewer bWDk
>
> We sincerely thank reviewer bWDk for the positive evaluation and thoughtful comments on our work. We greatly appreciate your recognition of the theoretical contribution and the careful reading of both the algorithm design and the empirical results. We find the comments valuable in helping us improve the clarity of the paper. We provide detailed responses to each of the raised concerns in the following part, and we will incorporate the corresponding revisions and discussions into the camera-ready version.
>
> ## 1. Report on wall-clock time v.s. accuracy (regarding Cons 1 and Question)
>
> We sincerely thank the reviewer for proposing this important point. We fully agree that comparing the wall-clock time to reach a target accuracy would more directly demonstrate the overall training efficiency of PaZO. However, due to rebuttal policy constraints, we regret that we are unable to include any PDF figures or external links in our response, and therefore cannot directly present the accuracy vs. wall-clock time plot at this stage. As a complementary reference you mentioned, we instead report the steps and wall-clock time (in seconds) required to reach 60% accuracy on a representative task RTE in the table below:
>
> |  |  MeZO | HiZOO|  PaZO |
> | -- | -- | -- | -- |
> |Steps | 15000 | 10000| 9000 |
> |Wall-clock Time (s) |  3848| 3785|3453|
>
> These results support our claim that PaZO achieves better convergence speed than existing zeroth-order methods by leveraging an ideal choice of the preconditioner order. Both the required number of steps and the total training time to reach the target accuracy are smaller for PaZO, validating our theoretical insights. Moreover, although the per-step cost of PaZO is slightly higher than MeZO—as we transparently report in the paper—this is more than offset by its improved convergence rate. In particular, PaZO reduces the total wall-clock time by approximately 10% compared to MeZO, demonstrating that it is not only theoretically sound but also efficient in practical settings. We will include these additional wall-clock time comparisons and corresponding wall-clock time v.s. steps plots in the camera-ready version.
>
> ## 2. The components of our algorithm and our main contributions (regarding Cons 2)
>
> We sincerely thank the reviewer for the insightful comment. As you mentioned, some components of our practical implementation, such as diagonal Hessian estimation via zeroth-order oracles and the use of moving averages, are adapted from existing techniques. However, as emphasized in the Introduction, our main focus lies in addressing three questions including:
> (1) Is preconditioning theoretically beneficial in zeroth-order optimization?
> (2) If so, what is the optimal order to obtain ideal convergence?
> (3) How can we translate this theoretical insight into practical algorithm design? While previous works have proposed related techniques and components, they do not provide clear theoretical answers to the above questions. In contrast, our work presents a novel analysis framework that leads to a principled justification for preconditioning in zeroth-order optimization, particularly when the preconditioner order is set to $\alpha=1/2$, and demonstrates how this insight can be effectively implemented in practice. We believe that this new perspective offers substantial theoretical and practical value, and may serve as a useful work for better understanding and further advancing the field of zeroth-order optimization.
>
> We sincerely appreciate the reviewer’s positive evaluation of our work. Your thoughtful comments and recognition of both the theoretical contributions and practical relevance of PaZO are highly encouraging to us. We will incorporate the additional experimental details on wall-clock time above into the camera-ready version to further strengthen the presentation. We would be deeply grateful if the reviewer bWDk could consider a higher score.

---

### Official Review · Reviewer_Gw49 · 2025-07-06

**Clarity:** 3
**Significance:** 3
**Originality:** 3
**Rating:** 5
**Confidence:** 3

**Summary:**

The paper introduces PaZO, a preconditioned accelerated zeroth-order optimization algorithm designed for fine-tuning large language models (LLMs) without backpropagation. The authors address three core challenges: (A) whether preconditioning is necessary in zeroth-order optimization, (B) how to optimally apply preconditioning for fastest convergence, and (C) how to practically estimate Hessian information to improve performance. They theoretically prove that zeroth-order stochastic gradient descent (ZO-SGD) alone is suboptimal and that using a preconditioner of order −1/2 yields the best convergence rate. PaZO incorporates this insight into a practical algorithm using diagonal Hessian estimates and moving averages for stability. Extensive experiments on models like RoBERTa-large and OPT-1.3B across various tasks and parameter-efficient fine-tuning (PEFT) methods demonstrate that PaZO outperforms existing zeroth-order methods (e.g., MeZO, HiZOO) in both accuracy and memory efficiency, achieving state-of-the-art results with manageable computational overhead.

**Questions:**

- How does PaZO compare to MeZO and FO methods on more recent LLMs, such as Llama 3?
- Can the authors show learning curves of the experiments reported in Tables 1 and 2, it would be curious to see how fast each method converges.

**Ethical Concerns:**

["NO or VERY MINOR ethics concerns only"]

**Final Justification:**

After the authors' rebuttal most of my questions/concerns have been addressed, and I have updated my score from 4 to 5.

**Limitations:**

The authors did not properly asses the limitations of their work.

**Quality:**

2

**Strengths And Weaknesses:**

Strengths:
- The paper is well written, and tackles an important problem: efficient LLM fine-tuning
- The authors rigorously prove that standard zeroth-order stochastic gradient descent (ZO-SGD) cannot achieve the optimal convergence rate, establishing the necessity of preconditioning in zeroth-order optimization. They introduce a generalized framework—Preconditioned Simultaneous Perturbation Stochastic Approximation (PSPSA)—and derive convergence bounds for both quadratic and general smooth functions. A key strength is the identification of the optimal preconditioning order (α = 1/2), which is shown to yield the fastest convergence rate. These results are supported by detailed proofs and assumptions, including extensions to scenarios with approximate Hessians.
- The authors present a practical algorithm, PaZO, and demonstrate its effectiveness empirically across a wide range of downstream tasks,  model architectures, and fine-tuning strategies.

Weaknesses:
- The practical version of PaZO is not so practical. It strongly resembles HiZOO, with slight differences in estimating the Hessian, and introduces extra hyperparameters ( $T_0$, $\beta_1$ and $\beta_2$) that are not properly studied. The choice of $T_0$ is not properly ablated, and Table 6 in the appendix shows that the PaZO is highly sensetive to the choices of $\beta_1$ and $\beta_2$.
- The authors claim that PaZO maintains the memory efficiency of HiZOO, but Figure 1 does not show the memory overhead of HiZOO.
- The performance improvement of PaZO over HiZOO is only demonstrated for OPT (and not RoBERTA). For OPT, the improvement does not seem to be significant.

---

> ### Author Rebuttal · Authors · 2025-07-31
>
> # Response to Reviewer Gw49
>
> We sincerely thank the reviewer Gw49 for the constructive and thoughtful feedback. We greatly appreciate your recognition of our work and your valuable suggestions. We provide point-by-point responses to each of your comments.
>
> ## 1. The hyperparameters in the practical version of PaZO (regarding Weaknesses 1)
>
> Thank you very much for pointing out this important issue and helping us improve the clarity of our presentation. First, we would like to clarify that although our algorithm shares some structural similarities with HiZOO in terms of estimating the Hessian diagonal, there is a fundamental difference in the core design. Specifically, our method is guided by theoretical analysis, which leads to the choice of the preconditioner order $\alpha=1/2$ in contrast to HiZOO’s $\alpha=-1/2$. This results in essential differences in how the optimization trajectory evolves. While the Hessian estimation procedures may appear similar, the use of the preconditioner are theoretically justified and practically distinct in PaZO.
>
> We also sincerely thank the reviewer for pointing out that our discussion of hyperparameters could be more comprehensive. We now provide additional clarification. First, we correct a typo in Table 6 of the appendix: the second column in the first row should be 1 instead of 0. We apologize for the confusion this may have caused. Then, regarding the role of the hyperparameters, it is sensitive to $\beta_1, \beta_2$ because $\beta_1$ controls the strength of the additive correction to the Hessian diagonal estimate. This correction ensures that the scaling effect in our algorithm behaves similarly to Adam-style updates, helping maintain numerical stability during training. $\beta_2$ governs the moving average applied to the Hessian diagonal estimates. This is a common technique to stabilize estimates over time and smooth the update directions from the zeroth-order oracle.
>
> $T_0$ determines the frequency at which the preconditioner is reset. We agree that the original manuscript lacked a dedicated ablation for this parameter. We have now conducted additional experiments to investigate the effect of varying $T_0$, and we provide the results and analysis below.
>
> In this experiment, we fine-tune the OPT-1.3B model and evaluate its performance on the SST2 dataset, while varying the value $T_0 \in\{64, 128,256, 512 \} $ and $\infty$ (without resetting). We aim to examine how the frequency of resetting the moving average of the preconditioning matrix
> influences training stability and final accuracy. The results are summarized in the table below:
>
>
> | $T_0$| 64| 128 |256 | 512 | $\infty$ |
> | -- | -- |--| -- | -- | -- |
> |SST2  |88.5|88.8 |89.0 | 88.3| 87.7|
>
> The reason behind this design and observation is as follows. As the number of iterations increases, the error in the Hessian estimation obtained from the zeroth-order oracle tends to accumulate over steps. This accumulated error may grow with the number of iterations. Therefore, when no resetting mechanism is used (as in the case of $\infty$ in the table above), the performance drop when overly old historical accumulated error may mislead the current update direction, ultimately harming final performance. On the other hand,  considering that $\Sigma$ is initialized as the identity matrix $I$, there exists a gap between $I$ and the Hessian $H^*$. When the resetting frequency is too small (e.g., 128), the information accumulated in $\Sigma_t$ is insufficient to effectively bridge this gap. The inaccurate estimation similarly leads to worse final model performance. The hyperparameter $T_0$ plays the role of a trade-off between error accumulation and Hessian estimation accuracy. Our experimental results also indicate that the model achieves the best performance when $T_0$ falls within a certain range. We will discuss more about the ablation study on $T_0$ in the camera-ready version. Thank you again for your feedback.
>
>
> ## 2. The memory overhead comparison between PaZO and HiZOO (regarding Weaknesses 2)
>
> Thank you for pointing out the missing comparison in Figure 1. Our statement that PaZO maintains the memory efficiency of HiZOO is meant in the literal sense: both HiZOO and PaZO incur additional memory overhead primarily due to storing the diagonal estimate of the Hessian, which is of the same size in the same model. Therefore, the peak memory consumption of the two algorithms is nearly identical. For example, when fine-tuning OPT-1.3B on MultiRC, both PaZO and HiZOO have a peak memory usage of approximately 9 GB. We will update Figure 1 in the camera-ready version to include HiZOO for completeness and clarity. Thank you again for your careful review.
>
> ## 3. More comparison of PaZO with another methods (regarding Weakness 3 and Question 1)
>
> We sincerely apologize for the limited scope of our experiments due to hardware constraints. However, we believe that the existing results are still indicative of the strengths of our method. While the improvement of PaZO over HiZOO on OPT is not dramatic, it is consistent across the majority of tasks. Moreover, PaZO shows improvement over HiZOO under different tuning paradigms, including LoRA and prefix tuning. This consistency suggests that the advantage of PaZO is not limited to a specific task or setup, but rather stems from a more principled design. This aligns well with the goal of our work to investigate how to better utilize preconditioning in zeroth-order optimization, and our theoretical analysis, which motivates a carefully chosen preconditioning order. We have indeed considered conducting experiments on LLaMA3. However, due to current hardware limitations, we were unable to properly apply this model. Now we are actively working to overcome these constraints and are making every effort to complete the experiments. If time permits, we may be able to report partial results during the review process, and we will include the complete results in the camera-ready version. We sincerely thank the reviewer for raising this point and appreciate your understanding.
>
> ## 4. The learning curve of experiments (regarding Question 2)
>
> We sincerely thank the reviewer for raising this important question. However, due to the rebuttal policy, we are unfortunately not allowed to include any PDF files or external links in our response, and thus we are unable to directly provide the learning curves here. However, we would like to explain that all of our experiments are conducted in fine-tuning tasks. This means that, rather than focusing solely on minimizing the final loss value, our primary objective is to improve the performance of the fine-tuned model on downstream tasks—as shown in Tables 1 and 2. For fine-tuning, a lower loss does not necessarily imply better task performance.
>
> As an alternative way, we can briefly describe an example of the validation loss v.s. steps: considering fine-tuning OPT-1.3B on RTE, the steps by which the validation loss achieves 0.62 are roughly 8000, 6000 and 5000 of MeZO, HiZOO and PaZO, respectively. These illustrate the efficiency of PaZO compared to others to some degree. We will include detailed learning curves for the experiments in Tables 1 and 2 in the camera-ready version to better illustrate the training dynamics of different methods.
>
>
> Once again, we sincerely thank the reviewer Gw49 for the thoughtful comments and constructive suggestions. We have carefully addressed each of the concerns raised, and we truly appreciate the opportunity to clarify and improve our work. We are grateful for the reviewer’s recognition of our contributions.  We would be deeply grateful if the reviewer Gw49 could consider a higher score.

---

> > ### Comment · Reviewer_Gw49 · 2025-08-05
> >
> > I thank the authors for their detailed rebuttal, as it has addressed most of my concerns. I highly encourage the authors to incorporate the suggested changes to their final manuscript. I will increase my score by 1.

---

> > > ### Author Response · Authors · 2025-08-06
> > >
> > > We sincerely thank you for your valuable comments and thoughtful suggestions, as well as your support for our work. We will make sure to incorporate the suggested changes and discussions into the final version. Thank you once again for your constructive feedback.

---

### Decision · Program_Chairs · 2025-09-17

**Decision:**

Accept (poster)

**Comment:**

This paper proposes PaZO, a preconditioned accelerated zeroth-order optimization method for fine-tuning LLMs. The work contributes both theoretical insights and a practical algorithm that incorporates diagonal Hessian estimates and moving averages for stability. Experiments demonstrate advantages over zeroth-order baselines like MeZO and HiZOO.

During the discussion, one reviewer stressed that PaZO’s practical design closely resembles HiZOO, raising concerns about algorithmic novelty. Additional issues included formulaic errors, hyperparameter sensitivity, and limited experimental breadth -- improvements are most evident on OPT, less so on RoBERTa, with no evaluation on newer LLMs (e.g., LLaMA3). While the authors addressed efficiency concerns in rebuttal, complete evidence is needed for the revised version.

Despite reservations about novelty and presentation, the paper offers solid theoretical contributions. Given that ZO fine-tuning relies only on model queries and represents a promising direction, and with overall ratings turning positive after rebuttal, the AC believes the work will interest both optimization and LLM communities. The AC strongly encourages the authors to incorporate the planned clarifications and additional results in the revised version.